# Bypassing the Simulator:
# Near-Optimal Adversarial Linear Contextual Bandits

**Haolin Liu***
University of Virginia
srs8rh@virginia.edu

**Chen-Yu Wei***[†]
University of Virginia
chenyu.wei@virginia.edu

**Julian Zimmert***
Google Research
zimmert@google.com

## Abstract

We consider the adversarial linear contextual bandit problem, where the loss vectors are selected fully adversarially and the per-round action set (i.e. the context) is drawn from a fixed distribution. Existing methods for this problem either require access to a simulator to generate free i.i.d. contexts, achieve a sub-optimal regret no better than $\widetilde{\mathcal{O}}(T^{5/6})$, or are computationally inefficient. We greatly improve these results by achieving a regret of $\widetilde{\mathcal{O}}(\sqrt{T})$ without a simulator, while maintaining computational efficiency when the action set in each round is small. In the special case of sleeping bandits with adversarial loss and stochastic arm availability, our result answers affirmatively the open question by Saha et al. [2020] on whether there exists a polynomial-time algorithm with $\text{poly}(d)\sqrt{T}$ regret. Our approach naturally handles the case where the loss is linear up to an additive misspecification error, and our regret shows near-optimal dependence on the magnitude of the error.

## 1 Introduction

Contextual bandit is a widely used model for sequential decision making. The interaction between the learner and the environment proceeds in rounds: in each round, the environment provides a context; based on it, the learner chooses an action and receive a reward. The goal is to maximize the total reward across multiple rounds. This model has found extensive applications in fields such as medical treatment [Tewari and Murphy, 2017], personalized recommendations [Beygelzimer et al., 2011], and online advertising [Chu et al., 2011].

Algorithms for contextual bandits with provable guarantees have been developed under various assumptions. In the linear regime, the most extensively studied model is the *stochastic linear contextual bandit*, in which the context can be arbitrarily distributed in each round, while the reward is determined by a fixed linear function of the context-action pair. Near-optimal algorithms for this setting have been established in, e.g., [Chu et al., 2011, Abbasi-Yadkori et al., 2011, Li et al., 2019, Foster et al., 2020]. Another model, which is the focus of this paper, is the *adversarial linear contextual bandit*, in which the context is drawn from a fixed distribution, while the reward is determined by a time-varying linear function of the context-action pair. [3] A computationally efficient algorithm for this setting is first proposed by Neu and Olkhovskaya [2020]. However, existing research for this setting still faces challenges in achieving near-optimal regret and sample complexity when the context distribution is unknown.

---

*The authors are listed in alphabetical order.

[†]This work was done when Chen-Yu Wei was at MIT Institute for Data, Systems, and Society.

[3]Apparently, the stochastic and adversarial linear contextual bandits defined here are incomparable, and their names do not fully capture their underlying assumptions. However, these are the terms commonly used in the literature (e.g., [Abbasi-Yadkori et al., 2011, Neu and Olkhovskaya, 2020]).

37th Conference on Neural Information Processing Systems (NeurIPS 2023).

The algorithm by Neu and Olkhovskaya [2020] requires the learner to have *full knowledge* on the context distribution, and access to an *exploratory policy* that induces a feature covariance matrix with a smallest eigenvalue at least $\lambda$. Under these assumptions, their algorithm provides a regret guarantee of $\widetilde{\mathcal{O}}(\sqrt{d\log(|\mathcal{A}|)T/\lambda})^4$, where $d$ is the feature dimension, $|\mathcal{A}|$ is the maximum size of the action set, and $T$ is the number of rounds. These assumptions are relaxed in the work of Luo et al. [2021], who studied a more general linear MDP setting. When specialized to linear contextual bandits, Luo et al. [2021] only requires access to a *simulator* from which the learner can draw free i.i.d. contexts. Their algorithm achieves a $\widetilde{\mathcal{O}}((d\log(|\mathcal{A}|)T^2)^{1/3}))$ regret. The regret is further improved to the near-optimal one $\widetilde{\mathcal{O}}(\sqrt{d\log(|\mathcal{A}|)T})$ by Dai et al. [2023] through refined loss estimator construction.

All results that attain $\widetilde{\mathcal{O}}(T^{2/3})$ or $\widetilde{\mathcal{O}}(\sqrt{T})$ regret bound discussed above rely on access to the simulator. In their algorithms, the number of calls to the simulator significantly exceeds the number of interactions between the environment and the learner, but this is concealed from the regret bound. Therefore, their regret bounds do not accurately reflect the sample complexity of their algorithms. Another set of results for linear MDPs [Luo et al., 2021, Dai et al., 2023, Sherman et al., 2023, Kong et al., 2023] also consider the simulator-free scenario, essentially using interactions with the environment to fulfill the original purpose of the simulator. When applying their techniques to linear contextual bandits, their algorithms only achieve a regret bound of $\widetilde{\mathcal{O}}(T^{5/6})$ at best (see detailed analysis and comparison in Appendix G).

Our result significantly improves the previous ones: without simulators, we develop an algorithm that ensures a regret bound of order $\widetilde{\mathcal{O}}(d^2\sqrt{T})$, and it is computationally efficient as long as the size of the action set is small in each round (similar to all previous work). Unlike previous algorithms which always collect new contexts (through simulators or interactions with the environment) to estimate the feature covariance matrix, we leverage the context samples the learner received in the past to do this. Although natural, establishing a near-tight regret requires highly efficient use of context samples, necessitating a novel way to construct the estimator of feature covariance matrix and a tighter concentration bound for it. Additionally, to address the potentially large magnitude and the bias of the loss estimator, we turn to the use of log-determinant (logdet) barrier in the follow-the-regularized-leader (FTRL) framework. Logdet accommodates larger loss estimators and induces a larger bonus term to cancel the bias of the loss estimator, both of which are crucial for our result.

Our setting subsumes sleeping bandits with stochastic arm availability [Kanade et al., 2009, Saha et al., 2020] and combinatorial semi-bandits with stochastic action sets [Neu and Valko, 2014]. Our result answers affirmatively the main open question left by Saha et al. [2020] on whether there exists a polynomial-time algorithm with $\text{poly}(d)\sqrt{T}$ regret for sleeping bandits with adversarial loss and stochastic availability.

As a side result, we give a computationally inefficient algorithm that achieves an improved $\widetilde{\mathcal{O}}(d\sqrt{T})$ regret without a simulator. While this is a direct extension from the EXP4 algorithm [Auer et al., 2002], such a result has not been established to our knowledge, so we include it for completeness.

## 1.1 Related work

We review the literature of various contextual bandit problems, classifying them based on the nature of the context and the reward function, specifically whether they are stochastic/fixed or adversarial.

**Contextual bandits with i.i.d. contexts and fixed reward functions (S-S)** Significant progress has been made in contextual bandits with i.i.d. contexts and fixed reward functions, under general reward function classes or policy classes [Langford and Zhang, 2007, Dudik et al., 2011, Agarwal et al., 2012, 2014, Simchi-Levi and Xu, 2022, Xu and Zeevi, 2020]. In the work by Dudik et al. [2011], Agarwal et al. [2012, 2014], the algorithms also use previously collected contexts to estimate the inverse probability of selecting actions under the current policy. However, these results only obtain regret bounds that polynomially depend on the number of actions. Furthermore, these results rely on having a fixed reward function, making their techniques not directly applicable to our case.

---

[4]The linear contextual bandit problem formulation in Neu and Olkhovskaya [2020] is different from ours. However, it can be reduced to our setting with dimension $d|\mathcal{A}|$, where $|\mathcal{A}|$ is the maximum number of actions in each round. The $\widetilde{\mathcal{O}}(\sqrt{d\log(|\mathcal{A}|)T/\lambda})$ bound reported here is obtained by adopting their techniques to our setting.

Table 1: Related works in the "S-A" category. CB stands for contextual bandits and SB stands for semi-bandits. The relations among settings are as follows: Sleeping Bandit $\subset$ Contextual SB $\subset$ Linear CB, Linear CB $\subset$ Linear MDP, and Linear CB $\subset$ General CB. The table compares our results with the Pareto frontier of the literature. For algorithms dealing more general settings, we have carefully translated their techniques to Linear CB and reported the resulting bounds. $\Sigma_\pi$ denotes the feature covariance matrix induced by policy $\pi$. $|\mathcal{A}|$ and $|\Pi|$ are sizes of the action set and the policy set.

| Target Setting | Algorithm | Regret | Simulator | Computation | Assumption |
|---|---|---|---|---|---|
| General CB | Syrgkanis et al. [2016b] | $(\log |\Pi|)^{1/3}(|\mathcal{A}|T)^{2/3}$ | ✓ | $\text{poly}(|\mathcal{A}|, \log |\Pi|, T)$ | ERM Oracle |
| Linear MDP | Dai et al. [2023] | $\sqrt{dT \log |\mathcal{A}|}$ | ✓ | $\text{poly}(|\mathcal{A}|, d, T)$ | |
| | Dai et al. [2023] Sherman et al. [2023] | $d(\log |\mathcal{A}|)^{1/6}T^{5/6}$ | | $\text{poly}(|\mathcal{A}|, d, T)$ | |
| | Kong et al. [2023] | $(d^7 T^4)^{1/5} + \text{poly}\left(\frac{1}{\lambda}\right)$ | | $T^d$ | $\exists \pi, \Sigma_\pi \succeq \lambda I$ |
| Linear CB | Algorithm 1 | $d^2\sqrt{T}$ | | $\text{poly}(|\mathcal{A}|, d, T)$ | |
| | Algorithm 2 | $d\sqrt{T}$ | | $T^d$ | |
| Contextual SB | Neu and Valko [2014] | $(dT)^{2/3}$ | | $\text{poly}(d, T)$ | |
| Sleeping Bandit | Saha et al. [2020] | $\sqrt{2^d T}$ | | $\text{poly}(d, T)$ ($|\mathcal{A}| \leq d$) | |

**Contextual bandits with adversarial contexts and fixed reward functions (A-S)** In this category, the most well-known results are in the linear setting [Chu et al., 2011, Abbasi-Yadkori et al., 2011, Zhao et al., 2023]. Besides the linear case, previous work has investigated specific reward function classes [Russo and Van Roy, 2013, Li et al., 2022, Foster et al., 2018]. Recently, Foster and Rakhlin [2020] introduced a general approach to deal with general function classes with a finite number of actions, which has since been improved or extended by Foster and Krishnamurthy [2021], Foster et al. [2021], Zhang [2022]. This category of problems is not directly comparable to the setting studied in this paper, but both capture a certain degree of non-stationarity of the environment.

**Contextual bandits with i.i.d. contexts and adversarial reward functions (S-A)** This is the category which our work falls into. Several oracle efficient algorithms that require simulators have been proposed for general policy classes [Rakhlin and Sridharan, 2016, Syrgkanis et al., 2016b]. The oracle they use (i.e., the empirical risk minimization, or ERM oracle), however, is not generally implementable in an efficient manner. For the linear case, the first computationally efficient algorithm is by Neu and Olkhovskaya [2020], under the assumption that the context distribution is known. This is followed by Olkhovskaya et al. [2023] to obtain refined data-dependent bounds. A series of works [Neu and Olkhovskaya, 2021, Luo et al., 2021, Dai et al., 2023, Sherman et al., 2023] apply similar techniques to linear MDPs, but when specialized to linear contextual bandits, they all assume known context distribution, or access to a simulator, or only achieves a regret no better than $\widetilde{\mathcal{O}}(T^{5/6})$. The work of Kong et al. [2023] also studies linear MDPs; when specialized to contextual bandits, they obtain a regret bound of $\widetilde{\mathcal{O}}(T^{4/5} + \text{poly}(\frac{1}{\lambda}))$ without a simulator but with a computationally inefficient algorithm and an undesired inverse dependence on the smallest eigenvalue of the covariance matrix. Related but simpler settings have also been studied. The sleeping bandit problem with stochastic arm availability and adversarial reward [Kleinberg et al., 2010, Kanade et al., 2009, Saha et al., 2020] is a special case of our problem where the context is always a subset of standard unit vectors. Another special case is the combinatorial semi-bandit problem with stochastic action sets and adversarial reward [Neu and Valko, 2014]. While these are special cases, the regret bounds in these works are all worse than $\widetilde{\mathcal{O}}(\text{poly}(d)\sqrt{T})$. Therefore, our result also improves upon theirs. [5]

**Contextual bandits with adversarial contexts and adversarial reward functions (A-A)** When both contexts and reward functions are adversarial, there are computational [Kanade and Steinke, 2014] and oracle-call [Hazan and Koren, 2016] lower bounds showing that no sublinear regret is achievable unless the computational cost scales polynomially with the size of the policy set. Even for the linear case, Neu and Olkhovskaya [2020] argued that the problem is at least as hard as online learning a one-dimensional threshold function, for which sublinear regret is impossible. For this

---

[5]For combinatorial semi-bandit problems, our algorithm is not as computationally efficient as Neu and Valko [2014], which can handle exponentially large action sets.

challenging category, besides using the inefficient EXP4 algorithm, previous work makes stronger assumptions on the contexts [Syrgkanis et al., 2016a] or resorts to alternative benchmarks such as dynamic regret [Luo et al., 2018, Chen et al., 2019] and approximate regret [Emamjomeh-Zadeh et al., 2021].

**Lifting and exploration bonus for high-probability adversarial linear bandits**  Our technique is related to those obtaining high-probability bounds for linear bandits. Early development in this line of research only achieves computational efficiency when the action set size is small [Bartlett et al., 2008] or only applies to special action sets such as two-norm balls [Abernethy and Rakhlin, 2009]. Recently, near-optimal high-probability bounds for general convex action sets have been obtained by lifting the problem to a higher dimensional one, which allows for a computationally efficient way to impose bonuses [Lee et al., 2020, Zimmert and Lattimore, 2022]. The lifting and the bonus ideas we use are inspired by them, though for different purposes. However, due to the extra difficulty arising in the contextual case, currently we only obtain a computationally efficient algorithm when the action set size is small.

## 1.2  Computational Complexity

Our main algorithm is based on log-determinant barrier optimization similar to Foster et al. [2020], Zimmert and Lattimore [2022]. Computing its action distribution is closely related to computing the D-optimal experimental design [Khachiyan and Todd, 1990]. Per step, this is shown to require $\widetilde{\mathcal{O}}(|\mathcal{A}_t| \operatorname{poly}(d))$ computational and $\widetilde{\mathcal{O}}(\log(|\mathcal{A}_t|) \operatorname{poly}(d))$ memory complexity [Foster et al., 2020, Proposition 1], where $|\mathcal{A}_t|$ is the action set size at round $t$. The computational bottleneck comes from (approximately) maximizing a quadratic function over the action set. It is an open question whether linear optimization oracles or other type of oracles can lead to efficient implementation of our algorithm for continuous action sets.

In the literature, there are few linear context bandit algorithms that provably avoid $|\mathcal{A}|$ computation per round. The LinUCB algorithm [Chu et al., 2011, Abbasi-Yadkori et al., 2011] suffers from the same quadratic function maximization issue, and therefore is computationally comparable to our algorithm. The SquareCB.Lin algorithm by Foster et al. [2020] is based on the same log-determinant barrier optimization. Another recent algorithm by Zhang [2022] only admits an efficient implementation for continuous action sets in the Bayesian setting but not in the frequentist setting (though they provided an efficient heuristic implementation in their experiments). The Thompson sampling algorithm by Agrawal and Goyal [2013], which has efficient implementation, also relies on well-specified Gaussian prior. The only work that we know can avoids $|\mathcal{A}|$ computation in the frequentist setting is Zhu et al. [2022], but their technique is only known to handle the A-S setting.

## 2  Preliminaries

We study the adversarial linear contextual bandit problem where the loss vectors are selected fully adversarially and the per-round action set (i.e. the context) is drawn from a fixed distribution. The learner and the environment interact in the following way. Let $\mathbb{B}_2^d$ be the L2-norm unit ball in $\mathbb{R}^d$.

For $t = 1, \cdots, T$,

1. The environment decides an adversarial loss vector $y_t \in \mathbb{B}_2^d$, and generates a random action set (i.e., context) $\mathcal{A}_t \subset \mathbb{B}_2^d$ from a fixed distribution $D$ independent from anything else.

2. The learner observes $\mathcal{A}_t$, and (randomly) chooses an action $a_t \in \mathcal{A}_t$.

3. The learner receives the loss $\ell_t \in [-1, 1]$ with $\mathbb{E}[\ell_t] = \langle a_t, y_t \rangle$.

A policy $\pi$ is a mapping which, given any action set $\mathcal{A} \subset \mathbb{R}^d$, maps it to an element in the convex hull of $\mathcal{A}$. We use $\pi(\mathcal{A})$ to refer to the element that it maps $\mathcal{A}$ to. The learner's *regret with respect to policy $\pi$* is defined as the expected performance difference between the learner and policy $\pi$:

$$\operatorname{Reg}(\pi) = \mathbb{E}\left[\sum_{t=1}^{T} \langle a_t, y_t \rangle - \sum_{t=1}^{T} \langle \pi(\mathcal{A}_t), y_t \rangle\right]$$

where the expectation is taken over all randomness from the environment ($y_t$ and $\mathcal{A}_t$) and from the learner ($a_t$). The *pseudo-regret* (or just *regret*) is defined as $\mathrm{Reg} = \max_\pi \mathrm{Reg}(\pi)$, where the maximization is taken over all possible policies.

**Notations** For any matrix $A$, we use $\lambda_{\max}(A)$ and $\lambda_{\min}(A)$ to denote the maximum and minimum eigenvalues of $A$, respectively. We use $\mathrm{Tr}(A)$ to denote the trace of matrix $A$. For any action set $\mathcal{A}$, let $\Delta(\mathcal{A})$ be the space of probability measures on $\mathcal{A}$. Let $\mathcal{F}_t = \sigma(\mathcal{A}_s, a_s, \forall s \le t)$ be the $\sigma$-algebra at round $t$. Define $\mathbb{E}_t[\cdot] = \mathbb{E}[\cdot | \mathcal{F}_{t-1}]$. Given a differentiable convex function $F : \mathbb{R}^d \to \mathbb{R} \cup \{\infty\}$, the Bregman divergence with respect to $F$ is defined as $D_F(x, y) = F(x) - F(y) - \langle \nabla F(y), x - y \rangle$. Given a positive semi-definite (PSD) matrix $A$, for any vector $x$, define the norm generated by $A$ as $\|x\|_A = \sqrt{x^\top A x}$. For any context $\mathcal{A} \subset \mathbb{R}^d$ and $p \in \Delta(\mathcal{A})$, define $\mu(p) = \mathbb{E}_{a \sim p}[a]$ and $\mathrm{Cov}(p) = \mathbb{E}_{a \sim p}[(a - \mu(p))(a - \mu(p))^\top]$. For any $a$, define the lifted action $\boldsymbol{a} = (a, 1)^\top$ and the lifted covariance matrix $\widehat{\mathrm{Cov}}(p) = \mathbb{E}_{a \sim p}[\boldsymbol{a}\boldsymbol{a}^\top] = \mathbb{E}_{a \sim p} \begin{bmatrix} aa^\top & a \\ a^\top & 1 \end{bmatrix} = \begin{bmatrix} \mathrm{Cov}(p) + \mu(p)\mu(p)^\top & \mu(p) \\ \mu(p)^\top & 1 \end{bmatrix}$.

We use **bold** matrices to denote matrices in the lifted space (e.g., in Algorithm 1 and Definition 1).

## 3    Follow-the-Regularized-Leader with the Log-Determinant Barrier

In this section, we present our main algorithm, Algorithm 1. This algorithm can be viewed as instantiating an individual Follow-The-Regularized-Leader (FTRL) algorithm on each action set (Line 2), with all FTRLs sharing the same loss vectors. This perspective has been taken by previous works Neu and Olkhovskaya [2020], Olkhovskaya et al. [2023] and simplifies the understanding of the problem. The rationale comes from the following calculation due to Neu and Olkhovskaya [2020]: for any policy $\pi$ that may depend on $\mathcal{F}_{t-1}$,

$$\mathbb{E}_t\left[\langle \pi(\mathcal{A}_t), y_t \rangle\right] = \mathbb{E}_{\mathcal{A}_t}\left[\mathbb{E}_{y_t}\left[\langle \pi(\mathcal{A}_t), y_t \rangle \mid \mathcal{F}_{t-1}\right]\right] = \mathbb{E}_{\mathcal{A}_0}\left[\mathbb{E}_{y_t}\left[\langle \pi(\mathcal{A}_0), y_t \rangle \mid \mathcal{F}_{t-1}\right]\right] = \mathbb{E}_t\left[\langle \pi(\mathcal{A}_0), y_t \rangle\right]$$

where $\mathcal{A}_0$ is a sample drawn from $D$ independent of all interaction history. This allows us to calculate the regret as

$$\mathbb{E}\left[\sum_{t=1}^T \langle \pi_t(\mathcal{A}_t) - \pi(\mathcal{A}_t), y_t \rangle\right] = \mathbb{E}\left[\sum_{t=1}^T \langle \pi_t(\mathcal{A}_0) - \pi(\mathcal{A}_0), y_t \rangle\right] \tag{1}$$

where $\pi_t$ is the policy used by the learner at time $t$. Note that this view does not require the learner to simultaneously "run" an algorithm on every action set since the learner only needs to calculate the policy on $\mathcal{A}$ whenever $\mathcal{A}_t = \mathcal{A}$. In the regret analysis, in view of Eq. (1), it suffices to consider a single fixed action set $\mathcal{A}_0$ drawn from $D$ and bound the regret on it, even though the learner may never execute the policy on it. This $\mathcal{A}_0$ is called a "ghost sample" in Neu and Olkhovskaya [2020].

### 3.1    The lifting idea and the execution of Algorithm 1

Our algorithm is built on the logdet-FTRL algorithm developed by Zimmert and Lattimore [2022] for high-probability adversarial linear bandits, which lifts the original $d$-dimensional problem over the feature space to a $(d + 1) \times (d + 1)$ one over the covariance matrix space, with the regularizer being the negative log-determinant function. In our case, we instantiate an individual logdet-FTRL on each action set. The motivation behind Zimmert and Lattimore [2022] to lift the problem to the space of covariance matrix is that it casts the problem to one in the positive orthant, which allows for an easier way to construct the *bonus* term that is crucial to compensate the variance of the losses, enabling a high-probability bound in their case. In our case, we use the same technique to introduce the bonus term, but the goal is to compensate the *bias* resulting from the estimation error in the covariance matrix (see Section 3.4). This bias only appears in our contextual case but not in the linear bandit problem originally considered in Zimmert and Lattimore [2022].

As argued previously, we can focus on the learning problem over a fixed action set $\mathcal{A}$, and our algorithm operates in the lifted space of covariance matrices $\mathcal{H}^{\mathcal{A}} = \{\widehat{\mathrm{Cov}}(p) : p \in \Delta(\mathcal{A})\} \subset \mathbb{R}^{(d+1) \times (d+1)}$. For this space, we define the lifted loss $\gamma_t = \begin{bmatrix} 0 & \frac{1}{2}y_t \\ \frac{1}{2}y_t^\top & 0 \end{bmatrix} \in \mathbb{R}^{(d+1) \times (d+1)}$ so that $\langle \widehat{\mathrm{Cov}}(p), \gamma_t \rangle = \mathbb{E}_{a \sim p}[a^\top y_t] = \langle \mu(p), y_t \rangle$ and thus the loss value in the lifted space (i.e., $\langle \widehat{\mathrm{Cov}}(p), \gamma_t \rangle$) is the same as that in the original space (i.e., $\langle \mu(p), y_t \rangle$).

---

**Algorithm 1** Logdet-FTRL for linear contextual bandits

---

**Definitions**: $F(\boldsymbol{H}) = -\log\det(\boldsymbol{H}), \eta_t = \frac{1}{64d\sqrt{t}}, \alpha_t = \frac{d}{\sqrt{t}}, \beta_t = \frac{100(d+1)^3\log(3T)}{t-1}$.

1 **for** $t = 1, 2, \ldots$ **do**

2     For all $\mathcal{A}$, define $\boldsymbol{H}_t^{\mathcal{A}} = \underset{\boldsymbol{H}\in\mathcal{H}^{\mathcal{A}}}{\arg\min} \sum_{s=1}^{t-1}\langle\boldsymbol{H}, \hat{\gamma}_s - \alpha_s\hat{\boldsymbol{\Sigma}}_s^{-1}\rangle + \frac{F(\boldsymbol{H})}{\eta_t}$.

3     For all $\mathcal{A}$, define $p_t^{\mathcal{A}} \in \Delta(\mathcal{A})$ such that $\boldsymbol{H}_t^{\mathcal{A}} = \widehat{\mathrm{Cov}}(p_t^{\mathcal{A}})$.

4     Receive $\mathcal{A}_t$ and sample $a_t \sim p_t^{\mathcal{A}_t}$.

5     Observe $\ell_t \in [-1, 1]$ with $\mathbb{E}[\ell_t] = a_t^\top y_t$ and construct $\hat{y}_t = \hat{\Sigma}_t^{-1}(a_t - \hat{x}_t)\ell_t$, where

$$\hat{x}_t = \frac{1}{t-1}\sum_{\tau=1}^{t-1}\mathbb{E}_{a\sim p_t^{\mathcal{A}_\tau}}[a], \ \ \hat{H}_t = \frac{1}{t-1}\sum_{\tau=1}^{t-1}\mathbb{E}_{a\sim p_t^{\mathcal{A}_\tau}}\left[(a-\hat{x}_t)(a-\hat{x}_t)^\top\right], \ \ \hat{\Sigma}_t = \hat{H}_t + \beta_t I.$$

6     Define $\hat{\boldsymbol{H}}_t = \frac{1}{t-1}\sum_{\tau=1}^{t-1}\boldsymbol{H}_t^{\mathcal{A}_\tau}$ and $\hat{\boldsymbol{\Sigma}}_t = \hat{\boldsymbol{H}}_t + \beta_t\boldsymbol{I}$ and $\hat{\gamma}_t = \begin{bmatrix} 0 & \frac{1}{2}\hat{y}_t \\ \frac{1}{2}\hat{y}_t^\top & 0 \end{bmatrix}$.

    (If $t = 1$, define $\hat{\Sigma}_t^{-1}$ and $\hat{\boldsymbol{\Sigma}}_t^{-1}$ as zeros).

---

In each round $t$, the FTRL on $\mathcal{A}$ outputs a lifted covariance matrix $\boldsymbol{H}_t^{\mathcal{A}} \in \mathcal{H}^{\mathcal{A}}$ that corresponds to a probability distribution $p_t^{\mathcal{A}} \in \Delta(\mathcal{A})$ such that $\widehat{\mathrm{Cov}}(p_t^{\mathcal{A}}) = \boldsymbol{H}_t^{\mathcal{A}}$ (Line 2 and Line 3). Upon receiving $\mathcal{A}_t$, the learner samples an action from $p_t^{\mathcal{A}_t}$ and the agent constructs the loss estimator $\hat{y}_t$ (Line 5).

Similarly to the construction of $\gamma_t$, we define the lifted loss estimator $\hat{\gamma}_t = \begin{bmatrix} 0 & \frac{1}{2}\hat{y}_t \\ \frac{1}{2}\hat{y}_t^\top & 0 \end{bmatrix}$ which makes $\langle\widehat{\mathrm{Cov}}(p), \hat{\gamma}_t\rangle = \mathbb{E}_{a\sim p}[a^\top\hat{y}_t] = \langle\mu(p), \hat{y}_t\rangle$. The lifted loss estimator, along with the *bonus* term $-\alpha_t\hat{\boldsymbol{\Sigma}}_t^{-1}$, is then fed to the FTRL on all $\mathcal{A}$'s. The purpose of the bonus term will be clear in Section 3.4.

In the rest of this section, we use the following notation in addition to those defined in Algorithm 1.

**Definition 1.** *Define* $x_t^{\mathcal{A}} = \mathbb{E}_{a\sim p_t^{\mathcal{A}}}[a], \ x_t = \mathbb{E}_{\mathcal{A}\sim D}[x_t^{\mathcal{A}}], \ H_t^{\mathcal{A}} = \mathbb{E}_{a\sim p_t^{\mathcal{A}}}[(a-\hat{x}_t)(a-\hat{x}_t)^\top], \ H_t = \mathbb{E}_{\mathcal{A}\sim D}[H_t^{\mathcal{A}}], \ \boldsymbol{H}_t = \mathbb{E}_{\mathcal{A}\sim D}[\boldsymbol{H}_t^{\mathcal{A}}].$ *Let* $p_\star^{\mathcal{A}} \in \Delta(\mathcal{A})$ *be the action distribution used by the benchmark policy on* $\mathcal{A}$, *and define* $u^{\mathcal{A}} = \mathbb{E}_{a\sim p_\star^{\mathcal{A}}}[a], \ u = \mathbb{E}_{\mathcal{A}\sim D}[u^{\mathcal{A}}], \ \boldsymbol{U}^{\mathcal{A}} = \mathbb{E}_{a\sim p_\star^{\mathcal{A}}}[\boldsymbol{a}\boldsymbol{a}^\top], \ \boldsymbol{U} = \mathbb{E}_{\mathcal{A}\sim D}[\boldsymbol{U}^{\mathcal{A}}].$ *Notice that the* $x_t^{\mathcal{A}}$ *and* $u^{\mathcal{A}}$ *defined here is equivalent to the* $\pi_t(\mathcal{A})$ *and* $\pi(\mathcal{A})$ *in Eq.* (1), *respectively.*

### 3.2 The construction of loss estimators and feature covariance matrix estimators

Our goal is to make $\hat{y}_t$ in Line 5 an estimator of $y_t$ with controllable bias and variance. If the context distribution is known (as in Neu and Olkhovskaya [2020]), then a standard unbiased estimator of $y_t$ is

$$\hat{y}_t = \Sigma_t^{-1}a_t\ell_t, \qquad \text{where} \quad \Sigma_t = \mathbb{E}_{\mathcal{A}\sim D}\mathbb{E}_{a\sim p_t^{\mathcal{A}}}\left[aa^\top\right]. \tag{2}$$

To see its unbiasedness, notice that $\mathbb{E}[a_t\ell_t] = \mathbb{E}_{\mathcal{A}\sim D}\mathbb{E}_{a\sim p_t^{\mathcal{A}}}[aa^\top y_t]$ and thus $\mathbb{E}[\hat{y}_t] = y_t$. This $\hat{y}_t$, however, can have a variance that is inversely related to the smallest eigenvalue of the covariance matrix $\hat{\Sigma}_t$, which can be unbounded in the worst case. This is the main reason why Neu and Olkhovskaya [2020] does not achieve the optimal bound, and requires the bias-variance-tradeoff techniques in Dai et al. [2023] to close the gap. When the context distribution is unknown but the learner has access to a simulator [Luo et al., 2021, Dai et al., 2023, Sherman et al., 2023, Kong et al., 2023], the learner can draw free contexts to estimate the covariance matrix $\hat{\Sigma}_t$ up to a very high accuracy without interacting with the environment, making the problem close to the case of known context distribution.

Challenges arise when the learner has no knowledge about the context distribution and there is no simulator. In this case, there are two natural ways to estimate the covariance matrix under the current policy. One is to draw new samples from the environment, treating the environment like a simulator. This approach is essentially taken by all previous work studying linear models in the "S-A" category. However, this is very expensive, and it causes the simulator-equipped bound $\sqrt{T}$ in Dai et al. [2023]

to deteriorate to the simulator-free bound $T^{5/6}$ at best (see Appendix G for details). The other is to use the contexts received in time 1 to $t$ to estimate the covariance matrix under the policy at time $t$. This demands a very high efficiency in reusing the contexts samples, and existing ways of constructing the covariance matrix and the accompanied analysis by Dai et al. [2023], Sherman et al. [2023] are insufficient to achieve the near-optimal bound even with context reuse. This necessitates our tighter construction of the covariance matrix estimator and tighter concentration bounds for it.

Our construction of the loss estimator (Line 5) is

$$\hat{y}_t = \hat{\Sigma}_t^{-1}(a_t - \hat{x}_t)\ell_t \qquad \text{where} \quad \hat{\Sigma}_t = \mathbb{E}_{\mathcal{A}\sim\hat{D}_t}\mathbb{E}_{a\sim p_t^{\mathcal{A}}}\left[(a - \hat{x}_t)(a - \hat{x}_t)^\top\right] + \beta_t I \qquad (3)$$

where $\hat{D}_t = \text{Uniform}\{\mathcal{A}_1, \mathcal{A}_2, \ldots, \mathcal{A}_{t-1}\}$, $\hat{x}_t = \mathbb{E}_{\mathcal{A}\sim\hat{D}_t}, \mathbb{E}_{a\sim p_t^{\mathcal{A}}}[a]$, and $\beta_t = \widetilde{\mathcal{O}}(d^3/t)$. Comparing Eq. (3) with Eq. (2), we see that besides using the empirical context distribution $\hat{D}_t$ in place of the ground truth $D$ and adding a small term $\beta_t I$ to control the smallest eigenvalue of the covariance matrix, we also centralize the features by $\hat{x}_t$, an estimation of the mean features under the current policy. The centralization is important in making the bias $y_t - \hat{y}_t$ appear in a nice form that can be compensated by a bonus term. The estimator might seem problematic on first sight, because $p_t^{\mathcal{A}}$ is strongly dependent on $\hat{D}_t$, which rules out canonical concentration bounds. We circumvent this issue by leveraging the special structure of $p_t$ in Algorithm 1, which allows for a union bound over a sufficient covering of all potential policies (Appendix C.3). The analysis on the bias of this loss estimator is also non-standard, which is the key to achieve the near-optimal bound . In the next two subsections, we explain how to bound the *bias* of this loss estimator (Section 3.3), and how the *bonus* term can be used to compensate the bias (Section 3.4).

### 3.3 The bias of the loss estimator

Since the true loss vector is $y_t$ and we use the loss estimator $\hat{y}_t$ in the update, there is a bias term emerging in the regret bound at time $t$:

$$\mathbb{E}_t\left[\langle x_t^{\mathcal{A}_0} - u^{\mathcal{A}_0}, y_t - \hat{y}_t\rangle\right] = \mathbb{E}_t\left[\langle x_t - u, y_t - \hat{y}_t\rangle\right] = \mathbb{E}_t\left[(x_t - u)^\top\left(I - \hat{\Sigma}_t^{-1}(a_t - \hat{x}_t)a_t^\top\right)y_t\right]$$

where definitions of $x_t^{\mathcal{A}}, u^{\mathcal{A}}, x_t, u$ can be found in Definition 1, and we use the definition of $\hat{y}_t$ in Eq. (3) in the last equality. Now taking expectation over $\mathcal{A}_t$ and $a_t$ conditioned on $\mathcal{F}_{t-1}$, we can further bound the expectation in the last expression by

$$(x_t - u)^\top\left(I - \hat{\Sigma}_t^{-1}H_t\right)y_t - (x_t - u)^\top\hat{\Sigma}_t^{-1}(x_t - \hat{x}_t)\hat{x}_t^\top y_t$$

$$\leq \|x_t - u\|_{\hat{\Sigma}_t^{-1}}\|(\hat{\Sigma}_t - H_t)y_t\|_{\hat{\Sigma}_t^{-1}} + \|x_t - u\|_{\hat{\Sigma}_t^{-1}}\|x_t - \hat{x}_t\|_{\hat{\Sigma}_t^{-1}} \qquad (4)$$

(see Definition 1 for the definition of $H_t$). The two terms $\|(\hat{\Sigma}_t - H_t)y_t\|_{\hat{\Sigma}_t^{-1}}$ and $\|x_t - \hat{x}_t\|_{\hat{\Sigma}_t^{-1}}$ in Eq. (4) are related to the error between the empirical context distribution $\hat{D}_t = \text{Uniform}\{\mathcal{A}_1, \ldots, \mathcal{A}_{t-1}\}$ and the true distribution $D$. We handle them through novel analysis and bound both of them by $\widetilde{\mathcal{O}}\left(\sqrt{d^3/t}\right)$. See Lemma 13, Lemma 14, Lemma 18, and Lemma 19 for details. The techniques we use in these lemmas surpass those in Dai et al. [2023], Sherman et al. [2023]. As a comparison, a similar term as $\|(\hat{\Sigma}_t - H_t)y_t\|_{\hat{\Sigma}_t^{-1}}$ is also presented in Eq. (16) of Dai et al. [2023] and Lemma B.5 of Sherman et al. [2023] when bounding the bias. While their analysis uses off-the-shelf matrix concentration inequalities, our analysis expands this expression by its definition, and applies concentration inequalities for *scalars* on individual entries. Overall, our analysis is more tailored for this specific expression. Previous works ensure that this term can be bounded by $\mathcal{O}(\sqrt{\beta})$ after collecting $\mathcal{O}(\beta^{-2})$ new samples (Lemma 5.1 of Dai et al. [2023] and Lemma B.1 of Sherman et al. [2023]), we are able to bound it by $\mathcal{O}(1/\sqrt{t})$ only using $t$ samples that the learner received up to time $t$. This essentially improves their $\mathcal{O}(\beta^{-2})$ sample complexity bound to $\mathcal{O}(\beta^{-1})$. See Appendix G for detailed comparison with Dai et al. [2023] and Sherman et al. [2023].

Now we have bounded the regret due to bias of $\hat{y}_t$ by the order of $\sqrt{d^3/t}\|x_t - u\|_{\hat{\Sigma}_t^{-1}}$. The next problem is how to mitigate this term. This is also a problem in previous work [Luo et al., 2021, Dai et al., 2023, Sherman et al., 2023], and it has become clear that this can be handled by incorporating *bonus* in the algorithm.

## 3.4 The bonus term

To handle a bias term in the form of $\|x_t - u\|_{\hat{\Sigma}_t^{-1}}$, we resort to the idea of *bonus*. To illustrate this, suppose that instead of feeding $\hat{y}_t$ to the FTRLs, we feed $\hat{y}_t - b_t$ for some $b_t$. Then this would give us a regret bound of the following form:

$$\text{Reg} = \mathbb{E}\left[\sum_{t=1}^{T}\langle x_t - u, \hat{y}_t - b_t\rangle\right] + \mathbb{E}\left[\sum_{t=1}^{T}\langle x_t - u, y_t - \hat{y}_t\rangle\right] + \mathbb{E}\left[\sum_{t=1}^{T}\langle x_t - u, b_t\rangle\right]$$

$$\lesssim \widetilde{\mathcal{O}}(d^2\sqrt{T}) + \mathbb{E}\left[\sum_{t=1}^{T}\sqrt{\frac{d^3}{t}}\|x_t - u\|_{\hat{\Sigma}_t^{-1}}\right] + \mathbb{E}\left[\sum_{t=1}^{T}\langle x_t - u, b_t\rangle\right] \qquad (5)$$

where we assume that FTRL can give us $\widetilde{\mathcal{O}}(d^2\sqrt{T})$ bound for the loss sequence $\hat{y}_t - b_t$. Our hope here is to design a $b_t$ such that $\langle x_t - u, b_t\rangle$ provides a negative term that can be used to cancel the bias term $\sqrt{d^3/t}\|x_t - u\|_{\hat{\Sigma}_t^{-1}}$ in the following manner:

$$\text{bias} + \text{bonus} = \sum_{t=1}^{T}\left(\sqrt{\frac{d^3}{t}}\|x_t - u\|_{\hat{\Sigma}_t^{-1}} + \langle x_t - u, b_t\rangle\right) \lesssim \widetilde{\mathcal{O}}(d^2\sqrt{T}). \qquad (6)$$

which gives us a $\widetilde{\mathcal{O}}(d^2\sqrt{T})$ overall regret by Eq. (5). This approach relies on two conditions to be satisfied. First, we have to find a $b_t$ that makes Eq. (6) hold. Second, we have to ensure that the FTRL algorithm achieves a $\widetilde{\mathcal{O}}(d^2\sqrt{T})$ bound under the loss sequence $\hat{y}_t - b_t$.

To meet the first condition, we take inspiration from Zimmert and Lattimore [2022] and lift the problem to the space of covariance matrix in $\mathbb{R}^{(d+1)\times(d+1)}$. Considering the bonus term $\alpha_t\hat{\Sigma}_t^{-1}$ in the lifted space, we have

$$\langle \boldsymbol{H}_t - \boldsymbol{U}, \alpha_t\hat{\boldsymbol{\Sigma}}_t^{-1}\rangle = \alpha_t \operatorname{Tr}(\boldsymbol{H}_t\hat{\boldsymbol{\Sigma}}_t^{-1}) - \alpha_t\operatorname{Tr}(\boldsymbol{U}\hat{\boldsymbol{\Sigma}}_t^{-1}) \qquad (7)$$

Using Lemma 17 and Corollary 22, we can upper bound Eq. (7) by $\mathcal{O}\left(d\alpha_t\right) - \frac{\alpha_t}{4}\|u - \hat{x}_t\|_{\hat{\Sigma}_t^{-1}}^2$. This gives

$$\text{bias} + \text{bonus} \le \sum_{t=1}^{T}\left(\sqrt{\frac{d^3}{t}}\|x_t - u\|_{\hat{\Sigma}_t^{-1}} + d\alpha_t - \frac{\alpha_t}{4}\|\hat{x}_t - u\|_{\hat{\Sigma}_t^{-1}}^2\right)$$

$$\le \widetilde{\mathcal{O}}(d^2\sqrt{T}) + \sum_{t=1}^{T}\sqrt{\frac{d^3}{t}}\|x_t - \hat{x}_t\|_{\hat{\Sigma}_t^{-1}} + \sum_{t=1}^{T}\left(\sqrt{\frac{d^3}{t}}\|\hat{x}_t - u\|_{\hat{\Sigma}_t^{-1}} - \frac{\alpha_t}{4}\|\hat{x}_t - u\|_{\hat{\Sigma}_t^{-1}}^2\right).$$

Using Lemma 18 to bound the second term above by $\widetilde{\mathcal{O}}(\sum_t d^3/t) = \widetilde{\mathcal{O}}(d^3)$, and AM-GM to bound the third term by $\widetilde{\mathcal{O}}(\sum_t d^3/(t\alpha_t)) = \widetilde{\mathcal{O}}(d^2\sqrt{T})$, we get Eq. (6), through the help of lifting.

To meet the second condition, we have to analyze the regret of FTRL under the loss $\hat{y}_t - b_t$. The key is to show that the bonus $\alpha_t\hat{\Sigma}_t^{-1}$ introduces small *stability term* overhead. Thanks to the use of the logdet regularizer and its self-concordance property, the extra stability term introduced by the bonus can indeed be controlled by the order $\sqrt{T}$. The key analysis is in Lemma 27.

Previous works rely on exponential weights [Luo et al., 2021, Dai et al., 2023, Sherman et al., 2023] rather than logdet-FTRL, which comes with the following drawbacks. 1) In Luo et al. [2021], Sherman et al. [2023] where exponential weights is combined with standard loss estimators, the bonus introduces large stability term overhead. Therefore, their bound can only be $T^{2/3}$ at best even with simulators. 2) In Dai et al. [2023] where exponential weights is combined with magnitude-reduced loss estimators, the loss estimator for action $a$ can no longer be represented as a simple linear function $a^\top\hat{y}_t$. Instead, it becomes a complex non-linear function. This restricts the algorithm's potential to leverage linear optimization oracle over the action set and achieve computational efficiency.

## 3.5 Overall regret analysis

With all the algorithmic elements discussed above, now we give a formal statement for our regret guarantee and perform a complete regret analysis. Our main theorem is the following.

**Theorem 2.** *Algorithm 1 ensures* $\text{Reg} \leq \mathcal{O}(d^2\sqrt{T}\log T)$.

*Proof sketch.* Let $\mathcal{A}_0$ be drawn from $D$ independently from all the interaction history between the learner and the environment. Recalling the definitions in Definition 1, we have

$$
\text{Reg} = \mathbb{E}\left[\sum_{t=1}^{T}\langle a_t - u^{\mathcal{A}_t}, y_t\rangle\right] = \mathbb{E}\left[\sum_{t=1}^{T}\langle \boldsymbol{H}_t^{\mathcal{A}_t} - \boldsymbol{U}^{\mathcal{A}_t}, \gamma_t\rangle\right] = \mathbb{E}\left[\sum_{t=1}^{T}\langle \boldsymbol{H}_t^{\mathcal{A}_0} - \boldsymbol{U}^{\mathcal{A}_0}, \gamma_t\rangle\right]
$$

$$
\leq \underbrace{\mathbb{E}\left[\sum_{t=1}^{T}\langle \boldsymbol{H}_t^{\mathcal{A}_0} - \boldsymbol{U}^{\mathcal{A}_0}, \gamma_t - \hat{\gamma}_t\rangle\right]}_{\textbf{Bias}} + \underbrace{\mathbb{E}\left[\sum_{t=1}^{T}\langle \boldsymbol{H}_t^{\mathcal{A}_0} - \boldsymbol{U}^{\mathcal{A}_0}, \alpha_t\hat{\boldsymbol{\Sigma}}_t^{-1}\rangle\right]}_{\textbf{Bonus}} + \underbrace{\mathbb{E}\left[\sum_{t=1}^{T}\langle \boldsymbol{H}_t^{\mathcal{A}_0} - \boldsymbol{U}^{\mathcal{A}_0}, \hat{\gamma}_t - \alpha_t\hat{\boldsymbol{\Sigma}}_t^{-1}\rangle\right]}_{\textbf{FTRL-Reg}}
$$

Each term can be bounded as follows:

- **Bias** $\leq \mathcal{O}(d^2\sqrt{T}\log T) + \frac{1}{4}\sum_{t=1}^{T}\alpha_t\|u - x_t\|_{\hat{\Sigma}_t^{-1}}^2$ (discussed in Section 3.3).
- **Bonus** $\leq \mathcal{O}(d^2\sqrt{T}\log T) - \frac{1}{4}\sum_{t=1}^{T}\alpha_t\|u - x_t\|_{\hat{\Sigma}_t^{-1}}^2$ (discussed in Section 3.4).
- **FTRL-Reg** $\leq \mathcal{O}(d^2\sqrt{T}\log T)$.

Combining all terms gives the desired bound. The complete proof is provided in Appendix D. □

### 3.6 Handling Misspecification

In this subsection, we show how our approach naturally handles the case when the expectation of the loss cannot be exactly realized by a linear function but with a misspecification error. In this case, we assume that the expectation of the loss is given by $\mathbb{E}[\ell_t|a_t = a] = f_t(a)$ for some $f_t : \mathbb{R}^d \to [-1, 1]$, and the realized loss $\ell_t$ still lies in $[-1, 1]$. We define the following notion of misspecification (slightly more refined than that in Neu and Olkhovskaya [2020]):

**Assumption 1** (misspecification). $\sqrt{\frac{1}{T}\sum_{t=1}^{T}\inf_{y\in\mathbb{B}_2^d}\sup_{\mathcal{A}\in\text{supp}(D)}\sup_{a\in\mathcal{A}}(f_t(a) - \langle a, y\rangle)^2} \leq \varepsilon$.

Based on previous discussions, the design idea of Algorithm 1 is to 1) identify the bias of the loss estimator, and 2) add necessary bonus to compensate the bias. When there is misspecification, this design idea still applies. The difference is that now the loss estimator $\hat{y}_t$ potentially has more bias due to misspecification. Therefore, the bias becomes larger by an amount related to $\varepsilon$. Consequently, we need to enlarge bonus (raising $\alpha_t$) to compensate it. Due to the larger bonus, we further need to tune down the learning rate $\eta_t$ to make the algorithm stable. Overall, to handle misspecification, when $\varepsilon$ is known, it boils down to using the same algorithm (Algorithm 1) with adjusted $\alpha_t$ and $\eta_t$. The case of unknown $\varepsilon$ can be handled by the standard meta-learning technique *Corral* [Agarwal et al., 2017, Foster et al., 2020, Luo et al., 2022]. We defer all details to Appendix E and only state the final bound here.

**Theorem 3.** *Under misspecification, there is an algorithm ensuring* $\text{Reg} \leq \widetilde{\mathcal{O}}(d^2\sqrt{T} + \sqrt{d}\varepsilon T)$, *without knowing* $\varepsilon$ *in advance.*

## 4 Linear EXP4

To tighten the $d$-dependence in the regret bound, we can use the computationally inefficient algorithm EXP4 [Auer et al., 2002]. The original regret bound for EXP4 has a polynomial dependence on the number of actions, but here we take the advantage of the linear structure to show a bound that only depends on the feature dimension $d$. The algorithm is presented in Algorithm 2.

**Algorithm 2** Linear EXP4

**input**: $\Pi, \eta, \gamma$.
**for** $t = 1, 2, \ldots$ **do**

    Receive $\mathcal{A}_t \subset \mathbb{R}^d$.

    Construct $\nu_t \in \Delta(\mathcal{A}_t)$ such that $\max_{a \in \mathcal{A}_t} \|a\|^2_{G_t^{-1}} \leq d$, where $G_t = \mathbb{E}_{a \sim \nu_t}[aa^\top]$. Set

$$P_{t,\pi} = \frac{\exp\left(-\eta \sum_{s=1}^{t-1} \hat{\ell}_{s,\pi}\right)}{\sum_{\pi' \in \Pi} \exp\left(-\eta \sum_{s=1}^{t-1} \hat{\ell}_{s,\pi'}\right)}$$

    and define $p_{t,a} = \sum_{\pi \in \Pi} P_{t,\pi} \mathbb{I}\{\pi(\mathcal{A}_t) = a\}$.

    Sample $a_t \sim \tilde{p}_t = (1 - \gamma)p_t + \gamma\nu_t$ and receive $\ell_t \in [-1, 1]$ with $\mathbb{E}[\ell_t] = \langle a_t, y_t \rangle$.

    Construct $\forall \pi \in \Pi$: $\hat{\ell}_{t,\pi} = \langle \pi(\mathcal{A}_t), \tilde{H}_t^{-1} a_t \ell_t \rangle$, where $\tilde{H}_t = \mathbb{E}_{a \sim \tilde{p}_t}[aa^\top]$.

To run Algorithm 2, we restrict ourselves to a finite policy class. The policy class we use in the algorithm is the set of linear policies defined as

$$\Pi = \left\{ \pi_\theta : \ \theta \in \Theta, \ \ \pi_\theta(\mathcal{A}) = \underset{a \in \mathcal{A}}{\operatorname{argmin}} \ a^\top \theta \right\} \tag{8}$$

where $\Theta$ is an 1-net of $[-T, T]^d$. The next theorem shows that this suffices to give us near-optimal bounds for our problem. The proof is given in Appendix F.

**Theorem 4.** *With* $\gamma = 2d\sqrt{(\log T)/T}$ *and* $\eta = \sqrt{(\log T)/T}$, *Algorithm 2 with the policy class defined in Eq. (8) guarantees* $\operatorname{Reg} = \mathcal{O}\left(d\sqrt{T \log T}\right)$.

Note that this result technically also holds in the "A-A" category with respect to the policy class defined in Eq. (8). However, this policy class is *not* necessarily a sufficient cover of all policies of interest when the contexts and losses are adversarial.

## 5 Conclusions

We derived the first algorithm that obtains $\sqrt{T}$ regret in contextual linear bandits with stochastic action sets in the absence of a simulator or prior knowledge on the distribution. As a side result, we obtained the first computationally efficient $\operatorname{poly}(d)\sqrt{T}$ algorithm for adversarial sleeping bandits with general stochastic arm availabilities. We believe the techniques in this paper will be useful for improving results for simulator-free linear MDPs as well.

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

# Appendices

# A Summary of Notation

We summarize the notations that have been defined in Algorithm 1 and Definition 1.

$$\beta_t = \Theta\left(\frac{(d+1)^3 \log(T/\delta)}{t-1}\right)$$

$$\hat{x}_t = \frac{1}{t-1}\sum_{\tau=1}^{t-1}\mathbb{E}_{a\sim p_t^{\mathcal{A}_\tau}}[a]$$

$$\hat{H}_t = \frac{1}{t-1}\sum_{\tau=1}^{t-1}\mathbb{E}_{a\sim p_t^{\mathcal{A}_\tau}}\left[(a-\hat{x}_t)(a-\hat{x}_t)^\top\right]$$

$$\hat{\boldsymbol{H}}_t = \frac{1}{t-1}\sum_{\tau=1}^{t-1}\mathbb{E}_{a\sim p_t^{\mathcal{A}_\tau}}\begin{bmatrix}aa^\top & a\\a^\top & 1\end{bmatrix} = \begin{bmatrix}\hat{H}_t+\hat{x}_t\hat{x}_t^\top & \hat{x}_t\\\hat{x}_t^\top & 1\end{bmatrix}$$

$$\hat{\Sigma}_t = \hat{H}_t + \beta_t I$$

$$\hat{\boldsymbol{\Sigma}}_t = \hat{\boldsymbol{H}}_t + \beta_t \boldsymbol{I} = \begin{bmatrix}\hat{\Sigma}_t+\hat{x}_t\hat{x}_t^\top & \hat{x}_t\\\hat{x}_t^\top & 1+\beta_t\end{bmatrix}$$

$$x_t = \mathbb{E}_{\mathcal{A}\sim\mathcal{D}}\mathbb{E}_{a\sim p_t^{\mathcal{A}}}[a]$$

$$H_t = \mathbb{E}_{\mathcal{A}\sim\mathcal{D}}\mathbb{E}_{a\sim p_t^{\mathcal{A}}}\left[(a-\hat{x}_t)(a-\hat{x}_t)^\top\right]$$

$$\boldsymbol{H}_t = \mathbb{E}_{\mathcal{A}\sim\mathcal{D}}\mathbb{E}_{a\sim p_t^{\mathcal{A}}}\begin{bmatrix}aa^\top & a\\a^\top & 1\end{bmatrix}$$

# B Auxiliary Lemmas

**Lemma 5** (FTRL regret bound, Lemma 18 of Dann et al. [2023a]). *Let $\Omega \subset \mathbb{R}^d$ be a convex set, $g_1,\ldots,g_T \in \mathbb{R}^d$, and $\eta_1,\ldots,\eta_T > 0$. Then the FTRL update*

$$w_t = \operatorname*{argmin}_{w\in\Omega}\left\{\left\langle w, \sum_{\tau=1}^{t-1}g_\tau\right\rangle + \frac{1}{\eta_t}\psi(w)\right\}$$

*ensures for any $u \in \Omega$ and $\eta_0 > 0$,*

$$\sum_{t=1}^T \langle w_t - u, g_t\rangle$$

$$\leq \underbrace{\frac{\psi(u)-\min_{w\in\Omega}\psi(w)}{\eta_0} + \sum_{t=1}^T(\psi(u)-\psi(w_t))\left(\frac{1}{\eta_t}-\frac{1}{\eta_{t-1}}\right)}_{\textbf{Penalty}} + \underbrace{\sum_{t=1}^T\left(\max_{w\in\Omega}\langle w_t-w, g_t\rangle - \frac{D_\psi(w,w_t)}{\eta_t}\right)}_{\textbf{Stability}}.$$

*When $\eta_0, \eta_1, \ldots, \eta_T$ is non-increasing, the penalty term can further be upper bounded by*

$$\textbf{Penalty} \leq \frac{\psi(u)-\min_{w\in\Omega}\psi(w)}{\eta_T}.$$

**Lemma 6** (Bernstein's inequality). *Let $X_1,\cdots,X_n$ be iid random variables; let $\mathbb{E}[X]$ be the expectation and $\mathrm{Var}(X)$ be the variance of these random variables. If for any $i$, $|X_i - \mathbb{E}[X_i]| \leq R$, then with probability of at least $1 - \delta$,*

$$\left|\frac{1}{n}\sum_{i=1}^n X_i - \mathbb{E}[X]\right| \leq \sqrt{\frac{4\,\mathrm{Var}(X)\log\frac{2}{\delta}}{n}} + \frac{4R\log\frac{2}{\delta}}{3n}.$$

**Lemma 7** (Hoeffding's inequality). *Let $X_1, \cdots, X_n$ be iid random variables; let $a \leq X_i \leq b$ and let $\mathbb{E}[X]$ be the expectation. Then with probability of at least $1 - \delta$,*

$$\left| \frac{1}{n} \sum_{i=1}^{n} X_i - \mathbb{E}[X] \right| \leq (b - a) \sqrt{\frac{1}{2n} \log(\frac{2}{\delta})}$$

Given $F(X) = -\log \det(X)$, $D^2 F(X) = X^{-1} \otimes X^{-1}$ where $\otimes$ is the Kronecker product. For any matrix $A = \begin{bmatrix} a_1 & a_2 & \cdots & a_n \end{bmatrix}$, let $\text{vec}(A) = \begin{bmatrix} a_1 \\ \vdots \\ a_n \end{bmatrix}$ which vectorizes matrix $A$ to a column vector by stacking the columns $A$. The second order directional derivative for $F$ is $D^2 F(X)[A, A] = \text{vec}(A)^T \left( X^{-1} \otimes X^{-1} \right) \text{vec}(A) = \text{Tr}(A^\top X^{-1} A X^{-1})$. We define $\|A\|_{\nabla^2 F(X)} = \sqrt{\text{Tr}(A^\top X^{-1} A X^{-1})}$ and $\|A\|_{\nabla^{-2} F(X)} = \sqrt{\text{Tr}(A^\top X A X)}$. It is a pseudo-norm and more discussion can be found in Appendix D of Zimmert et al. [2022]. In the following analysis, we will only use one property of this pseudo-norm shown below which is similar to the Holder inequality.

**Lemma 8.** *For any two symmetric matrices $A, B$ and positive definite matrix $X$,*

$$\langle A, B \rangle \leq \|A\|_{\nabla^2 F(X)} \|B\|_{\nabla^{-2} F(X)}$$

*Proof.* Since $(X \otimes X)^{-1} = X^{-1} \otimes X^{-1}$, from Holder inequality, we have

$$\langle A, B \rangle = \langle \text{vec}(A), \text{vec}(B) \rangle \leq \|\text{vec}(A)\|_{X^{-1} \otimes X^{-1}} \|\text{vec}(B)\|_{(X^{-1} \otimes X^{-1})^{-1}} = \|A\|_{\nabla^2 F(X)} \|B\|_{\nabla^{-2} F(X)}$$

$\square$

## C   Concentration Inequalities

The goal of this section is to show Lemma 18 and Lemma 19, which are key to bound the bias term. We first introduce a useful lemma from Dai et al. [2023], which will be used later to prove our concentration bounds.

### C.1   General Concentration Inequalities

**Lemma 9** (Lemma A.4 in Dai et al. [2023]). *Let $H_1, H_2, \ldots, H_n$ be i.i.d. PSD matrices such that $\mathbb{E}[H_i] = H$, $H_i \preceq I$ almost surely and $H \succeq \frac{1}{dn} \log \frac{d}{\delta} I$. Then with probability $1 - \delta$,*

$$\frac{1}{n} \sum_{i=1}^{n} H_i - H \succeq -\sqrt{\frac{d}{n} \log \frac{d}{\delta}} H^{1/2}$$

**Corollary 10.** *Let $H_1, H_2, \ldots, H_n$ be i.i.d. PSD matrices such that $\mathbb{E}[H_i] = H$ and $H_i \preceq cI$ almost surely for some positive constant c. Let $\hat{H} = \frac{1}{n} \sum_{i=1}^{n} H_i$, then with probability $1 - \delta$,*

$$\hat{H} + \frac{3c}{2} \cdot \frac{d}{n} \log \left( \frac{d}{\delta} \right) I \succeq \frac{1}{2} H \tag{9}$$

*Proof.* A simple corollary of Lemma 9 under the condition of Lemma 9 is that

$$\frac{1}{n} \sum_{i=1}^{n} H_i - H \succeq -\sqrt{\frac{d}{n} \log \frac{d}{\delta}} H^{1/2} \succeq -\frac{1}{2} H - \frac{d}{2n} \log \left( \frac{d}{\delta} \right) I$$

$$\Rightarrow \frac{1}{n} \sum_{i=1}^{n} H_i + \frac{d}{2n} \log \left( \frac{d}{\delta} \right) I \succeq \frac{1}{2} H, \tag{10}$$

where we use that $H^{\frac{1}{2}} \preceq \frac{k}{2} H + \frac{1}{2k}$ for any $k > 0$.

Now consider the condition of this corollary. We first consider the case where $\frac{d}{n}\log(\frac{d}{\delta}) \leq 1$. In this case, we apply Eq. (10) with $H'_i = \frac{1}{2c}H_i + \frac{d}{2n}\log(\frac{d}{\delta})I$, which satisfies the condition for Eq. (10) to hold. This gives

$$\frac{1}{n}\sum_{i=1}^{n}\left(\frac{1}{2c}H_i + \frac{d}{2n}\log\left(\frac{d}{\delta}\right)I\right) + \frac{d}{2n}\log\left(\frac{d}{\delta}\right)I \succeq \frac{1}{2}\left(\frac{1}{2c}H + \frac{d}{2n}\log\left(\frac{d}{\delta}\right)I\right)$$

$$\Rightarrow \hat{H} + \frac{3c}{2}\cdot\frac{d}{n}\log\left(\frac{d}{\delta}\right)I \succeq \frac{1}{2}H$$

with probability at least $1 - \delta$. When $\frac{d}{n}\log(\frac{d}{\delta}) > 1$. Eq. (9) is trivial because $\frac{1}{2}H \preceq \frac{c}{2}I \preceq \frac{c}{2}\cdot\frac{d}{n}\log(\frac{d}{\delta})I$.

$\square$

## C.2 Concentration Inequalities under a Fixed Policy $p$

In this subsection, we establish concentration bounds for a *fixed* policy $p$ (with $p^{\mathcal{A}} \in \Delta(\mathcal{A})$ denoting the action distribution it uses over $\mathcal{A}$) over i.i.d. contexts. The results in this subsection are preparation for Appendix C.3 where we take union bounds over policies.

The setting and notation to be used in this subsection are defined in Definition 11.

**Definition 11.** *Let $\{\mathcal{A}_1, \ldots, \mathcal{A}_n\}$ be i.i.d. context samples drawn from $D$. Let $\hat{D}$ be the uniform distribution over $\{\mathcal{A}_1, \ldots, \mathcal{A}_n\}$.*

*Over this set of context samples, define for any policy $p$,*

$$x(p) = \mathbb{E}_{\mathcal{A}\sim D}\mathbb{E}_{a\sim p^{\mathcal{A}}}[a],$$
$$\hat{x}(p) = \mathbb{E}_{\mathcal{A}\sim\hat{D}}\mathbb{E}_{a\sim p^{\mathcal{A}}}[a],$$
$$H(p) = \mathbb{E}_{\mathcal{A}\sim D}\mathbb{E}_{a\sim p^{\mathcal{A}}}\left[(a - \hat{x}(p))(a - \hat{x}(p))^{\top}\right],$$
$$\hat{H}(p) = \mathbb{E}_{\mathcal{A}\sim\hat{D}}\mathbb{E}_{a\sim p^{\mathcal{A}}}\left[(a - \hat{x}(p))(a - \hat{x}(p))^{\top}\right],$$
$$\boldsymbol{H}(p) = \mathbb{E}_{\mathcal{A}\sim D}\mathbb{E}_{a\sim p^{\mathcal{A}}}\left[\boldsymbol{a}\boldsymbol{a}^{\top}\right],$$
$$\hat{\boldsymbol{H}}(p) = \mathbb{E}_{\mathcal{A}\sim\hat{D}}\mathbb{E}_{a\sim p^{\mathcal{A}}}\left[\boldsymbol{a}\boldsymbol{a}^{\top}\right],$$
$$\hat{\Sigma}(p) = \hat{H}(p) + \beta I,$$
$$\hat{\boldsymbol{\Sigma}}(p) = \hat{\boldsymbol{H}}(p) + \beta\boldsymbol{I},$$

*where $\beta = \frac{5d\log(6d/\delta)}{n}$.*

**Lemma 12.** *Under the setting of Definition 11, for any fixed $p$, with probability at least $1 - \delta$,*

$$\hat{H}(p) + \frac{4d\log(6d/\delta)}{n}I \succeq \frac{1}{2}H(p),$$
$$\hat{\boldsymbol{H}}(p) + \frac{3d\log(d/\delta)}{n}\boldsymbol{I} \succeq \frac{1}{2}\boldsymbol{H}(p).$$

*Proof.* In this proof, we use $\hat{x}, x, \hat{H}, H, \hat{\boldsymbol{H}}, \boldsymbol{H}$ to denote $\hat{x}(p), x(p), \hat{H}(p), H(p), \hat{\boldsymbol{H}}(p), \boldsymbol{H}(p)$ since $p$ is fixed throughout the proof.

Since $\|a\| \leq 1$, $\boldsymbol{H} \preceq 2I$ and $\hat{\boldsymbol{H}} \preceq 2I$. Thus, we can directly apply Corollary 10 with $c = 2$ to get with probability $1 - \frac{\delta}{3}$

$$\hat{\boldsymbol{H}} + \frac{3d\log(3d/\delta)}{n}\boldsymbol{I} \succeq \frac{1}{2}\boldsymbol{H}.$$

To prove the first inequality, we first decompose $H$ and $\hat{H}$

$$
\begin{aligned}
H &= \mathbb{E}_{\mathcal{A}\sim D}\mathbb{E}_{a\sim p^{\mathcal{A}}}\left[(a-\hat{x})(a-\hat{x})^\top\right] \\
&= \mathbb{E}_{\mathcal{A}\sim D}\mathbb{E}_{a\sim p^{\mathcal{A}}}\left[(a-x+x-\hat{x})(a-x+x-\hat{x})^\top\right] \\
&= \mathbb{E}_{\mathcal{A}\sim D}\mathbb{E}_{a\sim p^{\mathcal{A}}}\left[(a-x)(a-x)^\top\right] + (x-\hat{x})(x-\hat{x})^\top \quad (\text{because } \mathbb{E}_{\mathcal{A}\sim D}\mathbb{E}_{a\sim p^{\mathcal{A}}}(a-x)=0)
\end{aligned}
\tag{11}
$$

$$
\begin{aligned}
\hat{H} &= \mathbb{E}_{\mathcal{A}\sim \hat{D}}\mathbb{E}_{a\sim p^{\mathcal{A}}}\left[(a-\hat{x})(a-\hat{x})^\top\right] \\
&= \mathbb{E}_{\mathcal{A}\sim \hat{D}}\mathbb{E}_{a\sim p^{\mathcal{A}}}\left[(a-x+x-\hat{x})(a-x+x-\hat{x})^\top\right] \\
&= \mathbb{E}_{\mathcal{A}\sim \hat{D}}\mathbb{E}_{a\sim p^{\mathcal{A}}}\left[(a-x)(a-x)^\top\right] - (x-\hat{x})(x-\hat{x})^\top \\
&\qquad\qquad\qquad (\text{because } \mathbb{E}_{\mathcal{A}\sim \hat{D}}\mathbb{E}_{a\sim p^{\mathcal{A}}}(a-x)=\hat{x}-x)
\end{aligned}
\tag{12}
$$

From Hoeffding inequality (Lemma 7) and union bound, with probability $1-\frac{\delta}{3}$, for all $k\in[d]$, we have

$$
|\mathrm{e}_k^\top x - \mathrm{e}_k^\top \hat{x}| \leq \sqrt{\frac{1}{2n}\log\left(\frac{6d}{\delta}\right)},
$$

which implies that $\mathrm{e}_k^\top (x-\hat{x})(x-\hat{x})^\top \mathrm{e}_k \leq \frac{1}{2n}\log(\frac{6d}{\delta})$ for all $k$, and thus

$$
(x-\hat{x})(x-\hat{x})^\top \preceq \frac{1}{2n}\log\left(\frac{6d}{\delta}\right)I.
\tag{13}
$$

By directly applying Corollary 10 with $c=2$, we get with probability at least $1-\frac{\delta}{3}$,

$$
\mathbb{E}_{\mathcal{A}\sim \hat{D}}\mathbb{E}_{a\sim p^{\mathcal{A}}}\left[(a-x)(a-x)^\top\right] + \frac{3d\log(3d/\delta)}{n}I \succeq \frac{1}{2}\mathbb{E}_{\mathcal{A}\sim D}\mathbb{E}_{a\sim p^{\mathcal{A}}}\left[(a-x)(a-x)^\top\right]
$$

Further using Eq. (11), Eq. (12) and Eq. (13), we get with probability at least $1-\frac{2\delta}{3}$,

$$
\hat{H} + \frac{4d\log(6d/\delta)}{n}I \succeq \frac{1}{2}H
$$

Taking union bound for both inequality finishes the proof. $\qquad\square$

**Lemma 13.** *Under the setting of Definition 11, for any fixed policy $p$, with probability at least $1-\mathcal{O}(\delta)$,*

$$
\|x(p)-\hat{x}(p)\|_{\hat{\Sigma}(p)^{-1}}^2 \leq \mathcal{O}\left(\frac{d\log(d/\delta)}{n}\right)
$$

*Proof.* In this proof, we use $\hat{x}, x, \hat{H}, H, \hat{\boldsymbol{H}}, \boldsymbol{H}, \hat{\Sigma}, \hat{\boldsymbol{\Sigma}}$ to denote $\hat{x}(p), x(p), \hat{H}(p), H(p), \hat{\boldsymbol{H}}(p), \boldsymbol{H}(p), \hat{\Sigma}(p), \hat{\boldsymbol{\Sigma}}(p)$ since $p$ is fixed throughout the proof.

We first rewrite $H$.

$$
\begin{aligned}
H &= \mathbb{E}_{\mathcal{A}\sim D}\mathbb{E}_{a\sim p^{\mathcal{A}}}\left[(a-\hat{x})(a-\hat{x})^\top\right] \\
&= \mathbb{E}_{\mathcal{A}\sim D}\mathbb{E}_{a\sim p^{\mathcal{A}}}\left[(a-x+x-\hat{x})(a-x+x-\hat{x})^\top\right] \\
&= \mathbb{E}_{\mathcal{A}\sim D}\mathbb{E}_{a\sim p^{\mathcal{A}}}\left[(a-x)(a-x)^\top\right] + (x-\hat{x})(x-\hat{x})^\top \quad (\text{because } \mathbb{E}_{\mathcal{A}\sim D}\mathbb{E}_{a\sim p^{\mathcal{A}}}(a-x)=0)
\end{aligned}
\tag{14}
$$

To simplify analysis, we perform diagonalization. Suppose that $\mathbb{E}_{\mathcal{A}\sim D}\mathbb{E}_{a\sim p^{\mathcal{A}}}[(a-x)(a-x)^\top]$ admits the following eigen-decomposition:

$$
\mathbb{E}_{\mathcal{A}\sim D}\mathbb{E}_{a\sim p^{\mathcal{A}}}[(a-x)(a-x)^\top] = V\Lambda V^\top
$$

where $V$ is an orthogonal matrix and $\Lambda$ is a diagonal matrix. By Lemma 12 and the definition of $\beta$ in Definition 11, we have with probability $1-\delta$,

$$
\hat{\Sigma} \succeq \frac{1}{2}H + \rho I \succeq \frac{1}{2}V\Lambda V^\top + \rho I
$$

with some $\rho = \Theta\left(\frac{d\log(d/\delta)}{n}\right)$, where the second inequality is by Eq. (14). Thus,

$$\|x - \hat{x}\|^2_{\hat{\Sigma}^{-1}} = (x - \hat{x})^\top \hat{\Sigma}^{-1}(x - \hat{x})$$

$$\leq (x - \hat{x})^\top \left(\frac{1}{2}V\Lambda V^\top + \rho I\right)^{-1}(x - \hat{x})$$

$$= (\hat{x} - x)^\top V \left(\frac{1}{2}\Lambda + \rho I\right)^{-1} V^\top(\hat{x} - x).$$

Define

$$\Delta_k = \mathrm{e}_k^\top V^\top(\hat{x} - x) = \frac{1}{n}\sum_{i=1}^n \underbrace{\mathrm{e}_k^\top V^\top \mathbb{E}_{a\sim p^{\mathcal{A}_i}}[a]}_{\text{Define as } Z_k^{(i)}} - \underbrace{\mathrm{e}_k^\top V^\top \mathbb{E}_{\mathcal{A}\sim D}\mathbb{E}_{a\sim p^{\mathcal{A}}}[a]}_{\text{Define as } Z_k}$$

Since $\mathbb{E}_{\mathcal{A}_i\sim D}\left[Z_k^{(i)}\right] = Z_k$, by Bernstein's inequality, with probability at least $1 - \delta$, we have

$$|\Delta_k| \leq \mathcal{O}\left(\sqrt{\frac{\mathrm{Var}(Z_k^{(i)})\log(d/\delta)}{n}} + \frac{\log(d/\delta)}{n}\right) \tag{15}$$

for all $k$, where

$$\mathrm{Var}(Z_k^{(i)}) = \mathbb{E}_{\mathcal{A}\sim D}\left[\left(\mathrm{e}_k^\top V^\top \mathbb{E}_{a\sim p^{\mathcal{A}}}[a] - \mathrm{e}_k^\top V^\top x\right)^2\right].$$

On the other hand,

$$\Lambda_{kk} = \mathrm{e}_k^\top \mathbb{E}_{\mathcal{A}\sim D}\mathbb{E}_{a\sim p^{\mathcal{A}}}[V^\top(a - x)(a - x)^\top V]\mathrm{e}_k$$

$$= \mathbb{E}_{\mathcal{A}\sim D}\mathbb{E}_{a\sim p^{\mathcal{A}}}\left[\left(\mathrm{e}_k^\top V^\top a - \mathrm{e}_k^\top V^\top x\right)^2\right].$$

From Jensen's inequality,

$$\Lambda_{kk} = \mathbb{E}_{\mathcal{A}\sim D}\mathbb{E}_{a\sim p^{\mathcal{A}}}\left[\left(\mathrm{e}_k^\top V^\top a - \mathrm{e}_k^\top V^\top x\right)^2\right] \geq \mathbb{E}_{\mathcal{A}\sim D}\left[\left(\mathrm{e}_k^\top V^\top \mathbb{E}_{a\sim p^{\mathcal{A}}}[a] - \mathrm{e}_k^\top V^\top x\right)^2\right] = \mathrm{Var}(Z_k^{(i)})$$

Thus,

$$\|x - \hat{x}\|^2_{\hat{\Sigma}^{-1}} \leq (\hat{x} - x)^\top V \left(\frac{1}{2}\Lambda + \rho I\right)^{-1} V^\top(\hat{x} - x)$$

$$= \sum_{k=1}^d \frac{(\Delta_k)^2}{\frac{1}{2}\Lambda_{kk} + \rho}$$

$$\leq \mathcal{O}\left(\frac{\log(d/\delta)}{n}\sum_{k=1}^d \frac{\mathrm{Var}(Z_k^{(i)}) + \frac{\log(d/\delta)}{n}}{\Lambda_{kk} + \rho}\right) \qquad \text{(by Eq. (15))}$$

$$\leq \mathcal{O}\left(\frac{d\log(d/\delta)}{n}\right). \qquad (\Lambda_{kk} \geq \mathrm{Var}(Z_k^{(i)}) \text{ and } \rho = \Theta(\frac{d\log(d/\delta)}{n}))$$

$\square$

**Lemma 14.** *Under the setting of Definition 11, for any fixed policy $p$, with probability at least $1 - \mathcal{O}(\delta)$,*

$$\|(\hat{\Sigma}(p) - H(p))y\|^2_{\hat{\Sigma}(p)^{-1}} \leq \mathcal{O}\left(\frac{d\log(d/\delta)}{n}\right)$$

*for any $y \in \mathbb{B}_2^d$.*

*Proof.* In this proof, we use $\hat{x}, x, \hat{H}, H, \hat{\boldsymbol{H}}, \boldsymbol{H}, \hat{\Sigma}, \hat{\boldsymbol{\Sigma}}$ to denote $\hat{x}(p), x(p), \hat{H}(p), H(p), \hat{\boldsymbol{H}}(p), \boldsymbol{H}(p), \hat{\Sigma}(p), \hat{\boldsymbol{\Sigma}}(p)$ since $p$ is fixed throughout the proof.

First, we re-write $H$ and $\hat{H}$:

$$H = \mathbb{E}_{\mathcal{A}\sim D}\mathbb{E}_{a\sim p^{\mathcal{A}}}\left[(a-\hat{x})(a-\hat{x})^{\top}\right]$$
$$= \mathbb{E}_{\mathcal{A}\sim D}\mathbb{E}_{a\sim p^{\mathcal{A}}}\left[(a-x+x-\hat{x})(a-x+x-\hat{x})^{\top}\right]$$
$$= \mathbb{E}_{\mathcal{A}\sim D}\mathbb{E}_{a\sim p^{\mathcal{A}}}\left[(a-x)(a-x)^{\top}\right] + (x-\hat{x})(x-\hat{x})^{\top} \quad (\text{because } \mathbb{E}_{\mathcal{A}\sim D}\mathbb{E}_{a\sim p^{\mathcal{A}}}(a-x)=0)$$

$$(16)$$

$$\hat{H} = \mathbb{E}_{\mathcal{A}\sim\hat{D}}\mathbb{E}_{a\sim p^{\mathcal{A}}}\left[(a-\hat{x})(a-\hat{x})^{\top}\right]$$
$$= \mathbb{E}_{\mathcal{A}\sim\hat{D}}\mathbb{E}_{a\sim p^{\mathcal{A}}}\left[(a-x+x-\hat{x})(a-x+x-\hat{x})^{\top}\right]$$
$$= \mathbb{E}_{\mathcal{A}\sim\hat{D}}\mathbb{E}_{a\sim p^{\mathcal{A}}}\left[(a-x)(a-x)^{\top}\right] - (x-\hat{x})(x-\hat{x})^{\top}$$
$$(\text{because } \mathbb{E}_{\mathcal{A}\sim\hat{D}}\mathbb{E}_{a\sim p^{\mathcal{A}}}(a-x)=\hat{x}-x)$$

Then, by definition (in Definition 11) and the calculation above,

$$\hat{\Sigma} - H$$
$$= \hat{H} - H + \beta I$$
$$= \underbrace{\frac{1}{n}\sum_{i=1}^{n}\mathbb{E}_{a\sim p^{\mathcal{A}_i}}\left[(a-x)(a-x)^{\top}\right] - \mathbb{E}_{\mathcal{A}\sim D}\mathbb{E}_{a\sim p^{\mathcal{A}}}\left[(a-x)(a-x)^{\top}\right] - 2(x-\hat{x})(x-\hat{x})^{\top} + \beta I}_{\text{define this as } \Gamma}.$$

Using $\|a+b+c\|^2 \le 3\|a\|^2 + 3\|b\|^2 + 3\|c\|^2$, we have

$$\|(\hat{\Sigma}-H)y\|^2_{\hat{\Sigma}^{-1}} \le 3\|\Gamma y\|^2_{\hat{\Sigma}^{-1}} + 12\|(x-\hat{x})(x-\hat{x})^{\top}y\|^2_{\hat{\Sigma}^{-1}} + \beta^2\|y\|^2_{\hat{\Sigma}^{-1}}$$
$$\le 3\|\Gamma y\|^2_{\hat{\Sigma}^{-1}} + 12\|x-\hat{x}\|^2_{\hat{\Sigma}^{-1}} + \mathcal{O}(\beta). \tag{17}$$

The second and third term are bounded by $\mathcal{O}\left(\frac{d\log(d/\delta)}{n}\right)$ using Lemma 13 and the definition of $\beta$, with probability at least $1 - \mathcal{O}(\delta)$. Below, we further deal with the first term. To simplify analysis, we perform diagonalization. Suppose that $\mathbb{E}_{\mathcal{A}\sim D}\mathbb{E}_{a\sim p^{\mathcal{A}}}[(a-x)(a-x)^{\top}]$ admits the following eigen-decomposition:

$$\mathbb{E}_{\mathcal{A}\sim D}\mathbb{E}_{a\sim p^{\mathcal{A}}}[(a-x)(a-x)^{\top}] = V\Lambda V^{\top}$$

where $V$ is an orthogonal matrix and $\Lambda$ is a diagonal matrix. Then

$$\|\Gamma y\|^2_{\hat{\Sigma}^{-1}} = y^{\top}\Gamma\hat{\Sigma}^{-1}\Gamma y = (V^{\top}y)^{\top}(V^{\top}\Gamma V)(V^{\top}\hat{\Sigma}V)^{-1}(V^{\top}\Gamma V)(V^{\top}y). \tag{18}$$

Below, we further deal with the $V^{\top}\Gamma V$ and $V^{\top}\Lambda V$ terms in Eq. (18). By Lemma 12, with probability at least $1-\delta$,

$$\hat{\Sigma} \succeq \frac{1}{2}H + \rho I \succeq \frac{1}{2}V\Lambda V^{\top} + \rho I,$$

for some $\rho = \Theta\left(\frac{d\log(d/\delta)}{n}\right)$, where we use Eq. (16) in the second inequality. Therefore,

$$V^{\top}\hat{\Sigma}V \succeq \frac{1}{2}\Lambda + \rho I. \tag{19}$$

Next, denote $\Delta = V^{\top}\Gamma V$. By definition, it can be written as the following:

$$\Delta = \frac{1}{n}\sum_{i=1}^{n}\underbrace{\mathbb{E}_{a\sim p^{\mathcal{A}_i}}\left[V^{\top}(a-x)(a-x)^{\top}V\right]}_{\text{defining this as } \Lambda^{(i)}} - \underbrace{\mathbb{E}_{\mathcal{A}\sim D}\mathbb{E}_{a\sim p^{\mathcal{A}}}\left[V^{\top}(a-x)(a-x)^{\top}V\right]}_{=\Lambda}$$

with $\Lambda^{(i)}$ being i.i.d. samples with mean $\mathbb{E}[\Lambda^{(i)}] = \Lambda$. While these are $d\times d$ matrices, we will apply concentration inequalities to individual entries.

Let $\lambda_{ikh} = \mathrm{e}_k^{\top}\Lambda^{(i)}\mathrm{e}_h$ be the $(k,h)$-th entry of $\Lambda^{(i)}$. Notice that $\mathbb{E}[\lambda_{ikh}] = \mathrm{e}_k^{\top}\Lambda\mathrm{e}_h = \Lambda_{kh}$, the $(k,h)$-th entry of $\Lambda$.

By Bernstein's inequality, with probability at least $1 - \delta$, we have

$$|\Delta_{kh}| = \left| \frac{1}{n} \sum_{i=1}^{n} (\lambda_{ikh} - \Lambda_{kh}) \right| \leq \mathcal{O} \left( \sqrt{\frac{\mathrm{Var}(\lambda_{ikh}) \log(d/\delta)}{n}} + \frac{\log(d/\delta)}{n} \right). \tag{20}$$

With the manipulations and notations above, we continue to bound Eq. (18) by

$$
\begin{aligned}
\|\Gamma y\|_{\hat{\Sigma}^{-1}}^2 &= y'^\top \Delta (V^\top \hat{\Sigma} V)^{-1} \Delta y' && \text{(let } y' = V^\top y) \\
&\leq 2 y'^\top \Delta (\Lambda + \rho I)^{-1} \Delta y' && \text{(by Eq. (19))} \\
&\leq 2 \mathrm{Tr} \left( \Delta (\Lambda + \rho I)^{-1} \Delta \right)
\end{aligned}
$$

By direct expansion and the fact that $\Lambda$ is diagonal,

$$
\begin{aligned}
\mathrm{Tr} \left( \Delta (\Lambda + \rho I)^{-1} \Delta \right) &= \sum_{k=1}^{d} \left( \Delta (\Lambda + \rho I)^{-1} \Delta \right)_{kk} \\
&= \sum_{k=1}^{d} \sum_{h=1}^{d} \frac{\Delta_{kh} \Delta_{hk}}{\Lambda_{hh} + \rho} \\
&\leq \mathcal{O} \left( \sum_{k=1}^{d} \sum_{h=1}^{d} \frac{1}{\Lambda_{hh} + \rho} \left( \frac{\mathrm{Var}(\lambda_{ikh}) \log(d/\delta)}{n} + \frac{\log^2(d/\delta)}{n^2} \right) \right) \\
&&& \text{(by Eq. (20))} \\
&\leq \mathcal{O} \left( \sum_{k=1}^{d} \sum_{h=1}^{d} \frac{1}{\Lambda_{hh} + \rho} \frac{\mathbb{E}(\lambda_{ikh}^2) \log(d/\delta)}{n} + \frac{d^2 \log^2(d/\delta)}{\rho n^2} \right) \tag{21}
\end{aligned}
$$

By definition,

$$\lambda_{ikh} = \mathbb{E}_{a \sim p^{A_i}} \left[ e_k V^\top (a - x)(a - x)^\top V e_h \right]$$

and thus

$$
\begin{aligned}
\sum_{k=1}^{d} \lambda_{ikh}^2 &\leq \mathbb{E}_{a \sim p^{A_i}} \left[ \sum_{k=1}^{d} \left( e_k V^\top (a - x)(a - x)^\top V e_h \right)^2 \right] \\
&= \mathbb{E}_{a \sim p^{A_i}} \left[ \sum_{k=1}^{d} e_h^\top V^\top (a - x)(a - x)^\top V e_k e_k^\top V^\top (a - x)(a - x)^\top V e_h \right] \\
&= \mathbb{E}_{a \sim p^{A_i}} \left[ e_h^\top V^\top (a - x)(a - x)^\top (a - x)(a - x)^\top V e_h \right] \\
&\leq \mathbb{E}_{a \sim p^{A_i}} \left[ e_h^\top V^\top (a - x)(a - x)^\top V e_h \right] \\
&= \lambda_{ihh}
\end{aligned}
$$

and $\sum_{k=1}^{d} \mathbb{E}[\lambda_{ikh}^2] \leq \mathbb{E}[\lambda_{ihh}] = \Lambda_{hh}$. Continuing from Eq. (21) and using that $\rho = \Theta \left( \frac{d \log(d/\delta)}{n} \right)$,

$$\mathrm{Tr} \left( \Delta (\Lambda + \rho I)^{-1} \Delta \right) \leq \mathcal{O} \left( \sum_{h=1}^{d} \frac{\Lambda_{hh} \log(d/\delta)}{(\Lambda_{hh} + \rho) n} + \frac{d^2 \log^2(d/\delta)}{n^2} \right) \leq \mathcal{O} \left( \frac{d \log(d/\delta)}{n} \right).$$

This gives a bound on $\|\Gamma y\|_{\hat{\Sigma}^{-1}}^2$ and finishes the proof after combining Eq. (17).

$\square$

### C.3 Union Bound over Policies

In Lemma 12, Lemma 13, and Lemma 14, we have obtained the desired concentration inequalities *under a fixed policy $p$.* In this subsection, we proceed to take union bound over *all policies* that are possibly used by Algorithm 1.

The set of policies that could be generated by [Algorithm 1](#) is the following:

$$\mathbf{P} = \left\{ p : \ \widehat{\text{Cov}}(p^{\mathcal{A}}) = \underset{\boldsymbol{H} \in \mathcal{H}^{\mathcal{A}}}{\arg\min} \left\{ \langle \boldsymbol{H}, \boldsymbol{Z} \rangle + F(\boldsymbol{H}) \right\}, \text{for } \boldsymbol{Z} \in \mathcal{Z} \right\}$$

where $\mathcal{Z} = [-T^2, T^2]^{(d+1) \times (d+1)} \cap \mathbb{S}$ with $\mathbb{S}$ denoting the set of symmetric matrices. To see this, notice that [Algorithm 1](#) at round $t$ corresponds to the policy defined above with $\boldsymbol{Z} = \eta_t \sum_{s=1}^{t-1} (\hat{\gamma}_s - \alpha_s \hat{\boldsymbol{\Sigma}}_s^{-1})$.

Our goal is to construct a $\epsilon$-cover $\mathbf{P}'$ so that every policy $p \in \mathbf{P}$ can find a policy $p' \in \mathbf{P}'$ making $-\epsilon I \preceq \widehat{\text{Cov}}(p^{\mathcal{A}}) - \widehat{\text{Cov}}(p'^{\mathcal{A}}) \preceq \epsilon I$ on *every* action set $\mathcal{A}$. The size of such a cover is bounded in the Proposition below.

**Lemma 15.** *There exists an $\epsilon$-cover $\mathbf{P}'$ of $\mathbf{P}$ with size $\log |\mathbf{P}'| = \mathcal{O}\left( d^2 \log \frac{d}{\epsilon} \right)$ such that for any $p \in \mathbf{P}$, there exists an $p' \in \mathbf{P}'$ satisfying*

$$\left\| \widehat{\text{Cov}}(p^{\mathcal{A}}) - \widehat{\text{Cov}}(p'^{\mathcal{A}}) \right\|_F \leq \epsilon$$

*for all $\mathcal{A}$.*

*Proof.* It is straightforward to construct an $\frac{\epsilon}{4}$-cover $\mathcal{C}$ for $\mathcal{Z} = [-T^2, T^2]^{(d+1) \times (d+1)} \cap \mathbb{S}$ in Frobenius norm with size $|\mathcal{C}| = \left( \frac{24(d+1)^2}{\epsilon} \right)^{(d+1)^2}$ (Exercise 27.6 of [Lattimore and Szepesvári [2020]](#)). Now define $\mathbf{P}'$ as

$$\mathbf{P}' = \left\{ p : \ \widehat{\text{Cov}}(p^{\mathcal{A}}) = \underset{\boldsymbol{H} \in \mathcal{H}^{\mathcal{A}}}{\arg\min} \left\{ \langle \boldsymbol{H}, \boldsymbol{Z} \rangle + F(\boldsymbol{H}) \right\}, \text{for } \boldsymbol{Z} \in \mathcal{C} \right\} \tag{22}$$

Below, we show that this is a $\epsilon$-cover for $\mathbf{P}$.

Consider two policies $p_1$ and $p_2$ defined as the following:

$$\widehat{\text{Cov}}(p_1^{\mathcal{A}}) = \underset{\boldsymbol{H} \in \mathcal{H}^{\mathcal{A}}}{\arg\min} \left\{ \langle \boldsymbol{H}, \boldsymbol{Z}_1 \rangle + F(\boldsymbol{H}) \right\}$$
$$\widehat{\text{Cov}}(p_2^{\mathcal{A}}) = \underset{\boldsymbol{H} \in \mathcal{H}^{\mathcal{A}}}{\arg\min} \left\{ \langle \boldsymbol{H}, \boldsymbol{Z}_2 \rangle + F(\boldsymbol{H}) \right\}$$

with $\|\boldsymbol{Z}_1 - \boldsymbol{Z}_2\|_F \leq \frac{\epsilon}{4}$. Consider an arbitrary $\mathcal{A}$ and define $\boldsymbol{H}_1 = \widehat{\text{Cov}}(p_1^{\mathcal{A}})$, $\boldsymbol{H}_2 = \widehat{\text{Cov}}(p_2^{\mathcal{A}})$. Below we show $\|\boldsymbol{H}_1 - \boldsymbol{H}_2\|_F \leq \epsilon$.

Since $F(\boldsymbol{H})$ is convex for $\boldsymbol{H}$, from the first-order optimality condition for convex function, we have

$$\langle \boldsymbol{H}_1, \boldsymbol{Z}_1 \rangle + F(\boldsymbol{H}_1) \leq \langle \boldsymbol{H}_2, \boldsymbol{Z}_1 \rangle + F(\boldsymbol{H}_2) - D_F(\boldsymbol{H}_2, \boldsymbol{H}_1)$$
$$= \langle \boldsymbol{H}_2, \boldsymbol{Z}_2 \rangle + \langle \boldsymbol{H}_2, \boldsymbol{Z}_1 - \boldsymbol{Z}_2 \rangle + F(\boldsymbol{H}_2) - D_F(\boldsymbol{H}_2, \boldsymbol{H}_1)$$
$$\langle \boldsymbol{H}_2, \boldsymbol{Z}_2 \rangle + F(\boldsymbol{H}_2) \leq \langle \boldsymbol{H}_1, \boldsymbol{Z}_2 \rangle + F(\boldsymbol{H}_1) - D_F(\boldsymbol{H}_1, \boldsymbol{H}_2)$$
$$= \langle \boldsymbol{H}_1, \boldsymbol{Z}_1 \rangle + \langle \boldsymbol{H}_1, \boldsymbol{Z}_2 - \boldsymbol{Z}_1 \rangle + F(\boldsymbol{H}_1) - D_F(\boldsymbol{H}_1, \boldsymbol{H}_2)$$

Adding up these the two inequalities, we get

$$2 \min\{ D_F(\boldsymbol{H}_1, \boldsymbol{H}_2), D_F(\boldsymbol{H}_2, \boldsymbol{H}_1) \} \leq D_F(\boldsymbol{H}_1, \boldsymbol{H}_2) + D_F(\boldsymbol{H}_2, \boldsymbol{H}_1) \leq \langle \boldsymbol{Z}_1 - \boldsymbol{Z}_2, \boldsymbol{H}_2 - \boldsymbol{H}_1 \rangle$$

Since the second order directional derivative for $F$ is $D^2 F(\boldsymbol{H})[\boldsymbol{X}, \boldsymbol{X}] = \text{Tr}(\boldsymbol{X} \boldsymbol{H}^{-1} \boldsymbol{X} \boldsymbol{H}^{-1})$ for any symmetric matrix $\boldsymbol{X}$, from the Taylor series, there exists $\boldsymbol{H}'$ that is a line segment between $\boldsymbol{H}_1$ and $\boldsymbol{H}_2$ such that

$$\|\boldsymbol{H}_1 - \boldsymbol{H}_2\|_{\nabla^2 F(\boldsymbol{H}')}^2 = 2 \min\{ D_F(\boldsymbol{H}_1, \boldsymbol{H}_2), D_F(\boldsymbol{H}_2, \boldsymbol{H}_1) \} \leq \langle \boldsymbol{Z}_1 - \boldsymbol{Z}_2, \boldsymbol{H}_2 - \boldsymbol{H}_1 \rangle$$
$$\leq \|\boldsymbol{Z}_1 - \boldsymbol{Z}_2\|_{\nabla^{-2} F(\boldsymbol{H}')} \|\boldsymbol{H}_1 - \boldsymbol{H}_2\|_{\nabla^2 F(\boldsymbol{H}')} \tag{Lemma 8}$$

Thus we have $\|\boldsymbol{H}_1 - \boldsymbol{H}_2\|_{\nabla^2 F(\boldsymbol{H}')} \leq \|\boldsymbol{Z}_1 - \boldsymbol{Z}_2\|_{\nabla^{-2} F(\boldsymbol{H}')}$. Since $\|a\|_2 \leq 1$, $\boldsymbol{H}' \preceq 2\boldsymbol{I}$. The left-hand side and right-hand side can be bounded as follows,

$$\|\boldsymbol{H}_1 - \boldsymbol{H}_2\|_{\nabla^2 F(\boldsymbol{H}')} = \sqrt{\text{Tr}\left( (\boldsymbol{H}_1 - \boldsymbol{H}_2)(\boldsymbol{H}')^{-1} (\boldsymbol{H}_1 - \boldsymbol{H}_2)(\boldsymbol{H}')^{-1} \right)} \geq \frac{1}{2} \|\boldsymbol{H}_1 - \boldsymbol{H}_2\|_F$$
$$\|\boldsymbol{Z}_1 - \boldsymbol{Z}_2\|_{\nabla^{-2} F(\boldsymbol{H}')} = \sqrt{\text{Tr}\left( (\boldsymbol{Z}_1 - \boldsymbol{Z}_2) \boldsymbol{H}' (\boldsymbol{Z}_1 - \boldsymbol{Z}_2) \boldsymbol{H}' \right)} \leq 2 \|\boldsymbol{Z}_1 - \boldsymbol{Z}_2\|_F \leq \frac{\epsilon}{2}$$

Combining the three inequalities above, we conclude that
$$\|\boldsymbol{H}_1 - \boldsymbol{H}_2\|_F \le 2\|\boldsymbol{H}_1 - \boldsymbol{H}_2\|_{\nabla^2 F(\boldsymbol{H}')} \le 2\|\boldsymbol{Z}_1 - \boldsymbol{Z}_2\|_{\nabla^{-2} F(\boldsymbol{H}')} \le 4\|\boldsymbol{Z}_1 - \boldsymbol{Z}_2\|_F \le \epsilon.$$
$$-\epsilon \boldsymbol{I} \preceq \boldsymbol{H}_1 - \boldsymbol{H}_2 \preceq \epsilon \boldsymbol{I}.$$

$\square$

**Lemma 16.** *Suppose that $p, p'$ are two policies such that for all action set $\mathcal{A}$,*
$$\left\|\widehat{\text{Cov}}(p^{\mathcal{A}}) - \widehat{\text{Cov}}(p'^{\mathcal{A}})\right\|_F \le \epsilon \tag{23}$$

*Then all quantities defined in Definition 11 under $p$ and $p'$ are close. That is,*
$$\|x(p) - x(p')\| \le \epsilon \tag{24}$$
$$\|\hat{x}(p) - \hat{x}(p')\| \le \epsilon \tag{25}$$
$$\|H(p) - H(p')\|_F \le 7\epsilon \tag{26}$$
$$\|\hat{H}(p) - \hat{H}(p')\|_F \le 7\epsilon \tag{27}$$
$$\|\boldsymbol{H}(p) - \boldsymbol{H}(p')\|_F \le \epsilon \tag{28}$$
$$\|\hat{\boldsymbol{H}}(p) - \hat{\boldsymbol{H}}(p')\|_F \le \epsilon \tag{29}$$
$$\|\hat{\Sigma}(p) - \hat{\Sigma}(p')\|_F \le 7\epsilon \tag{30}$$
$$\|\hat{\boldsymbol{\Sigma}}(p) - \hat{\boldsymbol{\Sigma}}(p')\|_F \le \epsilon \tag{31}$$

*Proof.* Eq. (28) and Eq. (29) are direct consequences of Eq. (23) since $\boldsymbol{H}(p)$ and $\hat{\boldsymbol{H}}(p)$ are expectations of $\widehat{\text{Cov}}(p^{\mathcal{A}})$ over distributions over $\mathcal{A}$. Eq. (31) is directly implied by Eq. (29) because $\hat{\boldsymbol{\Sigma}}(p) = \hat{\boldsymbol{H}}(p) + \beta \boldsymbol{I}$.

To show Eq. (24) and Eq. (25), observe that by the definition of $x(p)$ and $\boldsymbol{H}(p)$,

$$\boldsymbol{H}(p) = \mathbb{E}_{\mathcal{A}\sim D}\mathbb{E}_{a\sim p^{\mathcal{A}}}\begin{bmatrix} aa^\top & a \\ a^\top & 1 \end{bmatrix} = \begin{bmatrix} \mathbb{E}_{\mathcal{A}\sim D}\mathbb{E}_{a\sim p^{\mathcal{A}}}[aa^\top] & \mathbb{E}_{\mathcal{A}\sim D}\mathbb{E}_{a\sim p^{\mathcal{A}}}[a] \\ \mathbb{E}_{\mathcal{A}\sim D}\mathbb{E}_{a\sim p^{\mathcal{A}}}[a^\top] & 1 \end{bmatrix}$$
$$= \begin{bmatrix} \mathbb{E}_{\mathcal{A}\sim D}\mathbb{E}_{a\sim p^{\mathcal{A}}}[aa^\top] & x(p) \\ x(p)^\top & 1 \end{bmatrix}$$

Therefore, $\|x(p) - x(p')\| \le \|\boldsymbol{H}(p) - \boldsymbol{H}(p')\|_F \le \epsilon$. Similarly, $\|\hat{x}(p) - \hat{x}(p')\| \le \|\hat{\boldsymbol{H}}(p) - \hat{\boldsymbol{H}}(p')\|_F \le \epsilon$.

If remains to show Eq. (26), Eq. (27) and Eq. (30). Next, we show Eq. (26):

$H(p) - H(p')$
$= \mathbb{E}_{\mathcal{A}\sim D}\left[\mathbb{E}_{a\sim p^{\mathcal{A}}}[(a - \hat{x}(p))(a - \hat{x}(p))^\top] - \mathbb{E}_{a\sim p'^{\mathcal{A}}}[(a - \hat{x}(p'))(a - \hat{x}(p'))^\top]\right]$
$= \mathbb{E}_{\mathcal{A}\sim D}\left[\mathbb{E}_{a\sim p^{\mathcal{A}}}[aa^\top] - \mathbb{E}_{a\sim p'^{\mathcal{A}}}[aa^\top]\right]$
$\quad - x(p)\hat{x}(p)^\top - \hat{x}(p)x(p)^\top + x(p')\hat{x}(p')^\top + \hat{x}(p')x(p')^\top \quad (\text{using } \mathbb{E}_{\mathcal{A}\sim D}\mathbb{E}_{a\sim p^{\mathcal{A}}}[a] = x(p))$
$\quad + \hat{x}(p)\hat{x}(p)^\top - \hat{x}(p')\hat{x}(p')^\top \tag{32}$

Using the property
$$\|ab^\top - cd^\top\|_F \le \|ab^\top - cb^\top\|_F + \|cb^\top - cd^\top\|_F \le \|a - c\|\|b\| + \|c\|\|b - d\|$$

we continue from Eq. (32) and bound

$\|H(p) - H(p')\|_F$
$\le \|\boldsymbol{H}(p) - \boldsymbol{H}(p')\|_F + 2(\|\hat{x}(p) - \hat{x}(p')\| + \|x(p) - x(p')\|) + \|\hat{x}(p) - \hat{x}(p')\| + \|\hat{x}(p) - \hat{x}(p')\|$
$\le 7\epsilon.$

Eq. (27) can be shown in the same manner, which further implies Eq. (30) by the definition of $\hat{\Sigma}(p)$.

$\square$

**Lemma 17.** *With probability $1 - \delta$, for all $t = 1, \cdots, T$,*

$$\hat{H}_t + \frac{50(d+1)^3 \log(3T/\delta)}{t-1} I \succeq \frac{1}{2} H_t,$$

$$\hat{\boldsymbol{H}}_t + \frac{50(d+1)^3 \log(3T/\delta)}{t-1} \boldsymbol{I} \succeq \frac{1}{2} \boldsymbol{H}_t.$$

*Proof.* Notice that $\hat{H}_t, \hat{\boldsymbol{H}}_t, H_t, \boldsymbol{H}_t$ corresponds to $\hat{H}(p_t), \hat{\boldsymbol{H}}(p_t), H(p_t), \boldsymbol{H}(p_t)$ defined in Definition 11 with $n = t - 1$. To show the lemma, our strategy is to argue the following two facts: 1) the two desired inequalities hold for all policies in the cover $\mathbf{P}'$ (defined in Eq. (22)) with high probability. This is simply by applying Lemma 12 with an union bound over policies in $\mathbf{P}'$. 2) $p_t$ is sufficiently close to the nearest element in $\mathbf{P}'$ so the desired inequalities still approximately hold.

By Lemma 15, we can find $p' \in \mathbf{P}'$ such that for all $\mathcal{A}$,

$$\left\| \widehat{\text{Cov}}(p_t^{\mathcal{A}}) - \widehat{\text{Cov}}(p'^{\mathcal{A}}) \right\|_F \le \epsilon.$$

By Lemma 16, it holds that

$$\|H(p_t) - H(p')\|_F \le 7\epsilon, \quad \|\hat{H}(p_t) - \hat{H}(p')\|_F \le 7\epsilon \tag{33}$$

$$\|\boldsymbol{H}(p_t) - \boldsymbol{H}(p')\|_F \le \epsilon, \quad \|\hat{\boldsymbol{H}}(p_t) - \hat{\boldsymbol{H}}(p')\|_F \le \epsilon \tag{34}$$

On the other hand, using Lemma 12 and union bound, with probability $1 - \delta$, we have

$$\hat{H}(p') + \frac{4d \log(6d|\mathbf{P}'|/\delta)}{n} I \succeq \frac{1}{2} H(p'), \tag{35}$$

$$\hat{\boldsymbol{H}}(p') + \frac{3d \log(d|\mathbf{P}'|/\delta)}{n} \boldsymbol{I} \succeq \frac{1}{2} \boldsymbol{H}(p'). \tag{36}$$

Combining Eq. (35) and Eq. (33), we get

$$\hat{H}(p_t) + 7\epsilon I + \frac{4d \log(6d|\mathbf{P}'|/\delta)}{n} I \succeq \hat{H}(p') + \frac{4d \log(6d|\mathbf{P}'|/\delta)}{n} I \succeq \frac{1}{2} H(p') \succeq \frac{1}{2} H(p_t) - \frac{7}{2}\epsilon I$$

which implies the first inequality in the lemma by plugging in the choice of $\epsilon = \frac{1}{T^3}$ and the upper bound of $\log |\mathbf{P}'|$ in Lemma 16. The second inequality in the lemma can be obtained similarly by combining Eq. (34) and Eq. (36). $\qquad \square$

**Lemma 18.** *With probability of at least $1 - \delta$, for all $t = 1, \cdots, T$,*

$$\|x_t - \hat{x}_t\|_{\hat{\Sigma}_t^{-1}}^2 \le \mathcal{O}\left( \frac{d^3 \log(dT/\delta)}{t} \right)$$

*Proof.* Notice that $x_t, \hat{x}_t, \hat{\Sigma}_t$ corresponds to $x(p_t), \hat{x}(p_t), \hat{\Sigma}(p_t)$ defined in Definition 11 with $n = t - 1$. To show the lemma, our strategy is to argue the following two facts: 1) the two desired inequalities hold for all policies in the cover $\mathbf{P}'$ with high probability. This is simply by applying Lemma 13 with an union bound over policies in $\mathbf{P}'$. 2) $p_t$ is sufficiently close to the nearest element in $\mathbf{P}'$ so the desired inequalities still approximately hold.

By Lemma 15, we can find $p' \in \mathbf{P}'$ such that for all $\mathcal{A}$,

$$\left\| \widehat{\text{Cov}}(p_t^{\mathcal{A}}) - \widehat{\text{Cov}}(p'^{\mathcal{A}}) \right\|_F \le \epsilon.$$

By Lemma 16, we have

$$\|x(p') - x(p_t)\| \le \epsilon, \quad \|\hat{x}(p') - \hat{x}(p_t)\| \le \epsilon, \quad \|\hat{\Sigma}(p') - \hat{\Sigma}(p_t)\|_F \le 7\epsilon \tag{37}$$

Thus,

$$\|x(p_t) - \hat{x}(p_t)\|^2_{\hat{\Sigma}(p_t)^{-1}}$$

$$= \left( \|x(p_t) - \hat{x}(p_t)\|^2_{\hat{\Sigma}(p_t)^{-1}} - \|x(p') - \hat{x}(p')\|^2_{\hat{\Sigma}(p')^{-1}} \right) + \|x(p') - \hat{x}(p')\|^2_{\hat{\Sigma}(p')^{-1}}$$

$$\leq \left( \|x(p_t) - \hat{x}(p_t)\|^2_{\hat{\Sigma}(p_t)^{-1}} - \|x(p') - \hat{x}(p')\|^2_{\hat{\Sigma}(p')^{-1}} \right) + \mathcal{O}\left( \frac{d \log(d|\mathbf{P}'|/\delta)}{t-1} \right)$$
(by Lemma 13 with an union bound over $\mathbf{P}'$)

$$= \theta_t^\top \hat{\Sigma}(p_t)^{-1} \theta_t - \theta'^\top \hat{\Sigma}(p')^{-1} \theta' + \mathcal{O}\left( \frac{d \log(d|\mathbf{P}'|/\delta)}{t-1} \right)$$
(define $\theta_t = x(p_t) - \hat{x}(p_t)$ and $\theta' = x(p') - \hat{x}(p')$)

$$= (\theta_t - \theta')^\top \hat{\Sigma}(p_t)^{-1} \theta_t + \theta'^\top \left( \hat{\Sigma}(p_t)^{-1} - \hat{\Sigma}(p')^{-1} \right) \theta_t + \theta'^\top \hat{\Sigma}(p')^{-1}(\theta_t - \theta') + \mathcal{O}\left( \frac{d \log(d|\mathbf{P}'|/\delta)}{t-1} \right)$$

$$\leq (\theta_t - \theta')^\top \left( \hat{\Sigma}(p_t)^{-1} \theta_t + \hat{\Sigma}(p')^{-1} \theta' \right) + \theta'^\top \hat{\Sigma}(p')^{-1} \left( \hat{\Sigma}(p') - \hat{\Sigma}(p_t) \right) \hat{\Sigma}(p_t)^{-1} \theta_t + \mathcal{O}\left( \frac{d \log(d|\mathbf{P}'|/\delta)}{t-1} \right)$$

The first two terms above can be bounded by the order of $\mathcal{O}(\epsilon t^2)$ by Eq. (37). Using the choice $\epsilon = \frac{1}{T^3}$ and recalling that $\log|\mathbf{P}'| = \mathcal{O}(d^2 \log(d/\epsilon))$ finishes the proof. $\qquad \square$

**Lemma 19.** *With probability of at least $1 - \delta$, for all $t = 1, 2, \ldots, T$,*

$$\|(\hat{\Sigma}_t - H_t) y_t\|^2_{\hat{\Sigma}_t^{-1}} \leq \mathcal{O}\left( \frac{d^3 \log(dT/\delta)}{t} \right)$$

*Proof.* Notice that $x_t, \hat{x}_t, \hat{\Sigma}_t$ corresponds to $x(p_t), \hat{x}(p_t), \hat{\Sigma}(p_t)$ defined in Definition 11 with $n = t-1$. To show the lemma, our strategy is to argue the following two facts: 1) the two desired inequalities hold for all policies in the cover $\mathbf{P}'$ with high probability. This is simply by applying Lemma 13 with an union bound over policies in $\mathbf{P}'$. 2) $p_t$ is sufficiently close to the nearest element in $\mathbf{P}'$ so the desired inequalities still approximately hold.

By Lemma 15, we can find $p' \in \mathbf{P}'$ such that for all $\mathcal{A}$,

$$\left\| \widehat{\text{Cov}}(p_t^{\mathcal{A}}) - \widehat{\text{Cov}}(p'^{\mathcal{A}}) \right\|_F \leq \epsilon.$$

By Lemma 16, we have

$$\|x(p') - x(p_t)\| \leq \epsilon, \quad \|\hat{x}(p') - \hat{x}(p_t)\| \leq \epsilon, \quad \|\hat{\Sigma}(p') - \hat{\Sigma}(p_t)\|_F \leq 7\epsilon \qquad (38)$$

Thus, for any $\|y_t\|_2 \leq 1$,

$$\|(\hat{\Sigma}(p_t) - H(p_t)) y_t\|^2_{\hat{\Sigma}(p_t)^{-1}}$$

$$= \left( \|(\hat{\Sigma}(p_t) - H(p_t)) y_t\|^2_{\hat{\Sigma}(p_t)^{-1}} - \|(\hat{\Sigma}(p') - H(p')) y_t\|^2_{\hat{\Sigma}(p')^{-1}} \right) + \|(\hat{\Sigma}(p') - H(p')) y_t\|^2_{\hat{\Sigma}(p')^{-1}}$$

$$\leq \left( \|(\hat{\Sigma}(p_t) - H(p_t)) y_t\|^2_{\hat{\Sigma}(p_t)^{-1}} - \|(\hat{\Sigma}(p') - H(p')) y_t\|^2_{\hat{\Sigma}(p')^{-1}} \right) + \mathcal{O}\left( \frac{d \log(d|\mathbf{P}'|/\delta)}{t-1} \right)$$
(by Lemma 14 with an union bound over $\mathbf{P}'$)

$$= \theta_t^\top \hat{\Sigma}(p_t)^{-1} \theta_t - \theta'^\top \hat{\Sigma}(p')^{-1} \theta' + \mathcal{O}\left( \frac{d \log(d|\mathbf{P}'|/\delta)}{t-1} \right)$$
(define $\theta_t = (\hat{\Sigma}(p_t) - H(p_t)) y_t$ and $\theta' = (\hat{\Sigma}(p') - H(p')) y_t$)

$$= (\theta_t - \theta')^\top \hat{\Sigma}(p_t)^{-1} \theta_t + \theta'^\top \left( \hat{\Sigma}(p_t)^{-1} - \hat{\Sigma}(p')^{-1} \right) \theta_t + \theta'^\top \hat{\Sigma}(p')^{-1}(\theta_t - \theta') + \mathcal{O}\left( \frac{d \log(d|\mathbf{P}'|/\delta)}{t-1} \right)$$

$$\leq (\theta_t - \theta')^\top \left( \hat{\Sigma}(p_t)^{-1} \theta_t + \hat{\Sigma}(p')^{-1} \theta' \right) + \theta'^\top \hat{\Sigma}(p')^{-1} \left( \hat{\Sigma}(p') - \hat{\Sigma}(p_t) \right) \hat{\Sigma}(p_t)^{-1} \theta_t + \mathcal{O}\left( \frac{d \log(d|\mathbf{P}'|/\delta)}{t-1} \right)$$

The first two terms above can be bounded by the order of $\mathcal{O}(\epsilon t^2)$ by Eq. (38). Plugging in the choice of $\epsilon = \frac{1}{T^3}$ and recalling that $\log|\mathbf{P}'| = \mathcal{O}(d^2 \log(d/\epsilon))$ finishes the proof. $\qquad \square$

## D  Regret Analysis

Consider the regret decomposition in Section 3.5.

$$\text{Reg}(u) = \mathbb{E}\left[\sum_{t=1}^{T}\left\langle a_t - u^{\mathcal{A}_t}, y_t\right\rangle\right] = \mathbb{E}\left[\sum_{t=1}^{T}\left\langle \boldsymbol{H}_t^{\mathcal{A}_t} - \boldsymbol{U}^{\mathcal{A}_t}, \gamma_t\right\rangle\right] = \mathbb{E}\left[\sum_{t=1}^{T}\left\langle \boldsymbol{H}_t^{\mathcal{A}_0} - \boldsymbol{U}^{\mathcal{A}_0}, \gamma_t\right\rangle\right]$$

$$\leq \underbrace{\mathbb{E}\left[\sum_{t=1}^{T}\left\langle \boldsymbol{H}_t^{\mathcal{A}_0} - \boldsymbol{U}^{\mathcal{A}_0}, \gamma_t - \hat{\gamma}_t\right\rangle\right]}_{\textbf{Bias}} + \underbrace{\mathbb{E}\left[\sum_{t=1}^{T}\left\langle \boldsymbol{H}_t^{\mathcal{A}_0} - \boldsymbol{U}^{\mathcal{A}_0}, \alpha_t\hat{\boldsymbol{\Sigma}}_t^{-1}\right\rangle\right]}_{\textbf{Bonus}} + \underbrace{\mathbb{E}\left[\sum_{t=1}^{T}\left\langle \boldsymbol{H}_t^{\mathcal{A}_0} - \boldsymbol{U}^{\mathcal{A}_0}, \hat{\gamma}_t - \alpha_t\hat{\boldsymbol{\Sigma}}_t^{-1}\right\rangle\right]}_{\textbf{FTRL-Reg}}$$

where $\mathcal{A}_0$ is drawn from $D$ and is independent from the interaction between the learning and the environment. Recall that our algorithm is FTRL:

$$\boldsymbol{H}_t^{\mathcal{A}_0} = \underset{\boldsymbol{H}\in\mathcal{H}^{\mathcal{A}_0}}{\operatorname{argmin}}\left\{\sum_{s=1}^{t-1}\left\langle \boldsymbol{H}, \hat{\gamma}_s - \alpha_s\hat{\boldsymbol{\Sigma}}_s^{-1}\right\rangle + \frac{F(\boldsymbol{H})}{\eta_t}\right\}.$$

The **FTRL-Reg** term can be handled by the standard FTRL analysis (Lemma 5). In order to deal with the issue that $F$ can be unbounded on the boundary of $\mathcal{H}^{\mathcal{A}_0}$, we apply Lemma 5 with the regret comparator $\overline{\boldsymbol{U}}^{\mathcal{A}_0}$ defined as

$$\overline{\boldsymbol{U}}^{\mathcal{A}_0} = \left(1 - \frac{1}{T^2}\right)\boldsymbol{U}^{\mathcal{A}_0} + \frac{1}{T^2}\boldsymbol{H}_*^{\mathcal{A}_0}$$

where $\boldsymbol{H}_*^{\mathcal{A}_0} \triangleq \operatorname{argmin}_{\boldsymbol{H}\in\mathcal{H}^{\mathcal{A}_0}} F(\boldsymbol{H})$. Thus,

**FTRL-Reg**

$$\leq \mathbb{E}\left[\sum_{t=1}^{T}\left\langle \boldsymbol{H}_t^{\mathcal{A}_0} - \overline{\boldsymbol{U}}^{\mathcal{A}_0}, \hat{\gamma}_t - \alpha_t\hat{\boldsymbol{\Sigma}}_t^{-1}\right\rangle\right] + \mathbb{E}\left[\sum_{t=1}^{T}\left\langle \overline{\boldsymbol{U}}^{\mathcal{A}_0} - \boldsymbol{U}^{\mathcal{A}_0}, \hat{\gamma}_t - \alpha_t\hat{\boldsymbol{\Sigma}}_t^{-1}\right\rangle\right]$$

$$\leq \underbrace{\mathbb{E}\left[\frac{F(\overline{\boldsymbol{U}}^{\mathcal{A}_0}) - \min_{\boldsymbol{H}\in\mathcal{H}^{\mathcal{A}_0}} F(\boldsymbol{H})}{\eta_T}\right]}_{\textbf{Penalty}} + \underbrace{\mathbb{E}\left[\sum_{t=1}^{T}\max_{\boldsymbol{H}\in\mathcal{H}^{\mathcal{A}_0}}\left\langle \boldsymbol{H}_t^{\mathcal{A}_0} - \boldsymbol{H}, \hat{\gamma}_t\right\rangle - \frac{D(\boldsymbol{H}, \boldsymbol{H}_t^{\mathcal{A}_0})}{2\eta_t}\right]}_{\textbf{Stability-1}}$$

$$+ \underbrace{\mathbb{E}\left[\sum_{t=1}^{T}\max_{\boldsymbol{H}\in\mathcal{H}^{\mathcal{A}_0}}\left\langle \boldsymbol{H}_t^{\mathcal{A}_0} - \boldsymbol{H}, -\alpha_t\hat{\boldsymbol{\Sigma}}_t^{-1}\right\rangle - \frac{D(\boldsymbol{H}, \boldsymbol{H}_t^{\mathcal{A}_0})}{2\eta_t}\right]}_{\textbf{Stability-2}} + \underbrace{\mathbb{E}\left[\sum_{t=1}^{T}\left\langle \overline{\boldsymbol{U}}^{\mathcal{A}_0} - \boldsymbol{U}^{\mathcal{A}_0}, \hat{\gamma}_t - \alpha_t\hat{\boldsymbol{\Sigma}}_t^{-1}\right\rangle\right]}_{\textbf{Error}}$$

$$\tag{39}$$

In the rest of this section, we bound the following terms individually: **Bias**, **Bonus**, **Penalty**, **Stability-1**, **Stability-2**, **Error**.

For any $t = 2, \cdots, T$, let $\mathcal{E}_{t-1}$ be the event that the high-probability event in Lemma 17, Lemma 18, and Lemma 19 happens for all $1, \cdots, t-1$ and $\overline{\mathcal{E}_{t-1}}$ be the opposite event of $\mathcal{E}_{t-1}$(i.e. any of these three lemmas fails for any $1, \cdots, t-1$). We have $\mathcal{P}[\mathcal{E}_{t-1}] = 1 - \mathcal{O}(\delta)$ and $\mathcal{P}[\overline{\mathcal{E}_{t-1}}] = \mathcal{O}(\delta)$. Let $\mathbb{E}[\cdot \mid \mathcal{E}_{t-1}]$ be the conditional expectation that event $\mathcal{E}_{t-1}$ happens and let $\mathbb{E}_t^{\mathcal{E}} = \mathbb{E}[\cdot \mid \mathcal{F}_{t-1}, \mathcal{E}_{t-1}]$

### D.1  Bounding the Bias term

**Lemma 20.**

$$\textbf{Bias} = \mathbb{E}\left[\sum_{t=1}^{T}\left\langle \boldsymbol{H}_t^{\mathcal{A}_0} - \boldsymbol{U}^{\mathcal{A}_0}, \gamma_t - \hat{\gamma}_t\right\rangle\right] \leq \frac{1}{4}\sum_{t=1}^{T}\alpha_t\|x_t - u\|_{\hat{\Sigma}_t^{-1}}^2 + \mathcal{O}\left(\delta T^2 + \sum_{t=1}^{T}\frac{d^3\log(T/\delta)}{\alpha_t t}\right)$$

*Proof.* For any $t$, we have

$$\mathbb{E}_t^{\mathcal{E}}\left[\left\langle \boldsymbol{H}_t^{\mathcal{A}_0} - \boldsymbol{U}^{\mathcal{A}_0}, \gamma_t - \hat{\gamma}_t \right\rangle\right]$$

$$= \mathbb{E}_t^{\mathcal{E}}\left[\langle \boldsymbol{H}_t - \boldsymbol{U}, \gamma_t - \hat{\gamma}_t\rangle\right] \qquad\qquad\qquad\qquad \text{(taking expectation over } \mathcal{A}_0\text{)}$$

$$= \mathbb{E}_t^{\mathcal{E}}\left[\langle x_t - u, y_t - \hat{y}_t\rangle\right] \qquad\qquad\qquad\qquad\qquad \text{(by the definition of lifting)}$$

$$= \mathbb{E}_t^{\mathcal{E}}\left[(x_t - u)^\top \left(y_t - \hat{\Sigma}_t^{-1}(a_t - \hat{x}_t)a_t^\top y_t\right)\right] \qquad\qquad\qquad \text{(by the definition of } \hat{y}_t\text{)}$$

$$= \mathbb{E}_t^{\mathcal{E}}\left[(x_t - u)^\top \left(y_t - \hat{\Sigma}_t^{-1}(a_t - \hat{x}_t)(a_t - \hat{x}_t)^\top y_t\right)\right] - \mathbb{E}_t^{\mathcal{E}}\left[(x_t - u)^\top \hat{\Sigma}_t^{-1}(a_t - \hat{x}_t)\hat{x}_t^\top y_t\right]$$

$$= \mathbb{E}_t^{\mathcal{E}}\left[(x_t - u)^\top \left(I - \hat{\Sigma}_t^{-1}\mathbb{E}_{\mathcal{A}\sim\mathcal{D}}\mathbb{E}_{a_t\sim p_t^{\mathcal{A}}}\left[(a_t - \hat{x}_t)(a_t - \hat{x}_t)^\top\right]\right)y_t\right]$$

$$\qquad - \mathbb{E}_t^{\mathcal{E}}\left[(x_t - u)^\top \hat{\Sigma}_t^{-1}\left(\mathbb{E}_{\mathcal{A}\sim\mathcal{D}}\mathbb{E}_{a_t\sim p_t^{\mathcal{A}}}[a_t] - \hat{x}_t\right)\hat{x}_t^\top y_t\right] \text{ (taking expectation over } \mathcal{A}_t \text{ and } a_t\text{)}$$

$$= \mathbb{E}_t^{\mathcal{E}}\left[(x_t - u)^\top \hat{\Sigma}_t^{-1}\left(\hat{\Sigma}_t - H_t\right)y_t\right] - \mathbb{E}_t^{\mathcal{E}}\left[(x_t - u)^\top \hat{\Sigma}_t^{-1}(x_t - \hat{x}_t)\hat{x}_t^\top y_t\right]$$

$$\qquad\qquad\qquad\qquad\qquad\qquad\qquad\qquad\qquad\qquad \text{(by the definition of } H_t \text{ and } x_t\text{)}$$

$$\leq \mathbb{E}_t^{\mathcal{E}}\left[(x_t - u)^\top \hat{\Sigma}_t^{-1}\left(\hat{\Sigma}_t - H_t\right)y_t\right] + \mathbb{E}_t^{\mathcal{E}}\left[\left|(x_t - u)^\top \hat{\Sigma}_t^{-1}(x_t - \hat{x}_t)\right|\right] \qquad (|\hat{x}_t^\top y_t| \leq 1)$$

$$\leq \mathbb{E}_t^{\mathcal{E}}\left[\|x_t - u\|_{\hat{\Sigma}_t^{-1}}\left(\|(\hat{\Sigma}_t - H_t)y_t\|_{\hat{\Sigma}_t^{-1}} + \|x_t - \hat{x}_t\|_{\hat{\Sigma}_t^{-1}}\right)\right] \qquad \text{(Cauchy-Schwarz)}$$

$$\leq \mathcal{O}\left(\sqrt{\frac{d^3 \log(T/\delta)}{t}}\|x_t - u\|_{\hat{\Sigma}_t^{-1}}\right) \qquad\qquad \text{(Lemma 19 and Lemma 18 given } \mathcal{E}_{t-1}\text{)}$$

$$\leq \frac{\alpha_t}{4}\|x_t - u\|_{\hat{\Sigma}_t^{-1}}^2 + \mathcal{O}\left(\frac{d^3 \log(T/\delta)}{\alpha_t t}\right) \qquad\qquad\qquad\qquad \text{(AM-GM inequality)}$$

On the other hand, since $\hat{\Sigma}_t \succeq \frac{1}{t}I \succeq \frac{1}{T}I$, for any $t = 1, \cdots, T$,

$$\|\hat{y}_t\|_2 = \|\Sigma_t^{-1}(a_t - \hat{x}_t)a_t^\top y_t\|_2 \leq \|\Sigma_t^{-1}(a_t - \hat{x}_t)\|_2 \leq \mathcal{O}(T)$$

Thus, we have trivial bound

$$\mathbb{E}_t\left[\left\langle \boldsymbol{H}_t^{\mathcal{A}_0} - \boldsymbol{U}^{\mathcal{A}_0}, \gamma_t - \hat{\gamma}_t \right\rangle \,\Big|\, \overline{\mathcal{E}_{t-1}}\right] = \mathbb{E}_t\left[\langle \boldsymbol{H}_t - \boldsymbol{U}, \gamma_t - \hat{\gamma}_t\rangle \,|\, \overline{\mathcal{E}_{t-1}}\right] = \mathbb{E}_t\left[\langle x_t - u, y_t - \hat{y}_t\rangle \,|\, \overline{\mathcal{E}_{t-1}}\right] \leq \mathcal{O}(T)$$

Therefore, we have

$$\textbf{Bias} = \mathbb{E}\left[\sum_{t=1}^T \left\langle \boldsymbol{H}_t^{\mathcal{A}_0} - \boldsymbol{U}^{\mathcal{A}_0}, \gamma_t - \hat{\gamma}_t \right\rangle\right]$$

$$= \mathbb{E}\left[\sum_{t=1}^T \mathbb{E}_t\left[\left\langle \boldsymbol{H}_t^{\mathcal{A}_0} - \boldsymbol{U}^{\mathcal{A}_0}, \gamma_t - \hat{\gamma}_t \right\rangle\right]\right]$$

$$= \mathbb{E}\left[\sum_{t=1}^T \mathbb{E}_t\left[\left\langle \boldsymbol{H}_t^{\mathcal{A}_0} - \boldsymbol{U}^{\mathcal{A}_0}, \gamma_t - \hat{\gamma}_t \right\rangle \,\Big|\, \mathcal{E}_{t-1}\right]\mathbb{I}\{\mathcal{E}_{t-1}\}\right] + \mathbb{E}\left[\sum_{t=1}^T \mathbb{E}_t\left[\left\langle \boldsymbol{H}_t^{\mathcal{A}_0} - \boldsymbol{U}^{\mathcal{A}_0}, \gamma_t - \hat{\gamma}_t \right\rangle \,\Big|\, \overline{\mathcal{E}_{t-1}}\right]\mathbb{I}\{\overline{\mathcal{E}_{t-1}}\}\right]$$

$$\leq \frac{1}{4}\sum_{t=1}^T \alpha_t\|x_t - u\|_{\hat{\Sigma}_t^{-1}}^2 + \mathcal{O}\left(\sum_{t=1}^T \frac{d^3 \log(T/\delta)}{\alpha_t t} + \delta T^2\right)$$

$\qquad\qquad\qquad\qquad\qquad\qquad\qquad\qquad\qquad\qquad\qquad\qquad\qquad\qquad\qquad\qquad\qquad\qquad\qquad\qquad$ $\square$

## D.2 Bounding the Bonus term

We first prove the following useful technique lemma to bound the inner product of lifted matrices.

**Lemma 21.** *Let* $\boldsymbol{G} = \begin{bmatrix} G + gg^\top & g \\ g^\top & 1 \end{bmatrix}$, $\boldsymbol{H} = \begin{bmatrix} H + hh^\top & h \\ h^\top & 1 \end{bmatrix}$ *where $G$ and $H$ are positive semi-definite, and* $\boldsymbol{H}' = \boldsymbol{H} + vv^\top$ *where* $v = \begin{bmatrix} 0 \\ \sqrt{\beta} \end{bmatrix} \in \mathbb{R}^{d+1}$. *Then we have*

1. $\text{Tr}\left(\boldsymbol{H}^{-1}\boldsymbol{G}\right) = \text{Tr}(H^{-1}G) + \|g - h\|^2_{H^{-1}} + 1$

2. $\text{Tr}\left((\boldsymbol{H}')^{-1}\boldsymbol{G}\right) \geq \frac{1}{2\left(1 + \frac{\beta}{1+\beta}\|h\|^2_{H^{-1}}\right)}\|g - h\|^2_{H^{-1}} - \frac{\beta^2}{(1+\beta)^2}\|h\|^2_{H^{-1}}$

*Proof.* From Theorem 2.1 of Lu and Shiou [2002], for any block matrix $R = \begin{bmatrix} A & B \\ C & D \end{bmatrix}$ if $A$ is invertible and its Schur complement $S_A = D - CA^{-1}B$ is invertible, then

$$R^{-1} = \begin{bmatrix} A^{-1} + A^{-1}BS_A^{-1}CA^{-1} & -A^{-1}BS_A^{-1} \\ -S_A^{-1}CA & S_A^{-1} \end{bmatrix}$$

Using above equation, for the first equation, Since $(H + hh^\top)^{-1} = H^{-1} - \frac{H^{-1}hh^\top H^{-1}}{1 + h^\top H^{-1}h}$. The inverse Schur complement of $H + hh^\top$ is $1 + h^\top H^{-1}h$. Thus

$$\boldsymbol{H}^{-1} = \begin{bmatrix} (I + H^{-1}hh^\top)(H + hh^\top)^{-1} & -H^{-1}h \\ -h^\top H^{-1} & 1 + h^\top H^{-1}h \end{bmatrix} = \begin{bmatrix} H^{-1} & -H^{-1}h \\ -h^\top H^{-1} & 1 + h^\top H^{-1}h \end{bmatrix}$$

and

$$\begin{aligned}
\text{Tr}(\boldsymbol{H}^{-1}\boldsymbol{G}) &= \text{Tr}\left(H^{-1}G + H^{-1}gg^\top - H^{-1}hg^\top\right) - h^\top H^{-1}g + 1 + h^\top H^{-1}h \\
&= \text{Tr}\left(H^{-1}G\right) + g^\top H^{-1}g - 2g^\top H^{-1}h + h^\top H^{-1}h + 1 \\
&= \text{Tr}(H^{-1}G) + \|g - h\|^2_{H^{-1}} + 1.
\end{aligned}$$

For the second equation, observe that

$$\boldsymbol{H}' = \begin{bmatrix} H + hh^\top & h \\ h^\top & 1 + \beta \end{bmatrix} = (1 + \beta)\begin{bmatrix} \frac{1}{1+\beta}(H + hh^\top) & \frac{1}{1+\beta}h \\ \frac{1}{1+\beta}h^\top & 1 \end{bmatrix} = (1 + \beta)\begin{bmatrix} H' + h'h'^\top & h' \\ h'^\top & 1 \end{bmatrix}$$

where $h' = \frac{1}{1+\beta}h$ and $H' = \frac{1}{1+\beta}H + \left(\frac{1}{1+\beta} - \frac{1}{(1+\beta)^2}\right)hh^\top = \frac{1}{1+\beta}H + \frac{\beta}{(1+\beta)^2}hh^\top \succeq 0$.

Applying the first equality, we have

$$\text{Tr}((\boldsymbol{H}')^{-1}\boldsymbol{G}) = \frac{1}{1+\beta}\left(\text{Tr}((H')^{-1}G) + \|g - h'\|^2_{H'^{-1}} + 1\right) \geq \frac{1}{1+\beta}\|g - h'\|^2_{H'^{-1}}.$$

Below, we continue to lower bound this term. By the same formula above, we have

$$H'^{-1} = \left(\frac{1}{1+\beta}H + \frac{\beta}{(1+\beta)^2}hh^\top\right)^{-1} = (1+\beta)H^{-1} - \frac{\beta H^{-1}hh^\top H^{-1}}{1 + \frac{\beta}{1+\beta}h^\top H^{-1}h}.$$

Thus

$$\frac{1}{1+\beta}\|g - h'\|^2_{H'^{-1}}$$

$$\geq \frac{1}{2(1+\beta)}\|g - h\|^2_{H'^{-1}} - \frac{1}{1+\beta}\|h - h'\|^2_{H'^{-1}} \qquad \text{(using } \|a + b\|^2 \leq 2\|a\|^2 + 2\|b\|^2\text{)}$$

$$= \frac{1}{2}(g - h)^\top\left(H^{-1} - \frac{\frac{\beta}{1+\beta}H^{-1}hh^\top H^{-1}}{1 + \frac{\beta}{1+\beta}h^\top H^{-1}h}\right)(g - h) - (h - h')^\top\left(H^{-1} - \frac{\frac{\beta}{1+\beta}H^{-1}hh^\top H^{-1}}{1 + \frac{\beta}{1+\beta}h^\top H^{-1}h}\right)(h - h')$$

$$\geq \frac{1}{2}\|g - h\|^2_{H^{-1}} - \frac{\frac{\beta}{1+\beta}((g-h)^\top H^{-1}h)^2}{2\left(1 + \frac{\beta}{1+\beta}\|h\|^2_{H^{-1}}\right)} - \frac{\beta^2}{(1+\beta)^2}\|h\|^2_{H^{-1}} \qquad \text{(using } h - h' = \frac{\beta}{1+\beta}h\text{)}$$

$$\geq \frac{1}{2}\|g - h\|^2_{H^{-1}} - \frac{\frac{\beta}{1+\beta}\|h\|^2_{H^{-1}}}{2\left(1 + \frac{\beta}{1+\beta}\|h\|^2_{H^{-1}}\right)}\|g - h\|^2_{H^{-1}} - \frac{\beta^2}{(1+\beta)^2}\|h\|^2_{H^{-1}} \quad \text{(Cauchy-Schwarz)}$$

$$= \frac{1}{2\left(1 + \frac{\beta}{1+\beta}\|h\|^2_{H^{-1}}\right)}\|g - h\|^2_{H^{-1}} - \frac{\beta^2}{(1+\beta)^2}\|h\|^2_{H^{-1}}.$$

$\square$

Using Lemma 21, we are able to show Corollary 22 which bound part of the second term.

**Corollary 22.** $\mathrm{Tr}(\boldsymbol{U}\hat{\boldsymbol{\Sigma}}_t^{-1}) \geq \frac{1}{4}\|u - \hat{x}_t\|_{\hat{\Sigma}_t^{-1}}^2 - \frac{1}{4}$.

*Proof.* From Lemma 21, we have

$$\mathrm{Tr}(\boldsymbol{U}\hat{\boldsymbol{\Sigma}}_t^{-1}) \geq \frac{1}{2\left(1 + \frac{\beta_t}{1+\beta_t}\|\hat{x}_t\|_{\hat{\Sigma}_t^{-1}}^2\right)}\|u - \hat{x}_t\|_{\hat{\Sigma}_t^{-1}}^2 - \frac{\beta_t^2}{(1+\beta_t)^2}\|\hat{x}_t\|_{\Sigma_t^{-1}}^2.$$

Since $\hat{\Sigma}_t \succeq \beta_t I$, $\hat{\Sigma}_t^{-1} \preceq \frac{1}{\beta_t}I$. Since $\|\hat{x}_t\|_2 \leq 1$, we have $\|\hat{x}_t\|_{\hat{\Sigma}_t^{-1}}^2 \leq \frac{1}{\beta_t}$. Then

$$\mathrm{Tr}(\boldsymbol{U}\hat{\boldsymbol{\Sigma}}_t^{-1}) \geq \frac{1}{2\left(1 + \frac{1}{1+\beta_t}\right)}\|u - \hat{x}_t\|_{\hat{\Sigma}_t^{-1}}^2 - \frac{\beta_t}{(1+\beta_t)^2}$$

$$\geq \frac{1}{4}\|u - \hat{x}_t\|_{\hat{\Sigma}_t^{-1}}^2 - \frac{\beta_t}{(2\sqrt{\beta_t})^2} \qquad (\beta_t \geq 0)$$

$$= \frac{1}{4}\|u - \hat{x}_t\|_{\hat{\Sigma}_t^{-1}}^2 - \frac{1}{4}.$$

$\square$

**Lemma 23.**

$$\mathbf{Bonus} = \mathbb{E}\left[\sum_{t=1}^T \left\langle \boldsymbol{H}_t^{\mathcal{A}_0} - \boldsymbol{U}^{\mathcal{A}_0}, \alpha_t\hat{\boldsymbol{\Sigma}}_t^{-1}\right\rangle\right]$$

$$\leq 2(d+2)\sum_{t=1}^T \alpha_t - \frac{1}{4}\sum_{t=1}^T \alpha_t\|u - x_t\|_{\hat{\Sigma}_t^{-1}}^2 + \mathcal{O}\left(\sum_{t=1}^T \frac{d^3\alpha_t \log(T/\delta)}{t} + \delta T\sum_{t=1}^T \alpha_t\right).$$

*Proof.* For any $t$, we have

$$\mathbb{E}_t^{\mathcal{E}}\left[\left\langle \boldsymbol{H}_t^{\mathcal{A}_0} - \boldsymbol{U}^{\mathcal{A}_0}, \alpha_t\hat{\boldsymbol{\Sigma}}_t^{-1}\right\rangle\right]$$

$$= \mathbb{E}_t^{\mathcal{E}}\left[\mathrm{Tr}\left(\alpha_t\left(\boldsymbol{H}_t - \boldsymbol{U}\right)\hat{\boldsymbol{\Sigma}}_t^{-1}\right)\right] \qquad \text{(taking expectation over } \mathcal{A}_0)$$

$$= \mathbb{E}_t^{\mathcal{E}}\left[\alpha_t \mathrm{Tr}\left(\boldsymbol{H}_t\hat{\boldsymbol{\Sigma}}_t^{-1}\right) - \alpha_t \mathrm{Tr}\left(\boldsymbol{U}\hat{\boldsymbol{\Sigma}}_t^{-1}\right)\right]$$

$$\leq \alpha_t \mathrm{Tr}\left(\mathbb{E}_t^{\mathcal{E}}\left[\boldsymbol{H}_t\right]\hat{\boldsymbol{\Sigma}}_t^{-1}\right) - \mathbb{E}_t^{\mathcal{E}}\left[\frac{\alpha_t}{4}\|u - \hat{x}_t\|_{\hat{\Sigma}_t^{-1}}^2\right] + \frac{1}{4}\alpha_t \qquad \text{(Corollary 22)}$$

$$\leq 2\alpha_t(d+2) - \mathbb{E}_t^{\mathcal{E}}\left[\frac{\alpha_t}{4}\|u - \hat{x}_t\|_{\hat{\Sigma}_t^{-1}}^2\right]$$

$$\leq 2\alpha_t(d+2) - \mathbb{E}_t^{\mathcal{E}}\left[\frac{\alpha_t}{4}\|u - x_t\|_{\hat{\Sigma}_t^{-1}}^2 - \frac{\alpha_t}{4}\|\hat{x}_t - x_t\|_{\hat{\Sigma}_t^{-1}}^2\right]$$

$$\leq 2\alpha_t(d+2) - \frac{\alpha_t}{4}\|u - x_t\|_{\hat{\Sigma}_t^{-1}}^2 + \mathcal{O}\left(\frac{d^3\alpha_t \log(T/\delta)}{t}\right) \qquad \text{(Lemma 18)}$$

On the other hand, since $\hat{\Sigma}_t \succeq \frac{1}{t}I \succeq \frac{1}{T}I$, we have trivial bound

$$\mathbb{E}_t\left[\left\langle \boldsymbol{H}_t^{\mathcal{A}_0} - \boldsymbol{U}^{\mathcal{A}_0}, \alpha_t\hat{\boldsymbol{\Sigma}}_t^{-1}\right\rangle \mid \overline{\mathcal{E}_{t-1}}\right] \leq \mathcal{O}(\alpha_t T)$$

Therefore, we have

$$\textbf{Bonus} = \mathbb{E}\left[\sum_{t=1}^{T}\left\langle \boldsymbol{H}_t^{\mathcal{A}_0} - \boldsymbol{U}^{\mathcal{A}_0}, \alpha_t\hat{\boldsymbol{\Sigma}}_t^{-1}\right\rangle\right]$$

$$= \mathbb{E}\left[\sum_{t=1}^{T}\mathbb{E}_t\left[\left\langle \boldsymbol{H}_t^{\mathcal{A}_0} - \boldsymbol{U}^{\mathcal{A}_0}, \alpha_t\hat{\boldsymbol{\Sigma}}_t^{-1}\right\rangle\right]\right]$$

$$= \mathbb{E}\left[\sum_{t=1}^{T}\mathbb{E}_t\left[\left\langle \boldsymbol{H}_t^{\mathcal{A}_0} - \boldsymbol{U}^{\mathcal{A}_0}, \alpha_t\hat{\boldsymbol{\Sigma}}_t^{-1}\right\rangle \Big| \mathcal{E}_{t-1}\right]\mathbb{I}\{\mathcal{E}_{t-1}\}\right] + \mathbb{E}\left[\sum_{t=1}^{T}\mathbb{E}_t\left[\left\langle \boldsymbol{H}_t^{\mathcal{A}_0} - \boldsymbol{U}^{\mathcal{A}_0}, \alpha_t\hat{\boldsymbol{\Sigma}}_t^{-1}\right\rangle \Big| \overline{\mathcal{E}_{t-1}}\right]\mathbb{I}\{\overline{\mathcal{E}_{t-1}}\}\right]$$

$$\le 2(d+2)\sum_{t=1}^{T}\alpha_t - \frac{(1-\delta)}{4}\sum_{t=1}^{T}\alpha_t\|u - x_t\|_{\hat{\Sigma}_t^{-1}}^2 + \mathcal{O}\left(\sum_{t=1}^{T}\frac{d^3\alpha_t\log(T/\delta)}{t} + \delta T\sum_{t=1}^{T}\alpha_t\right)$$

$$\le 2(d+2)\sum_{t=1}^{T}\alpha_t - \frac{1}{4}\sum_{t=1}^{T}\alpha_t\|u - x_t\|_{\hat{\Sigma}_t^{-1}}^2 + \mathcal{O}\left(\sum_{t=1}^{T}\frac{d^3\alpha_t\log(T/\delta)}{t} + \delta T\sum_{t=1}^{T}\alpha_t\right)$$

$\square$

## D.3 Bounding the Penalty term

**Lemma 24.** $\overline{\boldsymbol{U}}^{\mathcal{A}_0}$, we have

$$\frac{F(\overline{\boldsymbol{U}}^{\mathcal{A}_0}) - \min_{\boldsymbol{H}\in\mathcal{H}^{\mathcal{A}_0}} F(\boldsymbol{H})}{\eta_T} \le \frac{2d\log(T)}{\eta_T}$$

*Proof.* Since $\overline{\boldsymbol{U}}^{\mathcal{A}_0} = \left(1 - \frac{1}{T^2}\right)\boldsymbol{U}^{\mathcal{A}_0} + \frac{1}{T^2}\boldsymbol{H}_*^{\mathcal{A}_0}$, we have $\overline{\boldsymbol{U}}^{\mathcal{A}_0} \succeq \frac{1}{T^2}\boldsymbol{H}_*^{\mathcal{A}_0}$. Then

$$\frac{F(\overline{\boldsymbol{U}}^{\mathcal{A}_0}) - \min_{\boldsymbol{H}\in\mathcal{H}^{\mathcal{A}_0}} F(\boldsymbol{H})}{\eta_T} = \frac{1}{\eta_T}\log\frac{\det(\boldsymbol{H}_*^{\mathcal{A}_0})}{\det(\overline{\boldsymbol{U}}^{\mathcal{A}_0})} \le \frac{2d\log(T)}{\eta_T}.$$

$\square$

## D.4 Bounding the Stability-1 term

[Zimmert and Lattimore [2022]] gave a useful identity to bound the Bregman divergence. We restate it in [Lemma 25] for completeness.

**Lemma 25.** *Let* $\boldsymbol{G} = \begin{bmatrix} G + gg^\top & g \\ g^\top & 1 \end{bmatrix}$ *and* $\boldsymbol{H} = \begin{bmatrix} H + hh^\top & h \\ h^\top & 1 \end{bmatrix}$, *we have*

$$D(\boldsymbol{G}, \boldsymbol{H}) = D(G, H) + \|g - h\|_{H^{-1}}^2 \ge \|g - h\|_{H^{-1}}^2$$

*Proof.*

$$D(\boldsymbol{G}, \boldsymbol{H}) = F(\boldsymbol{G}) - F(\boldsymbol{H}) - \langle \nabla F(\boldsymbol{H}), \boldsymbol{G} - \boldsymbol{H}\rangle$$

$$= \log\left(\frac{\det(\boldsymbol{H})}{\det(\boldsymbol{G})}\right) + \text{Tr}(\boldsymbol{H}^{-1}(\boldsymbol{G} - \boldsymbol{H}))$$

$$= \log\left(\frac{\det(\boldsymbol{H})}{\det(\boldsymbol{G})}\right) + \text{Tr}(\boldsymbol{H}^{-1}\boldsymbol{G}) - d - 1$$

$$= \log\left(\frac{\det(\boldsymbol{H})}{\det(\boldsymbol{G})}\right) + \text{Tr}(\boldsymbol{H}^{-1}\boldsymbol{G}) - d - 1$$

$$= \log\left(\frac{\det(H)}{\det(G)}\right) + \text{Tr}(H^{-1}G) + \|g - h\|_{H^{-1}}^2 - d \qquad \text{(Lemma 21)}$$

$$= D(G, H) + \|g - h\|_{H^{-1}}^2$$

$$\ge \|g - h\|_{H^{-1}}^2$$

$\square$

**Lemma 26.** *For any $\boldsymbol{H} \in \mathcal{H}^{\mathcal{A}_0}$, we have*

$$\textbf{Stability-1} = \mathbb{E}\left[\sum_{t=1}^{T}\left\langle \boldsymbol{H}_t^{\mathcal{A}_0} - \boldsymbol{H}, \hat{\gamma}_t\right\rangle - \frac{D(\boldsymbol{H}, \boldsymbol{H}_t^{\mathcal{A}_0})}{2\eta_t}\right] \leq 2d\sum_{t=1}^{T}\eta_t + \mathcal{O}(\delta T^2)$$

*Proof.* Recall that $\boldsymbol{H}_t^{\mathcal{A}_0} = \widehat{\mathrm{Cov}}(p_t^{\mathcal{A}_0})$ and $\widehat{\mathrm{Cov}}(p) = \begin{bmatrix} \mathrm{Cov}(p) + \mu(p)\mu(p)^\top & \mu(p) \\ \mu(p)^\top & 1 \end{bmatrix}$, we have

$$\left\langle \boldsymbol{H}_t^{\mathcal{A}_0} - \boldsymbol{H}, \hat{\gamma}_t\right\rangle - \frac{D(\boldsymbol{H}, \boldsymbol{H}_t^{\mathcal{A}_0})}{2\eta_t} \leq \left\langle x_t^{\mathcal{A}_0} - \mu(p), \hat{y}_t\right\rangle - \frac{\|\mu(p) - x_t^{\mathcal{A}_0}\|^2_{\mathrm{Cov}(p_t^{\mathcal{A}_0})^{-1}}}{2\eta_t} \quad \text{(Lemma 25)}$$

$$\leq \|x_t^{\mathcal{A}_0} - \mu(p)\|_{\mathrm{Cov}(p_t^{\mathcal{A}_0})^{-1}}\|\hat{y}_t\|_{\mathrm{Cov}(p_t^{\mathcal{A}_0})} - \frac{\|\mu(p) - x_t^{\mathcal{A}_0}\|^2_{\mathrm{Cov}(p_t^{\mathcal{A}_0})^{-1}}}{2\eta_t}$$

$$\leq \frac{\eta_t}{2}\|\hat{y}_t\|^2_{\mathrm{Cov}(p_t^{\mathcal{A}_0})} \quad \text{(AM-GM inequality)}$$

$$= \frac{\eta_t}{2}\|\hat{\Sigma}_t^{-1}(a_t - \hat{x}_t)\ell_t\|^2_{\mathrm{Cov}(p_t^{\mathcal{A}_0})}$$

$$\leq \frac{\eta_t}{2}(a_t - \hat{x}_t)^\top \hat{\Sigma}_t^{-1} \mathrm{Cov}(p_t^{\mathcal{A}_0})\hat{\Sigma}_t^{-1}(a_t - \hat{x}_t) \quad (|\ell_t| \leq 1)$$

$$= \frac{\eta_t}{2}\mathrm{Tr}\left((a_t - \hat{x}_t)(a_t - \hat{x}_t)^\top \hat{\Sigma}_t^{-1}\mathrm{Cov}(p_t^{\mathcal{A}_0})\hat{\Sigma}_t^{-1}\right)$$

Since $\mathbb{E}_{\mathcal{A} \sim \mathcal{D}}\mathbb{E}_{a \sim p^{\mathcal{A}}}\left[(a - \hat{x}_t)(a - \hat{x}_t)^\top\right] = H_t$, taking expectations over $\mathcal{A}_t$, $a_t$ and $\mathcal{A}_0$ conditioned on $\mathcal{E}_{t-1}$, we have

$$\mathbb{E}_t^{\mathcal{E}}\left[\left\langle \boldsymbol{H}_t^{\mathcal{A}_0} - \boldsymbol{H}, \hat{\gamma}_t\right\rangle - \frac{D(\boldsymbol{H}, \boldsymbol{H}_t^{\mathcal{A}_0})}{2\eta_t}\right] \leq \mathbb{E}_t^{\mathcal{E}}\left[\frac{\eta_t}{2}\mathrm{Tr}\left((a_t - \hat{x}_t)(a_t - \hat{x}_t)^\top \hat{\Sigma}_t^{-1}\mathrm{Cov}(p_t^{\mathcal{A}_0})\hat{\Sigma}_t^{-1}\right)\right]$$

$$= \mathbb{E}_t^{\mathcal{E}}\left[\frac{\eta_t}{2}\mathrm{Tr}\left(H_t\hat{\Sigma}_t^{-1}\mathbb{E}_{\mathcal{A}_0 \sim D}\left[\mathrm{Cov}(p_t^{\mathcal{A}_0})\right]\hat{\Sigma}_t^{-1}\right)\right].$$

Notice that given $\mathcal{E}_{t-1}$,

$$\hat{\Sigma}_t \succeq \frac{1}{2}H_t = \frac{1}{2}\mathbb{E}_{\mathcal{A} \sim D}[\mathrm{Cov}(p_t^{\mathcal{A}})] + \frac{1}{2}(\hat{x}_t - x_t)(\hat{x}_t - x_t)^\top \succeq \frac{1}{2}\mathbb{E}_{\mathcal{A} \sim D}[\mathrm{Cov}(p_t^{\mathcal{A}})]$$

Hence we continue to upper bound the last expression by

$$\mathbb{E}_t^{\mathcal{E}}\left[\eta_t\mathrm{Tr}\left(H_t\hat{\Sigma}_t^{-1}\hat{\Sigma}_t\hat{\Sigma}_t^{-1}\right)\right] \leq \mathbb{E}_t^{\mathcal{E}}\left[\eta_t\mathrm{Tr}\left(H_t\hat{\Sigma}_t^{-1}\right)\right] \leq 2\eta_t d.$$

On the other hand, since $\hat{\Sigma}_t \succeq \frac{1}{t}I \succeq \frac{1}{T}I$, we have trivial bound

$$\mathbb{E}_t\left[\left\langle \boldsymbol{H}_t^{\mathcal{A}_0} - \boldsymbol{H}, \hat{\gamma}_t\right\rangle - \frac{D(\boldsymbol{H}, \boldsymbol{H}_t^{\mathcal{A}_0})}{2\eta_t} \,\middle|\, \overline{\mathcal{E}_{t-1}}\right] \leq \mathcal{O}(T)$$

Combining everything, we get

$$\textbf{Stability-1} = \mathbb{E}\left[\sum_{t=1}^{T}\left\langle \boldsymbol{H}_t^{\mathcal{A}_0} - \boldsymbol{H}, \hat{\gamma}_t\right\rangle - \frac{D(\boldsymbol{H}, \boldsymbol{H}_t^{\mathcal{A}_0})}{2\eta_t}\right]$$

$$= \mathbb{E}\left[\sum_{t=1}^{T}\mathbb{E}_t\left[\left\langle \boldsymbol{H}_t^{\mathcal{A}_0} - \boldsymbol{H}, \hat{\gamma}_t\right\rangle - \frac{D(\boldsymbol{H}, \boldsymbol{H}_t^{\mathcal{A}_0})}{2\eta_t}\right]\right]$$

$$= \mathbb{E}\left[\sum_{t=1}^{T}\mathbb{E}_t\left[\left\langle \boldsymbol{H}_t^{\mathcal{A}_0} - \boldsymbol{H}, \hat{\gamma}_t\right\rangle - \frac{D(\boldsymbol{H}, \boldsymbol{H}_t^{\mathcal{A}_0})}{2\eta_t} \,\middle|\, \mathcal{E}_{t-1}\right]\mathbb{I}\{\mathcal{E}_{t-1}\}\right]$$

$$+ \mathbb{E}\left[\sum_{t=1}^{T}\mathbb{E}_t\left[\left\langle \boldsymbol{H}_t^{\mathcal{A}_0} - \boldsymbol{H}, \hat{\gamma}_t\right\rangle - \frac{D(\boldsymbol{H}, \boldsymbol{H}_t^{\mathcal{A}_0})}{2\eta_t} \,\middle|\, \overline{\mathcal{E}_{t-1}}\right]\mathbb{I}\{\overline{\mathcal{E}_{t-1}}\}\right]$$

$$\leq 2d\sum_{t=1}^{T}\eta_t + \mathcal{O}(\delta T^2).$$

$\square$

## D.5 Bounding the Stability-2 term

Note that [Lemma 8](#) does not require matrix $A, B$ to be positive semi-definite. We will use it to prove the following lemma based on Lemma 34 in [Dann et al. [2023b]](#).

**Lemma 27.** *If $\eta_t \alpha_t \leq \frac{1}{64t}$, then*

$$\textbf{Stability-2} = \mathbb{E}\left[\sum_{t=1}^T \max_{\boldsymbol{H} \in \mathcal{H}^{\mathcal{A}_0}} \left\langle \boldsymbol{H}_t^{\mathcal{A}_0} - \boldsymbol{H}, -\alpha_t \hat{\boldsymbol{\Sigma}}_t^{-1} \right\rangle - \frac{D(\boldsymbol{H}, \boldsymbol{H}_t^{\mathcal{A}_0})}{2\eta_t}\right] \leq d\sum_{t=1}^T \alpha_t + \mathcal{O}\left(\delta T^2\right)$$

*Proof.* We first show that $\max_{\boldsymbol{H} \in \mathcal{H}^{\mathcal{A}_0}} \left\langle \boldsymbol{H}_t^{\mathcal{A}_0} - \boldsymbol{H}, -\alpha_t \hat{\boldsymbol{\Sigma}}_t^{-1} \right\rangle - \frac{D(\boldsymbol{H}, \boldsymbol{H}_t^{\mathcal{A}_0})}{2\eta_t} \leq \frac{\alpha_t}{2}\|\hat{\boldsymbol{\Sigma}}_t^{-1}\|_{\nabla^{-2}F(\boldsymbol{H}_t^{\mathcal{A}_0})}$.

Define

$$G(\boldsymbol{H}) = \left\langle \boldsymbol{H}_t^{\mathcal{A}_0} - \boldsymbol{H}, -\alpha_t \hat{\boldsymbol{\Sigma}}_t^{-1} \right\rangle - \frac{D(\boldsymbol{H}, \boldsymbol{H}_t^{\mathcal{A}_0})}{2\eta_t}$$

and $\lambda = \|\alpha_t \hat{\boldsymbol{\Sigma}}_t^{-1}\|_{\nabla^{-2}F(\boldsymbol{H}_t^{\mathcal{A}_0})}$. Since $\hat{\boldsymbol{\Sigma}}_t \succeq \frac{1}{t}I$, $\boldsymbol{H}_t^{\mathcal{A}_0} \preceq 2I$, $\eta_t \alpha_t \leq \frac{1}{64t}$, we have

$$\eta_t \lambda = \eta_t \|\alpha_t \hat{\boldsymbol{\Sigma}}_t^{-1}\|_{\nabla^{-2}F(\boldsymbol{H}_t^{\mathcal{A}_0})} = \eta_t \alpha_t \sqrt{\text{Tr}(\boldsymbol{H}_t^{\mathcal{A}_0}\hat{\boldsymbol{\Sigma}}_t^{-1}\boldsymbol{H}_t^{\mathcal{A}_0}\hat{\boldsymbol{\Sigma}}_t^{-1})} \leq 2\eta_t \alpha_t t \leq \frac{1}{32}.$$

Let $\boldsymbol{H}'$ be the maximizer of $G$. Since $G(\boldsymbol{H}_t^{\mathcal{A}_0}) = 0$, we have $G(\boldsymbol{H}') \geq 0$. It suffices to show $\|\boldsymbol{H}' - \boldsymbol{H}_t^{\mathcal{A}_0}\|_{\nabla^2 F(\boldsymbol{H}_t^{\mathcal{A}_0})} \leq 16\eta_t\lambda$ because from [Lemma 8](#), it leads to

$$G(\boldsymbol{H}') \leq \|\boldsymbol{H}_t^{\mathcal{A}_0} - \boldsymbol{H}'\|_{\nabla^2 F(\boldsymbol{H}_t^{\mathcal{A}_0})}\|\alpha_t\hat{\boldsymbol{\Sigma}}_t^{-1}\|_{\nabla^{-2}F(\boldsymbol{H}_t^{\mathcal{A}_0})} \leq 16\eta_t\lambda\alpha_t\|\hat{\boldsymbol{\Sigma}}_t^{-1}\|_{\nabla^{-2}F(\boldsymbol{H}_t^{\mathcal{A}_0})} = \frac{\alpha_t}{2}\|\hat{\boldsymbol{\Sigma}}_t^{-1}\|_{\nabla^{-2}F(\boldsymbol{H}_t^{\mathcal{A}_0})}$$

To show $\|\boldsymbol{H}' - \boldsymbol{H}_t^{\mathcal{A}_0}\|_{\nabla^2 F(\boldsymbol{H}_t^{\mathcal{A}_0})} \leq 16\eta_t\lambda$, it suffices to show that for all $\boldsymbol{U}$ such that $\|\boldsymbol{U} - \boldsymbol{H}_t^{\mathcal{A}_0}\|_{\nabla^2 F(\boldsymbol{H}_t^{\mathcal{A}_0})} = 16\eta_t\lambda$, $G(\boldsymbol{U}) \leq 0$. This is because given this condition, if $\|\boldsymbol{H}' - \boldsymbol{H}_t^{\mathcal{A}_0}\|_{\nabla^2 F(\boldsymbol{H}_t^{\mathcal{A}_0})} > 16\eta_t\lambda$, then there is a $\boldsymbol{U}$ in the line segment between $\boldsymbol{H}_t^{\mathcal{A}_0}$ and $\boldsymbol{H}'$ such that $\|\boldsymbol{U} - \boldsymbol{H}_t^{\mathcal{A}_0}\|_{\nabla^2 F(\boldsymbol{H}_t^{\mathcal{A}_0})} = 16\eta_t\lambda$. From the condition, $G(\boldsymbol{U}) \leq 0 \leq \min\{G(\boldsymbol{H}_t^{\mathcal{A}_0}), G(\boldsymbol{H}')\}$ which contradicts to the strictly concave of $G$.

Now consider any $\boldsymbol{U}$ such that $\|\boldsymbol{U} - \boldsymbol{H}_t^{\mathcal{A}_0}\|_{\nabla^2 F(\boldsymbol{H}_t^{\mathcal{A}_0})} = 16\eta_t\lambda$. By Taylor expansion, there exists $\boldsymbol{U}'$ in the line segment between $\boldsymbol{U}$ and $\boldsymbol{H}_t^{\mathcal{A}_0}$ such that

$$G(\boldsymbol{U}) \leq \|\boldsymbol{U} - \boldsymbol{H}_t^{\mathcal{A}_0}\|_{\nabla^2 F(\boldsymbol{H}_t^{\mathcal{A}_0})}\|\alpha_t\hat{\boldsymbol{\Sigma}}_t^{-1}\|_{\nabla^{-2}F(\boldsymbol{H}_t^{\mathcal{A}_0})} - \frac{1}{4\eta_t}\|\boldsymbol{U} - \boldsymbol{H}_t^{\mathcal{A}_0}\|_{\nabla^2 F(\boldsymbol{U}')}^2$$

We have $\|\boldsymbol{U}' - \boldsymbol{H}_t^{\mathcal{A}_0}\|_{\nabla^2 F(\boldsymbol{H}_t^{\mathcal{A}_0})} \leq \|\boldsymbol{U} - \boldsymbol{H}_t^{\mathcal{A}_0}\|_{\nabla^2 F(\boldsymbol{H}_t^{\mathcal{A}_0})} = 16\eta_t\lambda \leq \frac{1}{2}$. From the Equation 2.2 in page 23 of [Nemirovski [2004]](#) (also appear in Eq.(5) of [Abernethy et al. [2009]](#)) and $\log \det$ is a self-concordant function, we have $\|\boldsymbol{U} - \boldsymbol{H}_t^{\mathcal{A}_0}\|_{\nabla^2 F(\boldsymbol{U}')}^2 \geq \frac{1}{4}\|\boldsymbol{U} - \boldsymbol{H}_t^{\mathcal{A}_0}\|_{\nabla^2 F(\boldsymbol{H}_t^{\mathcal{A}_0})}^2$. Thus, we have

$$G(\boldsymbol{U}) \leq \|\boldsymbol{U} - \boldsymbol{H}_t^{\mathcal{A}_0}\|_{\nabla^2 F(\boldsymbol{H}_t^{\mathcal{A}_0})}\|\alpha_t\hat{\boldsymbol{\Sigma}}_t^{-1}\|_{\nabla^{-2}F(\boldsymbol{H}_t^{\mathcal{A}_0})} - \frac{1}{16\eta_t}\|\boldsymbol{U} - \boldsymbol{H}_t^{\mathcal{A}_0}\|_{(\boldsymbol{H}_t^{\mathcal{A}_0})^{-1}}^2 = 16\eta_t\lambda^2 - \frac{(16\eta_t\lambda)^2}{16\eta_t} = 0$$

We have $\|\hat{\boldsymbol{\Sigma}}_t^{-1}\|_{\nabla^{-2}F(\boldsymbol{H}_t^{\mathcal{A}_0})} = \sqrt{\text{Tr}(\boldsymbol{H}_t^{\mathcal{A}_0}\hat{\boldsymbol{\Sigma}}_t^{-1}\boldsymbol{H}_t^{\mathcal{A}_0}\hat{\boldsymbol{\Sigma}}_t^{-1})} = \sqrt{\text{Tr}((\boldsymbol{H}_t^{\mathcal{A}_0}\hat{\boldsymbol{\Sigma}}_t^{-1})^2)}$. Observe the following two facts: 1) all eigenvalues of $\boldsymbol{H}_t^{\mathcal{A}_0}\hat{\boldsymbol{\Sigma}}_t^{-1}$ are non-negative since $\boldsymbol{H}_t^{\mathcal{A}_0}$ and $\hat{\boldsymbol{\Sigma}}_t^{-1}$ are both positive semi-definite, 2) for a square matrix $A$ with all non-negative eigenvalues, $\text{Tr}(A^2) \leq \text{Tr}(A)^2$ because $\text{Tr}(A^2) = \sum_i \lambda_i(A^2) = \sum_i \lambda_i(A)^2 \leq (\sum_i \lambda_i(A))^2$. We have

$$\sqrt{\text{Tr}((\boldsymbol{H}_t^{\mathcal{A}_0}\hat{\boldsymbol{\Sigma}}_t^{-1})^2)} \leq \text{Tr}(\boldsymbol{H}_t^{\mathcal{A}_0}\hat{\boldsymbol{\Sigma}}_t^{-1}).$$

This allows us to conclude

$$\mathbb{E}_t^{\mathcal{E}}\left[\frac{\alpha_t}{2}\|\hat{\boldsymbol{\Sigma}}_t^{-1}\|_{\nabla^{-2}F(\boldsymbol{H}_t^{\mathcal{A}_0})}\right] \leq \frac{\alpha_t}{2}\mathbb{E}_t^{\mathcal{E}}\left[\text{Tr}(\boldsymbol{H}_t^{\mathcal{A}_0}\hat{\boldsymbol{\Sigma}}_t^{-1})\right] \leq \alpha_t d$$

where we use that $\hat{\boldsymbol{\Sigma}}_t \succeq \frac{1}{2}\mathbb{E}_{\mathcal{A}_0 \sim D}[\boldsymbol{H}_t^{\mathcal{A}_0}]$ given $\mathcal{E}_{t-1}$.

On the other hand, since $\hat{\boldsymbol{\Sigma}}_t \succeq \frac{1}{t}\boldsymbol{I} \succeq \frac{1}{T}\boldsymbol{I}$, for any $t = 1, \cdots, T$, we have trivial bound

$$\mathbb{E}_t\left[\max_{\boldsymbol{H} \in \mathcal{H}^{\mathcal{A}_0}}\left\langle \boldsymbol{H}_t^{\mathcal{A}_0} - \boldsymbol{H}, -\alpha_t\hat{\boldsymbol{\Sigma}}_t^{-1}\right\rangle - \frac{D(\boldsymbol{H}, \boldsymbol{H}_t^{\mathcal{A}_0})}{2\eta_t} \,\bigg|\, \overline{\mathcal{E}_{t-1}}\right] \leq \mathcal{O}(T)$$

Overall,

$$\begin{aligned}
\textbf{Stability-2} &= \mathbb{E}\left[\sum_{t=1}^T \max_{\boldsymbol{H} \in \mathcal{H}^{\mathcal{A}_0}}\left\langle \boldsymbol{H}_t^{\mathcal{A}_0} - \boldsymbol{H}, -\alpha_t\hat{\boldsymbol{\Sigma}}_t^{-1}\right\rangle - \frac{D(\boldsymbol{H}, \boldsymbol{H}_t^{\mathcal{A}_0})}{2\eta_t}\right] \\
&\leq \mathbb{E}\left[\sum_{t=1}^T \mathbb{E}_t\left[\max_{\boldsymbol{H} \in \mathcal{H}^{\mathcal{A}_0}}\left\langle \boldsymbol{H}_t^{\mathcal{A}_0} - \boldsymbol{H}, -\alpha_t\hat{\boldsymbol{\Sigma}}_t^{-1}\right\rangle - \frac{D(\boldsymbol{H}, \boldsymbol{H}_t^{\mathcal{A}_0})}{2\eta_t}\right]\right] \\
&= \mathbb{E}\left[\sum_{t=1}^T \mathbb{E}_t\left[\max_{\boldsymbol{H} \in \mathcal{H}^{\mathcal{A}_0}}\left\langle \boldsymbol{H}_t^{\mathcal{A}_0} - \boldsymbol{H}, \hat{\gamma}_t\right\rangle - \frac{D(\boldsymbol{H}, \boldsymbol{H}_t^{\mathcal{A}_0})}{2\eta_t}\,\bigg|\,\mathcal{E}_{t-1}\right]\mathbb{I}\{\mathcal{E}_{t-1}\}\right] \\
&\quad + \mathbb{E}\left[\sum_{t=1}^T \mathbb{E}_t\left[\max_{\boldsymbol{H} \in \mathcal{H}^{\mathcal{A}_0}}\left\langle \boldsymbol{H}_t^{\mathcal{A}_0} - \boldsymbol{H}, \hat{\gamma}_t\right\rangle - \frac{D(\boldsymbol{H}, \boldsymbol{H}_t^{\mathcal{A}_0})}{2\eta_t}\,\bigg|\,\overline{\mathcal{E}_{t-1}}\right]\mathbb{I}\{\overline{\mathcal{E}_{t-1}}\}\right] \\
&\leq d\sum_{t=1}^T \alpha_t + \mathcal{O}\left(\delta T^2\right).
\end{aligned}$$

$\square$

## D.6 Bounding the Error term

**Lemma 28.**

$$\textbf{Error} = \mathbb{E}\left[\sum_{t=1}^T \left\langle \overline{\boldsymbol{U}}^{\mathcal{A}_0} - \boldsymbol{U}^{\mathcal{A}_0}, \hat{\gamma}_t - \alpha_t\hat{\boldsymbol{\Sigma}}_t^{-1}\right\rangle\right] \leq \mathcal{O}(1).$$

*Proof.* Since $\overline{\boldsymbol{U}}^{\mathcal{A}_0} = \left(1 - \frac{1}{T^2}\right)\boldsymbol{U}^{\mathcal{A}_0} + \frac{1}{T^2}\boldsymbol{H}_*^{\mathcal{A}_0}$, and $\hat{\Sigma}_t \succeq \frac{1}{T}I, \hat{\boldsymbol{\Sigma}}_t \succeq \frac{1}{T}\boldsymbol{I}$ we have

$$\begin{aligned}
\textbf{Error} &= \mathbb{E}\left[\sum_{t=1}^T \left\langle \overline{\boldsymbol{U}}^{\mathcal{A}_0} - \boldsymbol{U}^{\mathcal{A}_0}, \hat{\gamma}_t - \alpha_t\hat{\boldsymbol{\Sigma}}_t^{-1}\right\rangle\right] \\
&= \mathbb{E}\left[\frac{1}{T^2}\sum_{t=1}^T \left\langle -\boldsymbol{U}^{\mathcal{A}_0} + \boldsymbol{H}_*^{\mathcal{A}_0}, \hat{\gamma}_t - \alpha_t\hat{\boldsymbol{\Sigma}}_t^{-1}\right\rangle\right] \\
&\leq \mathcal{O}(1).
\end{aligned}$$

$\square$

## D.7 Finishing up

Recall the regret decomposition at the beginning of [Appendix D](#). From [Lemma 24](#), [Lemma 26](#), [Lemma 27](#), and [Lemma 28](#), we have

$$\textbf{FTRL-Reg} = \textbf{Penalty} + \textbf{Stability-1} + \textbf{Stability-2} + \textbf{Error}$$

$$\leq \mathcal{O}\left(\frac{d\log(T)}{\eta_T} + d\sum_{t=1}^T \eta_t + d\sum_{t=1}^T \alpha_t + \delta T^2\right)$$

From Lemma 20 and Lemma 23, we can cancel out the additional regret induced by bias through the well-designed bonus term. Namely,

$$
\begin{aligned}
\mathbf{Bias} + \mathbf{Bonus} &= \frac{1}{4} \sum_{t=1}^{T} \alpha_t \|x_t - u\|_{\hat{\Sigma}_t^{-1}}^2 + \mathcal{O}\left( \sum_{t=1}^{T} \frac{d^3 \log(T/\delta)}{\alpha_t t} + \delta T^2 \right) \\
&+ 2(d+2) \sum_{t=1}^{T} \alpha_t - \frac{1}{4} \sum_{t=1}^{T} \alpha_t \|u - x_t\|_{\hat{\Sigma}_t^{-1}}^2 + \mathcal{O}\left( \sum_{t=1}^{T} \frac{d^3 \alpha_t \log \frac{T}{\delta}}{t} + \delta \sum_{t=1}^{T} \alpha_t T \right) \\
&= \mathcal{O}\left( d \sum_{t=1}^{T} \alpha_t + \sum_{t=1}^{T} \frac{d^3 \log(T/\delta)}{\alpha_t t} + \sum_{t=1}^{T} \frac{d^3 \alpha_t \log (T/\delta)}{t} + \delta T^2 \right)
\end{aligned}
$$

Thus, we have

$$
\begin{aligned}
\mathrm{Reg} &= \mathbf{Bias} + \mathbf{Bonus} + \mathbf{FTRL\text{-}Reg} \\
&= \mathcal{O}\left( \frac{d \log(T)}{\eta_T} + d \sum_{t=1}^{T} \eta_t + d \sum_{t=1}^{T} \alpha_t + \sum_{t=1}^{T} \frac{d^3 \log(T/\delta)}{\alpha_t t} + \sum_{t=1}^{T} \frac{d^3 \alpha_t \log (T/\delta)}{t} + \delta T^2 \right)
\end{aligned}
$$

Recall that we have an additional condition in Lemma 27 such that for any $t$, $\eta_t \alpha_t \leq \frac{1}{64t}$. Picking $\alpha_t = \frac{d}{\sqrt{t}}, \eta_t = \frac{1}{64d\sqrt{t}}$ and $\delta = \frac{1}{T^2}$, we get

$$
\mathrm{Reg} = \mathcal{O}\left( d^2 \sqrt{T} \log(T) + d^4 \log(T) \right) = \mathcal{O}(d^2 \sqrt{T} \log(T))
$$

where we assume $d^2 \leq \sqrt{T}$ without loss of generality (otherwise the bound is vacuous).

# E   Handling Misspecification

In this section, we discuss how to handle misspecification as defined in Section 3.6. In Appendix E.1, we study the case where the amount of misspecification $\varepsilon$ is known by the learner. In Appendix E.2, we use a blackbox approach to turn it into an algorithm that achieves almost the same regret bound (up to $\log T$ factors) without knowning $\varepsilon$.

## E.1   Known misspecification

As discussed in Section 3.6, when the amount of misspecification $\varepsilon$ is known, we still use Algorithm 1, but with different $\alpha_t$ and $\eta_t$. Throughout this subsection, we let $\alpha_t = \frac{d}{\sqrt{t}} + \frac{\varepsilon}{\sqrt{d}}$ and $\eta_t = \frac{1}{64\left(d\sqrt{t} + \frac{\varepsilon}{\sqrt{d}}t\right)}$, and point out the modifications of the analysis from Appendix D.

We start with the regret decomposition similar to that in Appendix D, but here we define

$$
\begin{aligned}
y_t &= \operatorname*{argmin}_{y \in \mathbb{B}_2^d} \max_{\mathcal{A} \in \mathrm{supp}(D)} \max_{a \in \mathcal{A}} |f_t(a) - \langle a, y \rangle|, \\
\varepsilon_t &= \max_{\mathcal{A} \in \mathrm{supp}(D)} \max_{a \in \mathcal{A}} |f_t(a) - \langle a, y_t \rangle|, \\
c_t(a) &= f_t(a) - \langle a, y_t \rangle.
\end{aligned}
$$

The regret decomposition goes as follows:

$$\text{Reg}(u) = \mathbb{E}\left[\sum_{t=1}^{T}\left(f_t(a_t) - f_t(u^{\mathcal{A}_t})\right)\right]$$

$$\leq \mathbb{E}\left[\sum_{t=1}^{T}\left\langle a_t - u^{\mathcal{A}_t}, y_t\right\rangle\right] + \sum_{t=1}^{T}\varepsilon_t$$

$$\leq \mathbb{E}\left[\sum_{t=1}^{T}\left\langle \boldsymbol{H}_t^{\mathcal{A}_t} - \boldsymbol{U}^{\mathcal{A}_t}, \gamma_t\right\rangle\right] + \varepsilon T = \mathbb{E}\left[\sum_{t=1}^{T}\left\langle \boldsymbol{H}_t^{\mathcal{A}_0} - \boldsymbol{U}^{\mathcal{A}_0}, \gamma_t\right\rangle\right] + \varepsilon T$$

$$\leq \underbrace{\mathbb{E}\left[\sum_{t=1}^{T}\left\langle \boldsymbol{H}_t^{\mathcal{A}_0} - \boldsymbol{U}^{\mathcal{A}_0}, \gamma_t - \hat{\gamma}_t\right\rangle\right]}_{\textbf{Bias}} + \underbrace{\mathbb{E}\left[\sum_{t=1}^{T}\left\langle \boldsymbol{H}_t^{\mathcal{A}_0} - \boldsymbol{U}^{\mathcal{A}_0}, \alpha_t\hat{\boldsymbol{\Sigma}}_t^{-1}\right\rangle\right]}_{\textbf{Bonus}}$$

$$+ \underbrace{\mathbb{E}\left[\sum_{t=1}^{T}\left\langle \boldsymbol{H}_t^{\mathcal{A}_0} - \boldsymbol{U}^{\mathcal{A}_0}, \hat{\gamma}_t - \alpha_t\hat{\boldsymbol{\Sigma}}_t^{-1}\right\rangle\right]}_{\textbf{FTRL-Reg}} + \varepsilon T.$$

Now $\hat{y}_t = \hat{\Sigma}_t^{-1}(a_t - \hat{x}_t)\ell_t$ with $\mathbb{E}[\ell_t] = a_t^\top y_t + c_t(a_t)$.

For the **Bias** term, the proof is almost the same as Lemma 20. The only difference is that from the fourth line, we have

$$\mathbb{E}_t\left[(x_t - u)^\top\left(y_t - \hat{\Sigma}_t^{-1}(a_t - \hat{x}_t)\left(a_t^\top y_t + c_t(a_t)\right)\right)\right]$$

for some $c_t(a_t)$ such that $|c_t(a_t)| \leq \varepsilon_t$. This leads to an additional term of

$$\mathbb{E}_t^{\mathcal{E}}\left[-(x_t - u)^\top\hat{\Sigma}_t^{-1}(a_t - \hat{x}_t)c_t(a_t)\right]$$

$$\leq \mathbb{E}_t^{\mathcal{E}}\left[\sqrt{(x_t - u)^\top\hat{\Sigma}_t^{-1}c_t(a_t)^2(a_t - \hat{x}_t)(a_t - \hat{x}_t)^\top\hat{\Sigma}_t^{-1}(x_t - u)}\right]$$

$$\leq \mathbb{E}_t^{\mathcal{E}}\left[\sqrt{(x_t - u)^\top\hat{\Sigma}_t^{-1}\mathbb{E}_{\mathcal{A}_t,a_t}\left[c_t(a_t)^2(a_t - \hat{x}_t)(a_t - \hat{x}_t)^\top\right]\hat{\Sigma}_t^{-1}(x_t - u)}\right]$$

$$\leq \mathbb{E}_t^{\mathcal{E}}\left[\varepsilon_t\sqrt{(x_t - u)^\top\hat{\Sigma}_t^{-1}\left(\mathbb{E}_{\mathcal{A}_t,a_t}\left[(a_t - \hat{x}_t)(a_t - \hat{x}_t)^\top\right]\right)\hat{\Sigma}_t^{-1}(x_t - u)}\right]$$

$$\leq \mathbb{E}_t^{\mathcal{E}}\left[\varepsilon_t\sqrt{(x_t - u)^\top\hat{\Sigma}_t^{-1}H_t\hat{\Sigma}_t^{-1}(x_t - u)}\right]$$

$$\leq \varepsilon_t\|x_t - u\|_{\hat{\Sigma}_t^{-1}}$$

Plugging it into the proof of Lemma 20, we have

$$\mathbb{E}_t^{\mathcal{E}}\left[\left\langle \boldsymbol{H}_t^{\mathcal{A}_0} - \boldsymbol{U}^{\mathcal{A}_0}, \gamma_t - \hat{\gamma}_t\right\rangle\right] \leq \mathcal{O}\left(\sqrt{\frac{d^3\log(T/\delta)}{t}} + \varepsilon_t\right)\|x_t - u\|_{\hat{\Sigma}_t^{-1}}$$

$$\leq \frac{\alpha_t}{4}\|x_t - u\|_{\hat{\Sigma}_t^{-1}}^2 + \mathcal{O}\left(\frac{d^3\log(T/\delta)}{\alpha_t t} + \frac{\varepsilon_t^2}{\alpha_t}\right)$$

Other parts of the proof follow those in Lemma 20. Finally, we get

$$\textbf{Bias} = \mathbb{E}\left[\sum_{t=1}^{T}\left\langle \boldsymbol{H}_t^{\mathcal{A}_0} - \boldsymbol{U}^{\mathcal{A}_0}, \gamma_t - \hat{\gamma}_t\right\rangle\right]$$

$$\leq \frac{1}{4}\sum_{t=1}^{T}\alpha_t\|x_t - u\|_{\hat{\Sigma}_t^{-1}}^2 + \mathcal{O}\left(\sum_{t=1}^{T}\frac{d^3\log(T/\delta)}{\alpha_t t} + \sum_{t=1}^{T}\frac{\varepsilon_t^2}{\alpha_t} + \delta T^2\right)$$

The **Bonus** term will not be affected, according to Lemma 23, we have

$$\textbf{Bonus} \leq 2(d+2)\sum_{t=1}^{T}\alpha_t - \frac{1}{4}\sum_{t=1}^{T}\alpha_t\|u - x_t\|_{\hat{\Sigma}_t^{-1}}^2 + \mathcal{O}\left(\sum_{t=1}^{T}\frac{d^3\alpha_t\log\left(T/\delta\right)}{t} + \delta T^2\right)$$

The **Penalty** term will not be affected, according to Lemma 24, we have

$$\frac{F(\overline{\boldsymbol{U}}^{\mathcal{A}_0}) - \min_{\boldsymbol{H}\in\mathcal{H}^{\mathcal{A}_0}}F(\boldsymbol{H})}{\eta_T} \leq \frac{2d\log(T)}{\eta_T}$$

**Stability-1** term is also unchanged, as we assume that $\ell_t$ still lies in $[-1,1]$ even under misspecification. We still have

$$\textbf{Stability-1} \leq \mathcal{O}\left(d\sum_{t=1}^{T}\eta_t + \delta T^2\right)$$

The **Stability-2** term will not be affected as long as $\eta_t\alpha_t \leq \frac{1}{64t}$. According to Lemma 27, we have

$$\textbf{Stability-2} \leq \mathcal{O}\left(d\sum_{t=1}^{T}\alpha_t + \delta T^2\right)$$

The **Error** term is also unaffected. We still have $\textbf{Error} = \mathcal{O}(1)$.

Adding these terms together, the regret caused by bias and the negative term induced by bonus cancel out. We have

$$\text{Reg} = \mathcal{O}\left(\frac{d\log(T)}{\eta_T} + d\sum_{t=1}^{T}(\eta_t + \alpha_t) + \sum_{t=1}^{T}\frac{d^3\log(T/\delta)}{\alpha_t t} + \sum_{t=1}^{T}\frac{d^3\alpha_t\log\left(T/\delta\right)}{t} + \sum_{t=1}^{T}\frac{\varepsilon_t^2}{\alpha_t} + \delta T^2\right)$$

Recall that we pick $\alpha_t = \frac{d}{\sqrt{t}} + \frac{\varepsilon}{\sqrt{d}}$, $\eta_t = \frac{1}{64d\sqrt{t} + 64\frac{\varepsilon}{\sqrt{d}}t}$ and $\delta = \frac{1}{T^2}$. This gives

$$\text{Reg} = \mathcal{O}(d^2\sqrt{T}\log(T) + d^4\log(T) + \sqrt{d}\varepsilon T) = \mathcal{O}(d^2\sqrt{T}\log(T) + \sqrt{d}\varepsilon T)$$

where we assume $d^2 \leq \sqrt{T}$ without loss of generality.

### E.2 Unknown misspecification

In this subsection, we use a model selection technique to convert the algorithm in Appendix E.1 which requires knowledge on $\varepsilon$ into an algorithm that achieves a similar regret bound without knowing $\varepsilon$. Such a procedure to handle unknown misspecification/corruption has appeared in several previous works [Foster et al., 2020, Wei et al., 2022], though we adopt the technique in Jin et al. [2023] to handle the adversarial case.

The idea here is a black-box reduction which turns an algorithm that only deals with known $\varepsilon$ to one that handles unknown $\varepsilon$. More specifically, the reduction has two layers. The bottom layer takes as input an arbitrary misspecification-robust algorithm that operates under known $\varepsilon$ (e.g., Algorithm 1), and outputs a *stable* misspecification-robust algorithm (formally defined later) that still operates under known $\varepsilon$. The top layer follows the standard Corral idea and takes as input a stable algorithm that operates under known $\varepsilon$, and outputs an algorithm that operates under unknown $\varepsilon$. Below, we explain these two layers of reduction in details.

**Bottom Layer (from an Arbitrary Algorithm to a Stable Algorithm)** The input of the bottom layer is an arbitrary misspecification-robust algorithm, formally defined as:

**Definition 29.** *An algorithm is misspecification-robust if it takes $\theta$ as input, and achieves the following regret for any random stopping time $t' \leq T$ and any policy $u$:*

$$\mathbb{E}\left[\sum_{t=1}^{t'}(f_t(a_t) - f_t(u^{\mathcal{A}_t}))\right] \leq \mathbb{E}\left[c_1\sqrt{t'} + c_2\theta\right] + \Pr\left[\varepsilon_{1;t'} > \theta\right]T$$

*for problem-dependent and $\log(T)$ factors $c_1, c_2 \geq 1$ and $\varepsilon_{1:t'} \triangleq \sqrt{t'\sum_{\tau=1}^{t'}\varepsilon_\tau^2}$.*

---

**Algorithm 3 ST**able **A**lgorithm **B**y **I**ndependent **L**earners and **I**nstance **SE**lection (STABILISE)

---

**Input**: $\varepsilon$ and a base algorithm satisfying Definition 29.

**Initialize**: $\lceil \log_2 T \rceil$ instances of the base algorithm $\mathsf{ALG}_1, \ldots, \mathsf{ALG}_{\lceil \log_2 T \rceil}$, where $\mathsf{ALG}_j$ is configured with the parameter

$$\theta = \theta_j \triangleq 2^{-j}\varepsilon T + 4\sqrt{2^{-j}T \log T} + 8\log(T).$$

**for** $t = 1, 2, \ldots$ **do**

> Receive $w_t$.
>
> **if** $w_t \leq \frac{1}{T}$ **then**
>
>> play an arbitrary policy $\pi_t$
>>
>> **continue** (without updating any instances)
>
> Let $j_t$ be such that $w_t \in (2^{-j_t-1}, 2^{-j_t}]$.
>
> Let $\pi_t$ be the policy suggested by $\mathsf{ALG}_{j_t}$.
>
> Output $\pi_t$.
>
> If feedback is received, send it to $\mathsf{ALG}_{j_t}$ with probability $\frac{2^{-j_t-1}}{w_t}$, and discard it otherwise.

---

In our case, $c_1 = \Theta(d^2 \log T)$ and $c_2 = \Theta(\sqrt{d})$. While the regret bound in Definition 29 might look cumbersome, it is in fact fairly reasonable: if the guess $\theta$ is not smaller than the true amount of $\varepsilon_{1:t'}$, the regret should be of order $d^2\sqrt{t'} + \sqrt{d}\theta$; otherwise, the regret bound is vacuous since $T$ is its largest possible value. The only extra requirement is that the algorithm needs to be *anytime* (i.e., the regret bound holds for any stopping time $t'$), but even this is known to be easily achievable by using a doubling trick over a fixed-time algorithm. It is then clear that Algorithm 1 (together with a doubling trick) indeed satisfies Definition 29.

As mentioned, the output of the bottom layer is a stable robust algorithm. To characterize stability, we follow Agarwal et al. [2017] and define a new learning protocol that abstracts the interaction between the output algorithm of the bottom layer and the master algorithm from the top layer:

**Protocol 1.** In every round $t$, before the learner makes a decision, a probability $w_t \in [0, 1]$ is revealed to the learner. After making a decision, the learner sees the desired feedback from the environment with probability $w_t$, and sees nothing with probability $1 - w_t$.

One can convert any misspecification-robust algorithm (defined in Definition 29) into a stable misspecification-robust algorithm (characterized in Theorem 30).

This conversion is achieved by a procedure that called STABILISE (see Algorithm 3 for details). The high-level idea of STABILISE is as follows. Noticing that the challenge when learning in Protocol 1 is that $w_t$ varies over time, we discretize the value of $w_t$ and instantiate one instance of the input algorithm to deal with one possible discretized value, so that it is learning in Protocol 1 but with a *fixed* $w_t$, making it straightforward to bound its regret based on what it promises in Definition 29.

More concretely, STABILISE instantiates $\mathcal{O}(\log_2 T)$ instances $\{\mathsf{ALG}_j\}_{j=0}^{\lceil \log_2 T \rceil}$ of the input algorithm that satisfies Definition 29, each with a different parameter $\theta_j$. Upon receiving $w_t$ from the environment, it dispatches round $t$ to the $j$-th instance where $j$ is such that $w_t \in (2^{-j-1}, 2^{-j}]$, and uses the policy generated by $\mathsf{ALG}_j$ to interact with the environment (if $w_t \leq \frac{1}{T}$, simply ignore this round). Based on Protocol 1, the feedback for this round is received with probability $w_t$. To *equalize* the probability of $\mathsf{ALG}_j$ receiving feedback as mentioned in the high-level idea, when the feedback is actually obtained, STABILISE sends it to $\mathsf{ALG}_j$ only with probability $\frac{2^{-j-1}}{w_t}$ (and discards it otherwise). This way, every time $\mathsf{ALG}_j$ is assigned to a round, it always receives the desired feedback with probability $w_t \cdot \frac{2^{-j-1}}{w_t} = 2^{-j-1}$. This equalization step allows us to use the original guarantee of the base algorithm (Definition 29) and run it as it is, without requiring it to perform extra importance weighting steps as in Agarwal et al. [2017].

The choice of $\theta_j$ is crucial in making sure that STABILISE only has $\varepsilon T$ regret overhead instead of $\frac{\varepsilon T}{\min_{t \in [T]} w_t}$. Since $\mathsf{ALG}_j$ only receives feedback with probability $2^{-j-1}$, the expected total misspecification it experiences is on the order of $2^{-j-1}\varepsilon T$. Therefore, its input parameter $\theta_j$ only needs to be of this order instead of the total amount of misspecification $\varepsilon T$.

The formal guarantee of the conversion is stated in the following Theorem 30.

**Theorem 30.** *If an algorithm is misspecification robust according to Definition 29 for some constants $(c_1, c_2)$, then Algorithm 3 ensures*

$$\text{Reg} \leq \mathcal{O}\left(\mathbb{E}\left[c_1'\sqrt{T\rho_T}\right] + c_2'\varepsilon T\right)$$

*under Protocol 1, where $\rho_T = \frac{1}{\min_{t\in[T]} w_t}$, with $c_1' = \Theta((c_1 + c_2)\sqrt{\log T})$.*

*Proof of Theorem 30.* Define indicators

$$g_{t,j} = \mathbb{I}\{w_t \in (2^{-j-1}, 2^{-j}]\}$$
$$h_{t,j} = \mathbb{I}\{\mathsf{ALG}_j \text{ receives the feedback for episode } t\}.$$

Now we consider the regret of $\mathsf{ALG}_j$. Notice that $\mathsf{ALG}_j$ makes an update only when $g_{t,j}h_{t,j} = 1$. By the guarantee of the base algorithm (Definition 29), we have

$$\mathbb{E}\left[\sum_{t=1}^{T}(f_t(a_t) - f_t(u^{\mathcal{A}_t}))g_{t,j}h_{t,j}\right]$$

$$\leq \mathbb{E}\left[c_1\sqrt{\sum_{t=1}^{T} g_{t,j}h_{t,j}} + c_2\theta_j \max_{t\leq T} g_{t,j}\right] + \Pr\left[\sqrt{\left(\sum_{t=1}^{T} g_{t,j}h_{t,j}\right)\left(\sum_{t=1}^{T} \varepsilon_t^2 g_{t,j}h_{t,j}\right)} > \theta_j\right] T. \tag{40}$$

We first bound the last term: Notice that $\mathbb{E}[h_{t,j}|g_{t,j}] = 2^{-j-1}g_{t,j}$ by Algorithm 3. Therefore,

$$\sum_{t=1}^{T} \varepsilon_t^2 g_{t,j}\mathbb{E}[h_{t,j}|g_{t,j}] = 2^{-j-1}\sum_{t=1}^{T} \varepsilon_t^2 g_{t,j} \leq 2^{-j-1}\varepsilon^2 T \tag{41}$$

$$\sum_{t=1}^{T} g_{t,j}\mathbb{E}[h_{t,j}|g_{t,j}] = 2^{-j-1}\sum_{t=1}^{T} g_{t,j} \leq 2^{-j-1}T \tag{42}$$

By Freedman's inequality, with probability at least $1 - \frac{1}{T^2}$,

$$\sum_{t=1}^{T} \varepsilon_t^2 g_{t,j}h_{t,j} - \sum_{t=1}^{T} \varepsilon_t^2 g_{t,j}\mathbb{E}[h_{t,j}|g_{t,j}]$$

$$\leq 2\sqrt{\sum_{t=1}^{T} (\varepsilon_t)^4 g_{t,j}\mathbb{E}[h_{t,j}|g_{t,j}]\log(T) + 4\log(T)}$$

$$\leq 4\sqrt{\sum_{t=1}^{T} \varepsilon_t^2 g_{t,j}\mathbb{E}[h_{t,j}|g_{t,j}]\log(T) + 4\log(T)}$$

$$\leq \sum_{t=1}^{T} \varepsilon_t^2 g_{t,j}\mathbb{E}[h_{t,j}|g_{t,j}] + 8\log(T) \qquad \text{(AM-GM inequality)}$$

which gives

$$\sum_{t=1}^{T} \varepsilon_t^2 g_{t,j}h_{t,j} \leq 2\sum_{t=1}^{T} \varepsilon_t^2 g_{t,j}\mathbb{E}[h_{t,j}|g_{t,j}] + 8\log(T) \leq 2^{-j}\varepsilon^2 T + 8\log(T)$$

with probability at least $1 - \frac{1}{T^2}$ using Eq. (41). Similarly,

$$\sum_{t=1}^{T} g_{t,j}h_{t,j} \leq 2\sum_{t=1}^{T} g_{t,j}\mathbb{E}[h_{t,j}|g_{t,j}] + 8\log(T) \leq 2^{-j}T + 8\log(T)$$

with probability at least $1 - \frac{1}{T^2}$. Therefore, with probability at least $1 - \frac{2}{T^2}$,

$$\sqrt{\left(\sum_{t=1}^{T} g_{t,j} h_{t,j}\right)\left(\sum_{t=1}^{T} \varepsilon_t^2 g_{t,j} h_{t,j}\right)} \leq \sqrt{2^{-2j}\varepsilon^2 T^2 + 16 \cdot 2^{-j} T \log T + 64 \log^2 T}$$

$$\leq 2^{-j}\varepsilon T + 4\sqrt{2^{-j} T \log T} + 8 \log(T)$$
$$\leq \theta_j$$

Therefore, the last term in Eq. (40) is bounded by $\frac{2}{T^2} T \leq \frac{2}{T}$.

Next, we deal with other terms in Eq. (40). Again, by $\mathbb{E}[h_{t,j}|g_{t,j}] = 2^{-j-1} g_{t,j}$, Eq. (40) implies

$$2^{-j-1}\mathbb{E}\left[\sum_{t=1}^{T}(f_t(a_t) - f_t(u^{\mathcal{A}_t}))g_{t,j}\right] \leq \mathbb{E}\left[c_1\sqrt{2^{-j-1}\sum_{t=1}^{T}g_{t,j} + c_2\theta_j\max_{t\leq T}g_{t,j}}\right] + \frac{2}{T}.$$

which implies after rearranging:

$$\mathbb{E}\left[\sum_{t=1}^{T}(f_t(a_t) - f_t(u^{\mathcal{A}_t}))g_{t,j}\right]$$

$$\leq \mathbb{E}\left[c_1\sqrt{\frac{1}{2^{-j-1}}\sum_{t=1}^{T}g_{t,j} + \left(\frac{c_2\theta_j}{2^{-j-1}}\right)\max_{t\leq T}g_{t,j}}\right] + \frac{2}{T2^{-j-1}}$$

$$\leq \mathbb{E}\left[c_1\sqrt{\sum_{t=1}^{T}\frac{2g_{t,j}}{w_t} + 4c_2\left(\varepsilon T + \sqrt{\frac{T\log T}{2^{-j}}} + \log T\right)\max_{t\leq T}g_{t,j}}\right] + \frac{2}{T2^{-j-1}}.$$

(using that when $g_{t,j} = 1$, $\frac{1}{2^{-j-1}} \leq \frac{2}{w_t}$, and the definition of $\theta_j$)

Now, summing this inequality over all $j \in \{0, 1, \ldots, \lceil \log_2 T \rceil\}$, we get

$$\mathbb{E}\left[\sum_{t=1}^{T}(f_t(a_t) - f_t(u^{\mathcal{A}_t}))\mathbb{I}\left\{w_t > \frac{1}{T}\right\}\right]$$

$$\leq \mathcal{O}\left(\mathbb{E}\left[c_1\sqrt{N\sum_{t=1}^{T}\frac{1}{w_t} + Nc_2\varepsilon T + c_2\sqrt{\frac{T\log T}{\min_{t\leq T}w_t}} + c_2 N\log T}\right] + 1\right)$$

$$\leq \mathcal{O}\left(\mathbb{E}\left[(c_1 + c_2)\sqrt{T\log(T)\rho_T}\right] + c_2\varepsilon T\log T\right)$$

where $N \leq \mathcal{O}(\log T)$ is the number of $\mathsf{ALG}_j$'s that has been executed at least once.

On the other hand,

$$\mathbb{E}\left[\sum_{t=1}^{T}(f_t(a_t) - f_t(u^{\mathcal{A}_t}))\mathbb{I}\left\{w_t \leq \frac{1}{T}\right\}\right] < T\mathbb{E}\left[\mathbb{I}\left\{\rho_T \geq T\right\}\right] \leq \mathbb{E}\left[\rho_T\right].$$

Combining the two parts and using the assumption $c_2 \geq 1$ finishes the proof. $\qquad\square$

**Top Layer (from Known $\varepsilon$ to Unknown $\varepsilon$)**    In this subsection, we use the algorithm that we construct in Theorem 30 as a base algorithm, and further construct an algorithm with $\sqrt{T} + \varepsilon$ regret under unknown $\varepsilon$. The idea is to run multiple base algorithms, each with a different hypothesis on $\varepsilon$; on top of them, run another multi-armed bandit algorithm to adaptively choose among them. The goal is to let the top-level bandit algorithm perform almost as well as the best base algorithm. This is the Corral idea outlined in Agarwal et al. [2017], Foster et al. [2020], Luo et al. [2022], and the algorithm is presented in Algorithm 4.

**Theorem 31.** *Using an algorithm constructed in Theorem 30 as a base algorithm, Algorithm 4 ensures* $\text{Reg} = \mathcal{O}\left(c_1'\sqrt{T\log^3 T} + c_2'\varepsilon T\right)$ *without knowing $\varepsilon$.*

---

**Algorithm 4** (A Variant of) Corral

---

**Initialize**: a log-barrier algorithm with each arm being an instance of an algorithm satisfying the guarantee in Theorem 30. The hypothesis on $\varepsilon T$ is set to $2^i$ for arm $i$ ($i = 1, 2, \ldots, M \triangleq \lceil \log_2 T \rceil$).
**Initialize**: $\rho_{0,i} = M, \ \forall i$.

---

**for** $t = 1, 2, \ldots, T$ **do**

Let

$$w_t = \underset{w \in \Delta(M), w_i \geq \frac{1}{T}, \forall i}{\operatorname{argmin}} \left\{ \left\langle w, \sum_{\tau=1}^{t-1}(\hat{z}_\tau - r_\tau) \right\rangle + \frac{1}{\eta} \sum_{i=1}^{M} \log \frac{1}{w_i} \right\}$$

where $\eta = \frac{1}{4c_1'\sqrt{T}}$.
For all $i$, send $w_{t,i}$ to instance $i$.
Draw $i_t \sim w_t$.
Execute the $a_t$ output by instance $i_t$
Receive the loss $z_{t,i_t}$ for action $a_t$ (whose expectation is $f_t(a_t)$) and send it to instance $i_t$.
Define for all $i$:

$$\hat{z}_{t,i} = \frac{z_{t,i}\mathbb{I}[i_t = i]}{w_{t,i}},$$

$$\rho_{t,i} = \min_{\tau \leq t} \frac{1}{w_{\tau,i}},$$

$$r_{t,i} = c_1' \left( \sqrt{\rho_{t,i}T} - \sqrt{\rho_{t-1,i}T} \right).$$

---

The top-level bandit algorithm is an FTRL with log-barrier regularizer. We first state the standard regret bound of FTRL under log-barrier regularizer, whose proof can be found in, e.g., Theorem 7 of Wei and Luo [2018].

**Lemma 32.** *The FTRL algorithm over a convex subset $\Omega$ of the $(M-1)$-dimensional simplex $\Delta(M)$:*

$$w_t = \underset{w \in \Omega}{\operatorname{argmin}} \left\{ \left\langle w, \sum_{\tau=1}^{t-1} \ell_\tau \right\rangle + \frac{1}{\eta} \sum_{i=1}^{M} \log \frac{1}{w_i} \right\}$$

*ensures for all $u \in \Omega$,*

$$\sum_{t=1}^{T} \langle w - u, \ell_t \rangle \leq \frac{M \log T}{\eta} + \eta \sum_{t=1}^{T} \sum_{i=1}^{M} w_{t,i}^2 \ell_{t,i}^2$$

*as long as $\eta w_{t,i}|\ell_{t,i}| \leq \frac{1}{2}$ for all $t, i$.*

*Proof of Theorem 31.* The Corral algorithm is essential an FTRL with log-barrier regularizer. To apply Lemma 32, we first verify the condition $\eta w_{t,i}|\ell_{t,i}| \leq \frac{1}{2}$ where $\ell_{t,i} = \hat{z}_{t,i} - r_{t,i}$. By our choice of $\eta$,

$$\eta w_{t,i}|\hat{z}_{t,i}| \leq \eta z_{t,i} \leq \frac{1}{4}, \qquad \text{(because } c_1' \geq 1\text{)}$$

$$\eta w_{t,i} r_{t,i} = \eta c_1' \sqrt{T} w_{t,i}(\sqrt{\rho_{t,i}} - \sqrt{\rho_{t-1,i}}).$$

The right-hand side of the last equality is non-zero only when $\rho_{t,i} > \rho_{t-1,i}$, implying that $\rho_{t,i} = \frac{1}{w_{t,i}}$. Therefore, we further bound it by

$$
\begin{aligned}
\eta w_{t,i} r_{t,i} &\leq \eta c_1' \sqrt{T} \frac{1}{\rho_{t,i}} (\sqrt{\rho_{t,i}} - \sqrt{\rho_{t-1,i}}) \\
&= \eta c_1' \sqrt{T} \left( \frac{1}{\sqrt{\rho_{t,i}}} - \frac{\sqrt{\rho_{t-1,i}}}{\rho_{t,i}} \right) \\
&\leq \eta c_1' \sqrt{T} \left( \frac{1}{\sqrt{\rho_{t-1,i}}} - \frac{1}{\sqrt{\rho_{t,i}}} \right) \qquad (\frac{1}{\sqrt{a}} - \frac{\sqrt{b}}{a} \leq \frac{1}{\sqrt{b}} - \frac{1}{\sqrt{a}} \text{ for } a, b > 0)
\end{aligned}
$$

$$(43)$$

$$
\begin{aligned}
&\leq \eta c_1' \sqrt{T} && (\rho_{t,i} \geq 1) \\
&= \frac{1}{4} && (\text{definition of } \eta)
\end{aligned}
$$

which can be combined to get the desired property $\eta w_{t,i} |\hat{z}_{t,i} - r_{t,i}| \leq \frac{1}{2}$.

Hence, by the regret guarantee of log-barrier FTRL (Lemma 32), we have

$$
\mathbb{E}\left[ \sum_{t=1}^{T} (z_{t,i_t} - z_{t,i^\star}) \right]
$$

$$
\leq \mathcal{O}\left( \frac{M \log T}{\eta} + \eta \mathbb{E}\left[ \underbrace{\sum_{t=1}^{T} \sum_{i=1}^{M} w_{t,i}^2 (\hat{z}_{t,i} - r_{t,i})^2}_{\textbf{term}_1} \right] \right) + \mathbb{E}\left[ \underbrace{\sum_{t=1}^{T} \left( \sum_{i=1}^{M} w_{t,i} r_{t,i} - r_{t,i^\star} \right)}_{\textbf{term}_2} \right]
$$

where $i^\star$ is the smallest $i$ such that $2^i$ upper bounds the true total misspecification amount $\varepsilon T$.

**Bounding term$_1$:**

$$
\textbf{term}_1 \leq 2\eta \sum_{t=1}^{T} \sum_{i=1}^{M} w_{t,i}^2 (\hat{z}_{t,i}^2 + r_{t,i}^2)
$$

where

$$
2\eta \sum_{t=1}^{T} \sum_{i=1}^{M} w_{t,i}^2 \hat{z}_{t,i}^2 = 2\eta \sum_{t=1}^{T} \sum_{i=1}^{M} z_{t,i}^2 \mathbb{I}\{i_t = i\} \leq \mathcal{O}(\eta T)
$$

and

$$
\begin{aligned}
2\eta \sum_{t=1}^{T} \sum_{i=1}^{M} w_{t,i}^2 r_{t,i}^2 &\leq 4\eta \sum_{t=1}^{T} \sum_{i=1}^{M} (c_1' \sqrt{T})^2 \left( \frac{1}{\sqrt{\rho_{t-1,i}}} - \frac{1}{\sqrt{\rho_{t,i}}} \right)^2 && (\text{continue from Eq. (43)}) \\
&\leq 4\eta c_1'^2 T \times \sum_{t=1}^{T} \sum_{i=1}^{M} \left( \frac{1}{\sqrt{\rho_{t-1,i}}} - \frac{1}{\sqrt{\rho_{t,i}}} \right) \\
& && (\frac{1}{\sqrt{\rho_{t-1,i}}} - \frac{1}{\sqrt{\rho_{t,i}}} \leq 1 \text{ and } 1 - a \leq -\ln a) \\
&\leq 4\eta c_1'^2 T M^{\frac{3}{2}}. && (\text{telescoping and using } \rho_{0,i} = M \text{ and } \rho_{T,i} \leq T)
\end{aligned}
$$

**Bounding term$_2$:**

$$
\begin{aligned}
\textbf{term}_2 &= \sum_{t=1}^{T} \sum_{i=1}^{M} w_{t,i} r_{t,i} - \sum_{t=1}^{T} r_{t,i^\star} \\
&\leq c_1' \sqrt{T} \sum_{t=1}^{T} \sum_{i=1}^{M} \left( \frac{1}{\sqrt{\rho_{t-1,i}}} - \frac{1}{\sqrt{\rho_{t,i}}} \right) - \left( c_1' \sqrt{\rho_{T,i^\star} T} - c_1' \sqrt{\rho_{0,i^\star} T} \right) \\
& \qquad\qquad (\text{continue from Eq. (43) and using } 1 - a \leq -\ln a) \\
&\leq \mathcal{O}\left( c_1' \sqrt{T} M^{\frac{3}{2}} \right) - c_1' \sqrt{\rho_{T,i^\star} T}.
\end{aligned}
$$

Combining the two terms and using $\eta = \Theta\left(\frac{1}{c_1'\sqrt{T}+c_2'}\right)$, $M = \Theta(\log T)$, we get

$$\mathbb{E}\left[\sum_{t=1}^{T}(f_t(a_t) - z_{t,i^\star})\right] = \mathbb{E}\left[\sum_{t=1}^{T}(z_{t,i_t} - z_{t,i^\star})\right]$$

$$= \mathcal{O}\left(c_1'\sqrt{T\log^3 T}\right) - \mathbb{E}\left[c_1'\sqrt{\rho_{T,i^\star}T}\right] \tag{44}$$

On the other hand, by the guarantee of the base algorithm (Theorem 30) and that $\varepsilon T \in [2^{i^\star-1}, 2^{i^\star}]$, we have

$$\mathbb{E}\left[\sum_{t=1}^{T}(z_{t,i^\star} - f_t(u^{\mathcal{A}_t}))\right] \leq \mathbb{E}\left[c_1'\sqrt{\rho_{T,i^\star}T}\right] + c_2'\varepsilon T. \tag{45}$$

Combining Eq. (44) and Eq. (45), we get

$$\mathbb{E}\left[\sum_{t=1}^{T}(f_t(a_t) - f_t(u^{\mathcal{A}_t}))\right] \leq \mathcal{O}\left(c_1'\sqrt{T\log^3 T}\right) + c_2'\varepsilon T,$$

which finishes the proof. $\qquad\square$

*Proof of Theorem 3.* As shown in Appendix E.1, our Algorithm 1 can be adapted to satisfy Definition 29 with $c_1 = \Theta(d^2 \log T)$ and $c_2 = \Theta(\sqrt{d})$. By a concatenation of Theorem 30 and Theorem 31, we conclude that there is an algorithm that achieves

$$\mathcal{O}\left((c_1 + c_2)\sqrt{T}\log^2 T + c_2\varepsilon T\log T\right) = \mathcal{O}\left(d^2\sqrt{T}\log^2 T + \sqrt{d}\varepsilon T\log T\right).$$

regret under unknown $\varepsilon$. $\qquad\square$

# F    Analysis for Linear EXP4

*Proof of Theorem 4.* We first show that

$$\forall \pi \in \Pi: \;\; \text{Reg}(\pi) \triangleq \mathbb{E}\left[\sum_{t=1}^{T} a_t^\top y_t - \sum_{t=1}^{T}\pi(\mathcal{A}_t)^\top y_t\right] \leq \mathcal{O}\left(\gamma T + \frac{\ln|\Pi|}{\eta} + \eta dT\right). \tag{46}$$

The magnitude of the loss is bounded by

$$|\hat{\ell}_{t,\pi}| = \left|\left\langle \pi(\mathcal{A}_t), \tilde{H}_t^{-1}a_t\ell_t\right\rangle\right|$$

$$\leq \|\pi(\mathcal{A}_t)\|_{\tilde{H}_t^{-1}}\|a_t\|_{\tilde{H}_t^{-1}}$$

$$\leq \frac{1}{\gamma}\|\pi(\mathcal{A}_t)\|_{G_t^{-1}}\|a_t\|_{G_t^{-1}} \leq \frac{d}{\gamma}.$$

If $\gamma \geq 2d\eta$, then we have $|\hat{\ell}_{t,\pi}| \leq \frac{1}{2}$ and we can use the standard regret bound of exponential weights:

$$\forall \pi \in \Pi: \qquad \text{Reg}(\pi) \leq \gamma T + \frac{\ln|\Pi|}{\eta} + \eta\sum_{t=1}^{T}\mathbb{E}\left[\mathbb{E}_{a_t\sim p_t}\left[\sum_{\pi\in\Pi} P_{t,\pi}\hat{\ell}_{t,\pi}^2\right]\right].$$

Let $H_t = \mathbb{E}_{a\sim p_t}[aa^\top]$. Then we have $\tilde{H}_t^{-1} \preceq \frac{1}{1-\gamma}H_t^{-1}$, and thus

$$\mathbb{E}_{a_t\sim p_t}\left[\sum_{\pi\in\Pi} P_{t,\pi}\hat{\ell}_{t,\pi}^2\right] \leq \mathbb{E}_{a_t\sim p_t}\left[\sum_{\pi\in\Pi} P_{t,\pi}\cdot\langle\pi(\mathcal{A}_t),\tilde{H}_t^{-1}a_t\rangle^2\right]$$

$$= \mathbb{E}_{a_t\sim p_t}\mathbb{E}_{a\sim p_t}\left[\langle a, \tilde{H}_t^{-1}a_t\rangle^2\right] \qquad \text{(by the definition of } p_{t,a}\text{)}$$

$$\leq \frac{1}{(1-\gamma)^2}\text{Tr}\left(H_t H_t^{-1}H_t H_t^{-1}\right) = \mathcal{O}(d).$$

Combining all proves Eq. (46).

Next, we show that there exists $\theta \in \Theta$ such that

$$\mathbb{E}_{\mathcal{A} \sim D} \left[ \sum_{t=1}^{T} (\pi_\theta(\mathcal{A}) - \pi^\star(\mathcal{A}))^\top y_t \right] \leq \mathcal{O}(1). \tag{47}$$

Let $\hat{\theta}$ be the closest element in $\Theta$ to $\sum_{t=1}^{T} y_t$. By the definition of $\Theta$ and the assumption that $\|y_t\| \leq 1$, we have $\left\| \hat{\theta} - \sum_{t=1}^{T} y_t \right\| \leq \epsilon$. Thus, for any $\mathcal{A}$,

$$\sum_{t=1}^{T} (\pi_{\hat{\theta}}(\mathcal{A}) - \pi^\star(\mathcal{A}))^\top y_t \leq \sum_{a \in \mathcal{A}} (\pi_{\hat{\theta}}(\mathcal{A}) - \pi^\star(\mathcal{A}))^\top \hat{\theta} + \epsilon \leq \epsilon$$

where the last inequality is by the fact that $\pi_{\hat{\theta}}(\mathcal{A}) = \operatorname{argmin}_{a \in \mathcal{A}} a^\top \hat{\theta}$. Taking expectation over $\mathcal{A}$ gives Eq. (47).

Finally, combining Eq. (46) and Eq. (47), choosing $\epsilon = 1$ and $\gamma = 2d\eta = 2d\sqrt{\frac{\log T}{T}}$, we get

$$\begin{aligned}
\text{Reg} &= \mathbb{E} \left[ \sum_{t=1}^{T} a_t^\top y_t - \sum_{t=1}^{T} \pi^\star(\mathcal{A}_t)^\top y_t \right] \\
&= \mathbb{E} \left[ \sum_{t=1}^{T} a_t^\top y_t - \sum_{t=1}^{T} \pi_{\hat{\theta}}(\mathcal{A}_t)^\top y_t \right] + \mathbb{E}_{\mathcal{A} \sim D} \left[ \sum_{t=1}^{T} (\pi_{\hat{\theta}}(\mathcal{A}) - \pi^\star(\mathcal{A}))^\top y_t \right] \\
&= \mathcal{O} \left( \gamma T + \frac{\ln((2T)^d)}{\eta} + \eta dT + 1 \right) \\
&= \mathcal{O} \left( d\sqrt{T \log T} \right),
\end{aligned}$$

finishing the proof. $\qquad\square$

## G   Comparison with Dai et al. [2023] and Sherman et al. [2023]

We state the exponential weight algorithm adopted by Luo et al. [2021], Dai et al. [2023], Sherman et al. [2023] in Algorithm 5, which is an algorithm that we know to achieve the prior-art regret bound in our setting (though they studied a more general MDP setting).

Their algorithm proceeds in *epochs* (indexed by $k$), where every epoch consists of $W$ rounds. The policy on action set $\mathcal{A}$ in the $k$-th epoch is defined as

$$p_k^{\mathcal{A}}(a) \propto \exp \left( -\eta \sum_{s=1}^{k-1} (a^\top \hat{y}_s - b_s(a)) \right)$$

where $\hat{y}_k$ is the loss estimator for epoch $k$, and $b_k(a)$ is a (non-linear) bonus. In all $W$ rounds in epoch $k$, the same policy is executed. The samples obtained in these $W$ rounds are randomly divided into two halfs. One half is used to estimate the covariance matrix $\hat{\Sigma}_k$, and the other half is used to construct the loss estimator $\hat{y}_k$ (see Line 5 of Algorithm 5).

---

**Algorithm 5** Exponential weights with magnitude-reduced loss estimators

1 **for** $k = 1, 2, \ldots, \frac{T}{W}$ **do**

2     For all $\mathcal{A}$, define

$$p_k^{\mathcal{A}}(a) = \frac{\exp\left(-\eta \sum_{s=1}^{k-1} (a^\top \hat{y}_s - b_s(a))\right)}{\sum_{a' \in \mathcal{A}} \exp\left(-\eta \sum_{s=1}^{k-1} (a'^\top \hat{y}_s - b_s(a'))\right)} \qquad \text{for all } a \in \mathcal{A}.$$

3     Randomly partition $\{(k-1)W + 1, \ldots, kW\}$ into two equal parts $\mathcal{T}_k, \mathcal{T}_k'$.

4     **for** $t = (k-1)W + 1, \ldots, kW$ **do**
        receive $\mathcal{A}_t$, sample $a_t \sim p_k^{\mathcal{A}_t}$, and receive $\ell_t$.

5     Define

$$\hat{\Sigma}_k = \beta I + \frac{1}{|\mathcal{T}_k|} \sum_{t \in \mathcal{T}_k} a_t a_t^\top$$

$$\hat{y}_k = \hat{\Sigma}_k^{-1} \left( \frac{1}{|\mathcal{T}_k'|} \sum_{t \in \mathcal{T}_k'} a_t \ell_t \right)$$

$$b_k(a) = \alpha \|a\|_{\hat{\Sigma}_k^{-1}}.$$

---

### G.1 Regret Analysis Sketch

The regret analysis starts with a standard decomposition that is similar to ours. We abuse the notation by defining $y_k = \frac{1}{W} \sum_{t=(k-1)W}^{kW} y_t$. Then

$$\text{Reg} = W\mathbb{E}\left[ \sum_{k=1}^{T/W} p_k^{\mathcal{A}_0}(a) \langle a - u^{\mathcal{A}_0}, y_k \rangle \right]$$

$$= \underbrace{W\mathbb{E}\left[ \sum_{k=1}^{T/W} p_k^{\mathcal{A}_0}(a)\left( \langle a, \hat{y}_k \rangle - b_k(a) \right) - \left( u^{\mathcal{A}_0} - b_k(u^{\mathcal{A}_0}) \right) \right]}_{\textbf{EW-Reg}} + \underbrace{W\mathbb{E}\left[ \sum_{k=1}^{T/W} p_k^{\mathcal{A}_0}(a) b_k(a) - b_k(u^{\mathcal{A}_0}) \right]}_{\textbf{Bonus}}$$

$$+ \underbrace{W\mathbb{E}\left[ \sum_{k=1}^{T/W} p_k^{\mathcal{A}_0}(a)\langle a - u^{\mathcal{A}_0}, y_k - \hat{y}_k \rangle \right]}_{\textbf{Bias}}.$$

Bounding the regret term follows the standard analysis of exponential weight:

$$\textbf{EW-Reg} \leq W\mathbb{E}\left[ \frac{\ln|\mathcal{A}_0|}{\eta} + \eta \sum_{k=1}^{T/W} \sum_{a \in \mathcal{A}_0} p_k^{\mathcal{A}_0}(a)\langle a, \hat{y}_k \rangle^2 + \eta \sum_{k=1}^{T/W} \sum_{a \in \mathcal{A}_0} p_k^{\mathcal{A}_0}(a) b_k(a)^2 \right]$$

$$\leq W\mathbb{E}\left[ \frac{\ln|\mathcal{A}_0|}{\eta} + \eta \sum_{k=1}^{T/W} \sum_{a \in \mathcal{A}_0} p_k^{\mathcal{A}_0}(a) a^\top \hat{\Sigma}_k^{-1} H_k \hat{\Sigma}_k^{-1} a + \eta \sum_{k=1}^{T/W} \frac{\alpha^2}{\beta} \right]$$

where $H_k = \mathbb{E}_{\mathcal{A} \sim D} \mathbb{E}_{a \sim p_k^{\mathcal{A}}}[aa^\top]$. Then they use the following fact to bound the stability term: as long as $W \geq \frac{d}{\beta^2}$, it holds with high probability that $\hat{\Sigma}_k^{-1} H_k \hat{\Sigma}_k^{-1} \preceq 2\hat{\Sigma}_k^{-1}$. Thus **EW-Reg** can be

further bounded by

$$\textbf{EW-Reg} \lesssim W\left(\frac{\ln|\mathcal{A}_0|}{\eta} + \eta\mathbb{E}\left[\sum_{k=1}^{T/W}\sum_{a\in\mathcal{A}_0} p_k^{\mathcal{A}_0}(a)\|a\|_{\hat{\Sigma}_k^{-1}}^2\right] + \eta\frac{T}{W}\frac{\alpha^2}{\beta}\right)$$
$$\leq \frac{W\ln|\mathcal{A}_0|}{\eta} + \eta dT + \eta T\frac{\alpha^2}{\beta}.$$

By the definition of the bonus function $b_t$, it holds that

$$\textbf{Bonus} = W\mathbb{E}\left[\alpha\sum_{k=1}^{T/W}\sum_{a\in\mathcal{A}_0} p_k^{\mathcal{A}_0}(a)\|a\|_{\hat{\Sigma}_k^{-1}}\right] - W\mathbb{E}\left[\alpha\sum_{k=1}^{T/W}\|u^{\mathcal{A}_0}\|_{\hat{\Sigma}_k^{-1}}\right].$$

Finally, the bias term can be bounded as follows:

$$\textbf{Bias} = W\mathbb{E}\left[\sum_{k=1}^{T/W} p_k^{\mathcal{A}_0}(a)(a-u^{\mathcal{A}_0})^\top(y_k - \hat{\Sigma}_k^{-1}H_k y_k)\right]$$
$$= W\mathbb{E}\left[\sum_{k=1}^{T/W} p_k^{\mathcal{A}_0}(a)(a-u^{\mathcal{A}_0})^\top\hat{\Sigma}_k^{-1}(\hat{\Sigma}_k - H_k)y_k\right]$$
$$\leq W\mathbb{E}\left[\sum_{k=1}^{T/W} p_k^{\mathcal{A}_0}(a)\|a-u^{\mathcal{A}_0}\|_{\hat{\Sigma}_k^{-1}}\|(\hat{\Sigma}_k - H_k)y_k\|_{\hat{\Sigma}_k^{-1}}\right].$$

The bias here has a similar form as in our case. They use the following fact to bound the bias: as long as $W \geq \frac{d}{\beta^2}$, it holds that $\|(\hat{\Sigma}_k - H_k)y_k\|_{\hat{\Sigma}_k^{-1}} \leq \sqrt{\beta d}$. Therefore, the bias can further be upper bounded by

$$\textbf{Bias} \leq W\mathbb{E}\left[\sqrt{\beta d}\sum_{k=1}^{T/W}\sum_{a\in\mathcal{A}_0} p_k^{\mathcal{A}_0}(a)\|a\|_{\hat{\Sigma}_k^{-1}} + \sqrt{\beta d}\sum_{k=1}^{T/W}\|u^{\mathcal{A}_0}\|_{\hat{\Sigma}_k^{-1}}\right].$$

Combining the three parts, we get that the overall regret is of order

$$\mathbb{E}\left[\frac{W\ln|\mathcal{A}_0|}{\eta} + \eta dT + \eta T\frac{\alpha^2}{\beta} + W(\alpha + \sqrt{\beta d})\sum_{k=1}^{T/W}\sum_{a\in\mathcal{A}_0} p_k^{\mathcal{A}_0}(a)\|a\|_{\hat{\Sigma}_k^{-1}} + W(\sqrt{\beta d} - \alpha)\sum_{k=1}^{T/W}\|u^{\mathcal{A}_0}\|_{\hat{\Sigma}_k^{-1}}\right].$$

Choosing $\alpha \approx \sqrt{\beta d}$, we further bound it by

$$\mathbb{E}\left[\frac{W\ln|\mathcal{A}_0|}{\eta} + \eta dT + W\sqrt{\beta d}\sum_{k=1}^{T/W}\sum_{a\in\mathcal{A}_0} p_k^{\mathcal{A}_0}(a)\|a\|_{\hat{\Sigma}_k^{-1}}\right]$$
$$\leq \mathbb{E}\left[\frac{W\ln|\mathcal{A}_0|}{\eta} + \eta dT + W\sqrt{\beta d}\sum_{k=1}^{T/W}\sqrt{\sum_{a\in\mathcal{A}_0} p_k^{\mathcal{A}_0}(a)\|a\|_{\hat{\Sigma}_k^{-1}}^2}\right]$$
$$\leq \frac{W\ln|\mathcal{A}_0|}{\eta} + \eta dT + \sqrt{\beta}dT.$$

Recall the constraint $W \geq \frac{d}{\beta^2}$. Choosing $W = \frac{d}{\beta^2}$ gives

$$\frac{d\ln|\mathcal{A}_0|}{\eta\beta^2} + \eta dT + \sqrt{\beta}dT \tag{48}$$

which gives $d(\ln|\mathcal{A}_0|)^{\frac{1}{6}}T^{\frac{5}{6}}$ with the optimally chosen $\eta$ and $\beta$.

**Remark** Due to the restrictions on the magnitude of the loss estimator required by the exponential weight algorithm, there is actually another constraint $\frac{\eta}{\beta} \leq 1$, which makes Eq. (48) be $d(\ln|\mathcal{A}_0|)^{\frac{1}{7}}T^{\frac{6}{7}}$ at best. This is exactly the bound obtained by Sherman et al. [2023]. A more sophisticated way to construct $\hat{y}_k$ developed by Dai et al. [2023] removes this additional requirement and allows a bound of $d(\ln|\mathcal{A}_0|)^{\frac{1}{6}}T^{\frac{5}{6}}$. The sub-optimal bound $T^{\frac{8}{9}}$ reported in Dai et al. [2023] is due to issues related to MDPs, which are not presented in the contextual bandit case here.

