# Bypassing the Simulator:
# Near-Optimal Adversarial Linear Contextual Bandits

## Abstract

We consider the adversarial linear contextual bandit problem, where the loss vectors are selected fully adversarially and the per-round action set (i.e. the context) is drawn from a fixed distribution. Existing methods for this problem either require access to a simulator to generate free i.i.d. contexts, achieve a sub-optimal regret no better than $\widetilde{\mathcal{O}}(T^{5/6})$, or are computationally inefficient. We greatly improve these results by achieving a regret of $\widetilde{\mathcal{O}}(\sqrt{T})$ without a simulator, while maintaining computational efficiency when the action set in each round is small. In the special case of sleeping bandits with adversarial loss and stochastic arm availability, our result answers affirmatively the open question by [SGV20] on whether there exists a polynomial-time algorithm with $\mathrm{poly}(d)\sqrt{T}$ regret. Our approach naturally handles the case where the loss is linear up to an additive misspecification error, and our regret shows near-optimal dependence on the magnitude of the error.

## 1 Introduction

Contextual bandit is a widely used model for sequential decision making. The interaction between the learner and the environment proceeds in rounds: in each round, the environment provides a context; based on it, the learner chooses an action and receive a reward. The goal is to maximize the total reward across multiple rounds. This model has found extensive applications in fields such as medical treatment [TM17], personalized recommendations [BLL+11], and online advertising [CLRS11].

Algorithms for contextual bandits with provable guarantees have been developed under various assumptions. In the linear regime, the most extensively studied model is the *stochastic linear contextual bandit*, in which the context can be arbitrarily distributed in each round, while the reward is determined by a fixed linear function of the context-action pair. Near-optimal algorithms for this setting have been established in, e.g., [CLRS11, AYPS11, LWZ19, FGMZ20]. Another model, which is the focus of this paper, is the *adversarial linear contextual bandit*, in which the context is drawn from a fixed distribution, while the reward is determined by a time-varying linear function of the context-action pair. [1] A computationally efficient algorithm for this setting is first proposed by [NO20]. However, existing research for this setting still faces challenges in achieving near-optimal regret and sample complexity when the context distribution is unknown.

The algorithm by [NO20] requires the learner to have *full knowledge* on the context distribution, and access to an *exploratory policy* that induces a feature covariance matrix with a smallest eigenvalue at least $\lambda$. Under these assumptions, their algorithm provides a regret guarantee of $\widetilde{\mathcal{O}}(\sqrt{dT/\lambda})$,

---

[1] Apparently, the stochastic and adversarial linear contextual bandits defined here are incomparable, and their names do not fully capture their underlying assumptions. However, these are the terms commonly used in the literature (e.g., [AYPS11, NO20]).

Table 1: Related works in the "S-A" category. CB stands for contextual bandits and SB stands for semi-bandits. The relations among settings are as follows: Sleeping Bandit $\subset$ Contextual SB $\subset$ Linear CB, Linear CB $\subset$ Linear MDP, and Linear CB $\subset$ General CB. The table compares our results with the Pareto frontier of the literature. For algorithms dealing more general settings, we have carefully translated their techniques to Linear CB and reported the resulting bounds. $\Sigma_\pi$ denotes the feature covariance matrix induced by policy $\pi$. $|\mathcal{A}|$ and $|\Pi|$ are sizes of the action set and the policy set.

| Target Setting | Algorithm | Regret | Simulator | Computation | Assumption |
|---|---|---|---|---|---|
| General CB | [SLKS16] | $(\log|\Pi|)^{1/3}(|\mathcal{A}|T)^{2/3}$ | ✓ | poly$(|\mathcal{A}|, \log|\Pi|, T)$ | ERM Oracle |
| Linear MDP | [DLWZ23] | $\sqrt{dT\log|\mathcal{A}|}$ | ✓ | poly$(|\mathcal{A}|, d, T)$ | |
| | [DLWZ23, SKM23] | $d(\log|\mathcal{A}|)^{1/6}T^{5/6}$ | | poly$(|\mathcal{A}|, d, T)$ | |
| | [KZWL23] | $(d^7T^4)^{1/5} + \text{poly}\left(\frac{1}{\lambda}\right)$ | | $T^d$ | $\exists\pi, \Sigma_\pi \succeq \lambda I$ |
| Linear CB | Algorithm 1 | $d^2\sqrt{T}$ | | poly$(|\mathcal{A}|, d, T)$ | |
| | Algorithm 2 | $d\sqrt{T}$ | | $T^d$ | |
| Contextual SB | [NV14] | $(dT)^{2/3}$ | | poly$(d, T)$ | |
| Sleeping Bandit | [SGV20] | $\sqrt{2^dT}$ | | poly$(d, T)$ $(|\mathcal{A}| \le d)$ | |

where $d$ is the feature dimension and $T$ is the number of rounds. These assumptions are relaxed in the work of [LWL21], who studied a more general linear MDP setting. When specialized to linear contextual bandits, [LWL21] only requires access to a *simulator* from which the learner can draw free i.i.d. contexts. Their algorithm achieves a $\widetilde{\mathcal{O}}((dT)^{2/3})$ regret. The regret is further improved to the near-optimal one $\widetilde{\mathcal{O}}(d\sqrt{T})$ by [DLWZ23] through refined loss estimator construction.

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

---

[2]For combinatorial semi-bandit problems, our algorithm is not as computationally efficient as [NV14], which can handle exponentially large action sets.

## 1.2 Computational Complexity

Our main algorithm is based on log-determinant barrier optimization similar to [FGMZ20, ZL22]. Computing its action distribution is closely related to computing the D-optimal experimental design [KT90]. Per step, this is shown to require $\widetilde{\mathcal{O}}(|\mathcal{A}_t|\mathrm{poly}(d))$ computational and $\widetilde{\mathcal{O}}(\log(|\mathcal{A}_t|)\mathrm{poly}(d))$ memory complexity [FGMZ20, Prop 1], where $|\mathcal{A}_t|$ is the action set size at round $t$. The computational bottleneck comes from (approximately) maximizing a quadratic function over the action set. It is an open question whether linear optimization oracles or other type of oracles can lead to efficient implementation of our algorithm for continuous action sets.

On the other hand, we are unaware of *any* linear context bandit algorithm that provably avoids $|\mathcal{A}|$ computation per round while maintaining a $o(|\mathcal{A}|)$ regret dependence in the frequentist setting. The LinUCB algorithm [CLRS11, AYPS11]

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

---

[3]Since [Ano23] has not been published, for completeness, we restate all their results in Appendix E.2. The goal is to use their reduction idea to handle the unknown misspecification case. We do not claim our contribution in the reduction idea.