# OpenReview forum: "Bypassing the Simulator: Near-Optimal Adversarial Linear Contextual Bandits"
_NeurIPS.cc/2023/Conference — NeurIPS 2023 poster_

### Official Review · Reviewer_6s4U · 2023-07-04

**Soundness:** 3 good
**Presentation:** 3 good
**Contribution:** 3 good
**Rating:** 7
**Confidence:** 3

**Summary:**

The paper addresses the problem of adversarial linear contextual bandits, where loss vectors are selected adversarially and the context for each round is drawn from a fixed distribution; traditional approaches require access to a simulator for generating free i.i.d. contexts or achieve a sub-optimal regret no better than $\widetilde O(T^{5/6})$.

The authors significantly improve on these methods by achieving a regret of $\widetilde O(\sqrt{T})$ without the need for a simulator, while maintaining computational efficiency when the action set per round is small. The results affirmatively answer an open question about the existence of a polynomial-time algorithm with $\mathrm{poly}(d)\sqrt{T}$ regret. The approach allows for the case where the loss is linear up to an additive misspecification error, showing a near-optimal dependence on the magnitude of this error.

The paper also presents a computationally inefficient algorithm that provides an improved $\widetilde O(d\sqrt{T})$ regret without a simulator, expanding on the EXP4 algorithm.

**Strengths:**

The paper presents an algorithm that innovatively instantiates an individual Follow-The-Regularized-Leader (FTRL) algorithm on each action set. This development has a wide range of implications for fields. It promises more efficient and effective optimization algorithms in the face of adversarial environments.

One of the major strengths of the paper lies in the construction of loss estimators and feature covariance matrix estimators, especially in challenging situations where the learner has no knowledge about the context distribution and there is no simulator available.

The paper manages to achieve an improved bound on the bias. The techniques used surpass those used in previous studies [DLWZ23, SKM23], leading to a significant improvement in the algorithm's performance and making it more effective in practice.

The introduction of feature centralization is also a notable strength. By centralizing the features by $\hat x_t$, an estimation of the mean features under the current policy, the bias $y_t - \hat y_t$ appears in a nice form that can be compensated by a bonus term. This approach provides an elegant solution to handling bias in the estimator.

The paper successfully handles the issue of strong dependence between the policy and the empirical context distribution, Dˆt, which typically rules out canonical concentration bounds.

The paper addresses the need for high efficiency in reusing the context samples, which is a challenging problem often not adequately handled by existing methods.

**Weaknesses:**

A notable weakness of the paper is the absence of empirical or numerical experiments to validate the theoretical results. Without practical implementation and testing of the proposed estimators on different datasets, it becomes difficult to assess their efficacy and performance.

**Questions:**

How might these estimators handle high-dimensional data? Are there potential ways to improve their performance in such scenarios in future work?

Could the presented estimators be generalized to other types of bandit problems, such as non-linear contextual bandit problems?

**Limitations:**

The authors adequately addressed the limitations as I could tell.



==================================================

Post-Rebuttal:

I appreciate the response from the authors, and it addresses most of my concerns. Though I still believe the empirical experiments and the comparisons to other existing methods should be an important part of the work, this is still a strong theoretical paper.

---

> ### Author Rebuttal · Authors · 2023-08-06
>
> Thank you for the valuable feedback. We agree that empirical validation is important to prove the efficacy of the algorithm. It will be an important future work to experimentally compare our algorithm to other existing linear contextual bandit algorithms.
>
> $\textbf{Q1:}$ How might these estimators handle high-dimensional data? Are there potential ways to improve their performance in such scenarios in future work?
>
> $\textbf{Reply:}$ We do not have any constraint on the dimension $d$, so our algorithm can also be applied when $d$ is large. In general, a $\text{poly}(d)$ regret is inevitable. In some special case, a $\text{poly}(d)$ regret can be avoided. For example, for the case where the number of actions and the size of the policy set are small, the algorithm in [SLKS16] can be used to achieve a regret of $\tilde{\mathcal{O}}\left((\log|\Pi|)^{\frac{1}{3}}(|\mathcal{A}|T)^{\frac{2}{3}}\right)$, where $|\mathcal{A}|$ and $|\Pi|$ are the sizes of the action set and the policy set, respectively. Their algorithm relies on simulators though, and it would be an interesting direction to try to remove it.
>
> [SLKS16] Vasilis Syrgkanis, Haipeng Luo,Akshay Krishnamurthy, and Robert E Schapire. Improved regret bounds for oracle-based adversarial contextual bandits. Advances in Neural Information Processing Systems, 29, 2016.
>
>
> $\textbf{Q2:}$ Could the presented estimators be generalized to other types of bandit problems, such as non-linear contextual bandit problems?
>
> $\textbf{Reply:}$ The current algorithmic framework, including the estimator and the regularizer designs, is heavily tailored for the linear case. To generalize the problem to the non-linear case, we notice that even for the generalized linear bandit *without context*, there is no known computationally efficient algorithm that handles adversarial losses and has a regret bound of $\text{poly}(d)\times o(T)$. The difficulty exactly lies in the the construction of an unbiased low-dimensional loss estimator that can be shared among actions. The study of an efficient algorithm for particular non-linear bandit problems (e.g., generalized linear bandits) with adversarial loss is indeed an interesting direction.

---

### Official Review · Reviewer_f8CU · 2023-07-06

**Soundness:** 3 good
**Presentation:** 3 good
**Contribution:** 3 good
**Rating:** 6
**Confidence:** 2

**Summary:**

The paper studies an adversarial linear bandit problem. At each round, the adversary selects a hidden loss vector $y_t$, the environment 'stochastically generates' an action set $A_t$, and the learner chooses an action in the action set. The learner is able to observe noisy loss $\ip{y_t}{a_t}$.

**Strengths:**

Update after rebuttal. This paper proposed an optimal algorithm for the adversarial contextual linear bandit problem. The algorithm consists of OMD with logdet barrier as well as the lifting technique.

**Weaknesses:**

Update after rebuttal.
Can dependence on d be improved?

**Questions:**

First, I did not fully understand the problem setting studied in this paper and would need some clarifications.

1. In this work context vectors = actions, I have never seen this formulation in previous literature. The action set $A_t$ is generated from the distribution $D$. What exactly is distribution $D$? Is it defined over the power set of $B_2^d$? What assumption do you make on the size of $A_t$?

2. The formulation does not align with previous work (e.g. NO20). In NO20, they assume the action is chosen by an adversary for each arm in $[K]$ (action for arm $a$ is $\theta_{t,a}$), and the context $x_t$ is drawn from (known) distribution and revealed to the learner. The learner chooses one of the arms and observes the noisy loss centered at ${x_t^T}\cdot{\theta_{t, A_t}}$. Regret is compared to best policy, which is mapping from context to action.
This current work assumes the loss vector is chosen adversarially and the action set is drawn 'stochastically' (which I don't completely understand, see point above. ) Regret is compared to best policy, which is mapping from action set to a distribution over action set.
Given the different formulations, I do not believe the claims in this work, stating they improve over prior work, are accurate.

Based on my understanding of this work, if the action set $A_t$ were constant each round, then this is the bandit linear optimization problem, as in (competing in the dark, abernethy et al. ). Hence this paper really seems to be solving a bandit linear optimization problem where the action set is changing each round, and not the 'adversarial linear contextual bandit' problem as introduced in the intro section.

**Limitations:**

The author did not address limitations.

---

> ### Author Rebuttal · Authors · 2023-08-06
>
> Thank you for raising concerns about the clarity of the setting. We will clarify them in the new version.
>
> $\textbf{Q1:}$ In this work context vectors = actions, I have never seen this formulation in previous literature. The action set $A_t$ is generated from the distribution $D$. What exactly is the distribution $D$? Is it defined over the power set of $B_2^d$? What assumption do you make on the size of $A_t$?
>
>
> $\textbf{Reply:}$ Condensing the context into a time-varying action set is common in the literature of contextual linear bandits. See page 239 of [LS20], page 1 of [HYF23], or other previous works [CLRS11, LWCZ21] that also take this view. Furthermore, the alternative linear contextual bandit setting of [NO20] reduces to our setting as well (see the response for Q2).
> $D$ is a distribution over subsets of $B_2^d$. In other words, $D$ is a distribution over the power set of $B_2^d$. We only assume $A_t \subset B_2^d$ without any additional assumption on its size, so it can be infinite. The reviewer can also refer to [HYF23] for more characterization on $D$, which is the same as ours.
>
> [LS20] Tor Lattimore and Csaba Szepesvári. Bandit algorithms. Cambridge University Press, 2020.
>
> [HYF23] Osama Hanna, Lin Yang, Christina Fragouli. Contexts can be Cheap: Solving Stochastic Contextual Bandits with Linear Bandit Algorithms. COLT 2023.
>
> [CLRS11] Wei Chu, Lihong Li, Lev Reyzin, Robert Schapire. Contextual Bandits with Linear Payoff Functions. AISTAT 2011.
>
> [LWCZ21] Yingkai Li, Yining Wang, Xi Chen, Yuan Zhou. Tight Regret Bounds for Infinite-armed Linear Contextual Bandits. AISTAT 2021.
>
>
> $\textbf{Q2:}$ The formulation does not align with previous work (e.g. NO20). I do not believe the claims in this work, stating they improve over prior work, are accurate.
>
> $\textbf{Reply: }$ In fact, the setting of [NO20] is a special case of our setting. [NO20] assume that at every round $t$, the environment picks an adversarial loss vector $\theta_{t,a}$ for any action $a \in [K]$, and a random context $X_t\in\mathbb{R}^d$ is drawn from a fixed distribution. The learner chooses an action $a_t\in[K]$ based on $X_t$ and the final observed loss is $\ell_t = \langle X_t, \theta_{t,a_t} \rangle$. To convert this to our formulation, define $\theta_t = \left[\theta_{t,1}, \cdots, \theta_{t,K}\right] \in \mathbb{R}^{d \times K}$. Then we have $\ell_t = \langle X_t, \theta_{t,a_t} \rangle = X_t^\top \theta_t e_{a_t} = \text{Tr}(e_{a_t}X_t^\top \theta_t)  = \langle X_te_{a_t}^\top,  \theta_t \rangle = \langle \text{Vec}(X_te_{a_t}^\top),  \text{Vec}(\theta_t) \rangle $ where $\text{Vec}(X)$ is the column vector generating by stacking the columns of matrix $X$, and $e_i$ is the $i$-th standard basis. Then by defining the action set $A_t = ${$\text{Vec}(X_te_{i}^\top), i \in [K]$} and adversarial loss vector $y_t =  \text{Vec}(\theta_t)$, their setting becomes a special case of our setting with dimension $dK$. Our formulation is more flexible than that in [NO20] because in their formulation, the number of parameters always scales with the number of actions (i.e., $d\times K$), while in our formulation, the number of parameters is always $d$. Thus, our formulation is able to capture problems with large number of actions but small intrinsic dimension, while theirs cannot.
>
> Based on the discussion above, if we apply our algorithm to their problem, our $d$ becomes $dK$, and the regret bound is $\tilde{\mathcal{O}}\left(d^2K^2\sqrt{T}\right)$, which has worse $d$ and $K$ dependences than [NO20], but removes an inverse smallest eigenvalue of the covariance matrix in their bound.  However, we emphasize that our goal is not to improve the regret bound in [NO20], but to remove strong assumptions made in previous work on the knowledge of context distribution. Actually, as discussed in the introduction, [DLWZ23] has already strictly improved [NO20] (simultaneously improving the regret bound, considering a more general setting, and weakening assumptions), and got a near-optimal bound $\tilde{\mathcal{O}}\left(\sqrt{dT\log K}\right)$ (or $\tilde{O}(\sqrt{dKT\log K})$ in the formulation of [NO20]) with simulators. Therefore, in Table 1, we only compare our results with [DLWZ23].
>
>  [DLWZ23] Yan Dai, Haipeng Luo, Chen-Yu Wei, and Julian Zimmert. Refined regret for adversarial mdps with linear function approximation. arXiv preprint arXiv:2301.12942,374 2023.

---

> > ### Author Response · Authors · 2023-08-18
> >
> > We thank the reviewer for the time and effort spent on reviewing our paper. As the discussion phase is about to end, we want to ensure our responses have adequately addressed your inquiries. We look forward to your feedback.

---

> > ### Comment · Reviewer_f8CU · 2023-08-19
> >
> > Thanks for the response, it addressed my questions. I was initially confused since there are different formulations for 'linear bandit' vs 'linear contextual bandit', but now it is clear. I have adjusted my score.

---

> > > ### Author Response · Authors · 2023-08-19
> > >
> > > We thank the reviewer for the update and the positive evaluation. For the new question in the updated review, we answer it as below.
> > >
> > > $\textbf{Q3:}$  Can dependence on d be improved?
> > >
> > > $\textbf{Reply:}$  It is possible that the additional $d$ dependence can be removed, but it requires new techniques.  The additional $d$ dependence comes from a union bound over $(1/\epsilon)^{O(d^2)}$ policies when dealing with the covariance matrix estimation error, since the policy used by the algorithm is parameterized by $O(d^2)$ parameters. Due to the strong dependency between the contexts received in previous rounds and the the current policy, we do not believe that such a union bound can be avoided. We suspect that it might be possible to improve the bound to $\tilde{O}(\sqrt{d^3T})$. Recall that the reason we have $(1/\epsilon)^{O(d^2)}$ is because we lift the problem to $(d+1)^2$ dimension and the bonus term is a $(d+1)\times(d+1)$ matrix. However, there are other ways of imposing bonus in *linear bandits* (without contexts) that allows the bonus term to be in $(d+1)$ dimension ([LLWZ20] and [ZL22]), which potentially only requires a union bound over $(1/\epsilon)^{O(d)}$ policies. To achieve the optimal $\tilde{O}(d\sqrt{T})$ bound, we might need a very different algorithm design that does not rely on concentration arguments.
> > >
> > > [LLWZ20] Chung-Wei Lee, Haipeng Luo, Chen-Yu Wei, Mengxiao Zhang. Bias no more: high-probability data-dependent regret bounds for adversarial bandits and mdps. NeurIPS 2020.
> > >
> > > [ZL22] Return of the bias: Almost minimax optimal high probability bounds for adversarial linear bandits. COLT 2022.

---

### Official Review · Reviewer_V7BF · 2023-07-07

**Soundness:** 4 excellent
**Presentation:** 4 excellent
**Contribution:** 4 excellent
**Rating:** 7
**Confidence:** 4

**Summary:**

Summary:
This paper presents a near-optimal computationally efficient simulator-free algorithm in contextual bandits setting with i.i.d contexts and adversarial reward functions. Most of the past work on this setting either requires a simulator that allows them to draw a large number of contexts from a distribution, or is computationally inefficient. Existing algorithms that do not require a simulator and are computationally efficient only achieve a regret of O(T^{5/6}). This work greatly improves the dependence on T, achieving a regret of O(d^2 \sqrt{T}), while still computationally efficient without a simulator. Their algorithm is based on FTRL with a log-determinant barrier. They also present an algorithm that improves the dependence on d, achieving a regret of O(d \sqrt{T}) but is only computationally efficient for a finite policy class.


**Strengths:**

I think this paper makes a solid theoretical contribution to the field. The improvement from O(T^{5/6}) to O(\sqrt{T}) is substantial and simulator-free computationally efficient algorithms are important. The authors exploit the algorithmic ideas of ZL22, the ghost sample trick of NO20, along with novel analysis to provide an improved regret bound. I am slightly concerned about the originality of the algorithmic approach - but this is made up for with the improved analysis and improved regret bound

The paper was clearly well written and the contributions clearly stated. I enjoyed how the authors decomposed the regret and the analysis of the estimator in 3.2-3.4.


**Weaknesses:**

The main weakness is that the paper was challenging to read inplaces, especially for readers not already familiar with the ideas of ZL22. There were other places where a bit of extra detail could have helped the exposition greatly. I have tried to remark on these in the list below.

Remarks/Comments
1. On bottom of page 5, you mention the loss in the “original space”, can you explain precisely what that loss is? For readers not familiar with ZL22, Line 2 in the algorithm may be opaque - in particular the sequence H_s^{-1} is chosen to control the bias. Moving a bit of the discussion from 3.4/3.5 up - or even simply remarking that it will be explained later may help.
2. The discussion on line 188-189 is similar to 193-194. You can drop one.
3. In equation 2, you use a \hat{Sigma} - however the point of this equations seems to be that Sigma is known! Maybe remove the empirical hat?
4. Is it possible to elaborate a bit more in the paper how your methods “surpass those in [DLWZ23,SKM23]” as described in line 246?
5. \gamma_t should be defined in algorithm for readability


**Questions:**

1. To analyze the first term |\hat{Sigma}_t - H_ty_t|_hat{Sigma}_t^{-1} in line 242 did you need to use any kind of matrix concentration? If so, did you need control of the minimal eigenvalue of H_t? If not, how were you able to avoid it?
2. The linear EXP4 algorithm looks very much similar to the original EXP4 algorithm. Could you provide more explanation of why we cannot directly use other computationally inefficient benchmark algorithms to get the same regret, i.e. why it’s necessary to present and prove linear EXP4?
3. The computationally efficient algorithm, Algorithm 1, achieves a O(d^2\sqrt{T}) regret while having a squared dependence on d; while the algorithm that achieves O(d\sqrt{T}) regret is optimal but computationally inefficient. Do you believe algorithm 1 could achieve a O(d\sqrt(T)) regret?

---

> ### Author Rebuttal · Authors · 2023-08-06
>
> Thanks for providing suggestions on improving the readability of the paper. We will incorporate them in the final version.
>
> $\textbf{Q1:}$ To control the term $||(\hat{\Sigma}_t - H_t) y_t||\_{\hat{\Sigma}_t^{-1}}$, did you need to use any kind of matrix concentration? If so, did you need control of the minimal eigenvalue of $H_t$? If not, how were you able to avoid it?
>
> $\textbf{Reply: }$ We elaborate on the main idea of our proof here and more detail can be found in the proof of Lemma 14 in the full version of our paper. To control this term, we do not use any off-the-shelf matrix concentration inequality. Instead, we expand $\text{Tr}\left((\hat{\Sigma}_t - H_t)^\top\hat{\Sigma}_t^{-1}(\hat{\Sigma}_t - H_t)\right)$ by its definition and use the standard Bernstein’s inequality for scalars to handle it.
>
> More details are below. First, we show $\text{Tr}\left((\hat{\Sigma}\_t - H_t)^\top\hat{\Sigma}\_t^{-1}(\hat{\Sigma}\_t - H_t)\right) \leq 8\text{Tr}\left((\hat{\Sigma}\_t - H_t)^\top(H_t + \beta_t I)^{-1}(\hat{\Sigma}\_t - H_t)\right)$ where $\beta_t=\Theta(\text{poly}(d)/t)$ by some simple concentration inequality. Below, for simplicity, we assume that $H_t$ is a diagonal matrix (otherwise, we can diagonalize it as in the formal proof of Lemma 14). Recall that $\hat{\Sigma}\_t$ is equal to  $\beta_t I$ plus a covariance matrix estimated from the empirical context distribution. Therefore, we have $\hat{\Sigma}\_t - H_t = \Delta_t + \beta_t I$, where $\Delta_t$ is the covariance matrix estimation error related to the difference between the empirical and the true context distribution at time $t$. By direct expansion and the assumption that $H_t$ is diagonal, we have $\text{Tr}\left((\hat{\Sigma}\_t - H_t)^\top(H_t + \beta_t I)^{-1}(\hat{\Sigma}\_t - H_t)\right)=\sum_{i=1}^d \left(\frac{(\Delta_t(i,i) + \beta_t)^2}{H_t(i) + \beta_t} +  \sum_{j \neq  i} \frac{\Delta_t(i,j)^2}{H_t(i) + \beta_t}\right)\leq 2\sum_{i=1}^d \sum_{j=1}^d \frac{\Delta_t(i,j)^2}{H_t(i)+\beta_t} + 2d\beta_t$, where $\Delta_t(i,j)$ is the $(i,j)$-th entry of $\Delta_t$ and $H_t(i)$ is the $(i,i)$-th entry of $H_t$.
>
>
> Finally, we argue that with high probability $\sum_{j=1}^d \Delta_t(i,j)^2\leq \text{poly}(d)\times\tilde{O}(\frac{H_t(i)}{t} + \frac{1}{t^2})$, which, when combined with the argument above, gives the desired bound. The key calculations for this inequality are in Line 610-620. To establish this inequality, we have to use the fact that $\Delta_t(i,j)$ is an average of a martingale difference sequence and so we can use Bernstein's inequality to relate it to the variance of individual terms.  On the other hand, $H_t(i)$ is related to the variance of the individual terms in this martingale difference sequence. The inequality can be shown by combining these two components.
>
> From a high level, the key we are able to avoid the minimal eigenvalue of $H_t$ is the fact that in those directions where $H_t + \beta_t I$ has small eigenvalues, the error term $\hat{\Sigma}_t- H_t=\Delta_t + \beta_t I$ is also small.
>
>
> $\textbf{Q2:}$ The linear EXP4 algorithm looks very much similar to the original EXP4 algorithm. Could you provide more explanation of why we cannot directly use other computationally inefficient benchmark algorithms to get the same regret, i.e. why it’s necessary to present and prove linear EXP4?
>
> $\textbf{Reply:}$ We want to emphasize that we do not claim linear EXP4 as an important contribution of this paper, as this is indeed a fairly straightforward adaptation of the classical EXP4.
> Vanilla EXP4 and all benchmark algorithms we are aware of would suffer regret of $\sqrt{KT\ln(|\Pi|)}$, where $|\Pi|=\Theta(T^d)$ is the $1/T$-net of policies and $K$ the maximal number of actions. Since we want to handle the setting $K\gg d$, we have to present a modification that utilizes the linear feedback structure of the actions.
>
>
> $\textbf{Q3:}$ Algorithm 1 achieves a $O(d^2\sqrt{T})$ regret. Do you believe algorithm 1 could achieve a $O(d\sqrt{T})$ regret?
>
> $\textbf{Reply: }$ We are unsure whether the additional $d$ dependence can be removed. The additional $d$ dependence comes from a union bound over $(1/\epsilon)^{O(d^2)}$ policies when dealing with the covariance matrix estimation error, since the policy used by the algorithm is parameterized by $O(d^2)$ parameters. Due to the strong dependency between the contexts received in previous rounds and the the current policy, we do not believe that such a union bound can be avoided. We suspect that it might be possible to improve the bound to $\tilde{O}(\sqrt{d^3T})$. Recall that the reason we have $(1/\epsilon)^{O(d^2)}$ is because we lift the problem to $(d+1)^2$ dimension and the bonus term is a $(d+1)\times(d+1)$ matrix. However, there are other ways of imposing bonus in $\textit{linear bandits}$ that allows the bonus term to be in $(d+1)$ dimension ([LLWZ20] and [ZL22]), which potentially only requires a union bound over $(1/\epsilon)^{O(d)}$ policies. To achieve the optimal $\tilde{O}(d\sqrt{T})$ bound, we might need a very different algorithm design that does not rely on concentration arguments.
>
> [DLWZ23] Yan Dai, Haipeng Luo, Chen-Yu Wei, and Julian Zimmert. Refined regret for adversarial mdps with linear function approximation. arXiv preprint arXiv:2301.12942,374 2023.
>
> [LLWZ20] Chung-Wei Lee, Haipeng Luo, Chen-Yu Wei, Mengxiao Zhang. Bias no more: high-probability data-dependent regret bounds for adversarial bandits and mdps. NeurIPS 2020.
>
> [ZL22] Return of the bias: Almost minimax optimal high probability bounds for adversarial linear bandits. COLT 2022.

---

> > ### Comment · Reviewer_V7BF · 2023-08-15
> >
> > Thanks for the detailed response and walking me through the minimum eigenvalue concerns. I will leave my score where it is.

---

### Official Review · Reviewer_He1a · 2023-07-07

**Soundness:** 3 good
**Presentation:** 3 good
**Contribution:** 4 excellent
**Rating:** 8
**Confidence:** 4

**Summary:**

This paper studied the contextual bandits with i.i.d. contexts and adversarial linear loss functions without simulators and proposed a follow-the-regularized-leader with log-determinant  barrier (Logdet-FTRL) algorithm with a carefully designed covariance matrix estimator that is computationally efficient given small contexts set and achieves order-optimal (in T) regret \tilde{O}(d^2 \sqrt{T}). Previous algorithms achieving \sqrt{T} regret either require the knolwedge of contexts distribution or a simulator from which the learner can learn the contexts distribution by drawing free i.i.d. contexts. The proposed algorithm estimates the key covariance matrix via previously observes contexts with centralization and ridge regularizer.

The setting contains stochastic sleeping bandits as special case and the proposed algorithm  answers the open problem in stochastic sleeping bandits affirmtively. A computationally inefficient algorithm that achieves regret with improved d factor is also proposed.

**Strengths:**

The analysis is novel and the contribution is significant to linear adversarial bandits with i.i.d. contexts without simulator.
Multiple related scenarios and extentions are discussed.
I went over the proofs of the main regret analysis in App. D and several technical lemmas. The proofs are sound to me.

**Weaknesses:**

I do not see perticular weaknesses of the current version.

**Questions:**

Minor typos:
Line 164, has been taken
The second equality between line 796 and line 797, U^{\Ac_0} instead of \bar{U}^{\Ac_0}.

**Limitations:**

Yes.

---

> ### Author Rebuttal · Authors · 2023-08-06
>
> We thank the reviewer for the support and valuable feedback. We have changed the typos you mentioned in our new version.

---

### Decision · Program_Chairs · 2023-09-21

**Decision:**

Accept (poster)

**Comment:**

All reviewers agree that this is a good theoretical contribution to the bandit literature with a good motivation.